# Global increase in tropical cyclone ocean surface waves

Jian Shi [1,2], Xiangbo Feng [3,4], Ralf Toumi [4], Chi Zhang[2,5], Kevin I. Hodges[3], Aifeng Tao[1,2], Wei Zhang [2,5] & Jinhai Zheng [1,2] ✉

The long-term changes of ocean surface waves associated with tropical cyclones (TCs) are poorly observed and understood. Here, we present the global trend analysis of TC waves for 1979–2022 based on the ERA5 wave reanalysis. The maximum height and the area of the TC wave footprint in the six h reanalysis have increased globally by about 3%/decade and 6%/decade, respectively. The TC wave energy transferred at the interface from the atmosphere to the ocean has increased globally by about 9%/decade, which is three times larger than that reported for all waves. The global energy changes are mostly driven by the growing area of the wave footprint. Our study shows that the TC-associated wave hazard has increased significantly and these changes are larger than those of the TC maximum wind speed. This suggests that the wave hazard should be a concern in the future.

Tropical cyclones (TCs) cause extensive damage through strong winds and heavy rainfall. They also generate destructive oceanic extremes, such as storm surges and surface waves[1–3], which pose a significant threat to infrastructure, navigation, and communities[4–10]. Even in the absence of storm surges, coastal and offshore waves generated by TCs can still cause dangerous surf conditions, rip currents, and severe coastal erosion[10,11]. Surface waves can be an important contributor to extreme sea levels, which are recognised as a major natural hazard in the present and future. Compared to atmospheric extremes (e.g., heatwaves and heavy rainfall), long-term changes in oceanic extremes have not been well studied, due to the sparsity and uncertainty in ocean observations under extreme conditions[12–16].

Ocean surface waves are fundamentally forced by winds, but these two cross-medium fields at the interface do not necessarily have the same trends[17,18], due to the complicated nonlinear wind-wave interaction and the long persistence of swells. Satellite altimeters show a global increase in ocean surface wind speeds over the last four decades, but no significant trend has been found in global wave height largely because of the inhomogeneous changes in regional waves[19–22]. However, global wave energy has increased by about 4%/decade[23]. TCs are an important contributor to the global wave climate, especially for

extreme waves. Unlike the overall ocean surface waves, or the high-latitude extreme waves driven by extratropical cyclones embedded in strong westerlies, TC-associated waves only occur in a restricted area centred at the TC track positions. It is important to develop an analysis approach that is tailored to capture the synoptic features of TC waves (TCWs) and also mitigates the uncertainties in extreme wave values. Then, the next science questions will be: Are global trends in such TC-associated waves over recent decades detectable? and if so, how comparable are these trends relative to the overall wave trends and relative to the TC intensity changes?

Observed trends at both global and basin scales over the last decades have recently been found for some TC metrics, such as frequency, position, intensity, and translation speed[24–29]. The signs and values of these trends depend on the TC metric and can vary greatly with the basins. Attribution of these changes to anthropogenic forcing is problematic because of the uncertainties in TC observations and the effect of interdecadal climate variability[30,31]. TCWs are affected by multiple properties of TCs (such as intensity, size, and translation speed). There have been model studies examining the future impact of global warming on TCWs[32]. But, it remains unknown whether the TC wave conditions have actually changed over the past decades.

[1]Key Laboratory of Ministry of Education for Coastal Disaster and Protection, Hohai University, Nanjing, China. [2]College of Harbor, Coastal and Offshore Engineering, Hohai University, Nanjing, China. [3]National Centre for Atmospheric Science and Department of Meteorology, University of Reading, Reading, UK. [4]Department of Physics, Imperial College London, London, UK. [5]The National Key Laboratory of Water Disaster Prevention, Nanjing, China. ✉e-mail: jhzheng@hhu.edu.cn

Documenting any long-term trends of TCWs will be a significant addition to understand the links between climate change and TC-related hazards.

In this study, we examine the historical global long-term changes in TCWs by synthesising the latest ocean wave reanalysis with TC observations (see subsections 1 and 2 in the Methods section for details). To mitigate uncertainty of extremely high waves in global ocean wave reanalysis, we focus on relative changes in the TC wave footprint that are less sensitive to absolute values of extreme waves (see subsections 3, 5–7 in the Methods section for details). The TC wave footprint is defined by a threshold of significant wave height (Hs) that the wave reanalysis can well represent under TC conditions. The TC wave footprint also boosts the sample size of TC wave data, benefiting detection of robust long-term trends. The climatology of TCWs is summarised in Supplementary Fig. 1.

We have found in this study that over the last 44 years the annual mean of maximum Hs within the 6-h TC footprint has increased by 3.2 ± 1.3%/decade globally, relative to the 44-year mean (± representing the 95% confidence interval of trend value, with the statistical details in "Statistical analyses" in the Methods section). This relative increase of the maximum wave height is about 40% larger than the increase of TC surface maximum wind speed (2.3 ± 0.8%/decade). All ocean basins show a significant long-term increase of the maximum wave height, with the largest increase of 5.0 ± 4.2%/decade in the North Atlantic (NA). We also find that the area of TC wave footprint has increased by 5.7 ± 3.8%/decade globally. The TC wave energy, which measures the

total accumulated energy of TCs transferred from the atmosphere to the ocean within the TC wave footprint, has increased by 8.9 ± 7.3%/decade globally, with the fastest increasing rates (17–32%/decade) in the NA, eastern Pacific (EP) and North Indian Ocean (NI). The large upward trend of the global wave energy is mainly caused by the increase of the area of TCWs. The western North Pacific (WNP) and South Pacific (SP) show no significant basin-wide trends in the energy over the last 44 years. This is due to a reduction of annual accumulated TC duration time that counteracts the upward trends in the height and the area of TCWs.

## Results

### Global increase of the height and the area of TC waves

Figure 1a, b, d, e shows the composite height of the 6-h TC wave footprint around the TC track position for two equal epochs of the satellite era (1979–2000 and 2001–2022) in the Northern Hemisphere (NH) and Southern Hemisphere (SH). In the composite analysis, the 6-h wave fields are collected and averaged around the TC track position over each epoch. The height of TCWs distributes asymmetrically around the storm centre, with larger values to the right of the centre in the NH and to the left in the SH, related to the longer fetch for wave generation along the storm propagation direction[33]. Between the two epochs, the average height of TCWs within the footprint has increased by up to 44 cm in the NH and 51 cm in the SH, which are equivalent to 13.1% and 14.9% relative to the values in the first epoch. The spatial scale and magnitudes of TCWs in the SH are larger than those in the

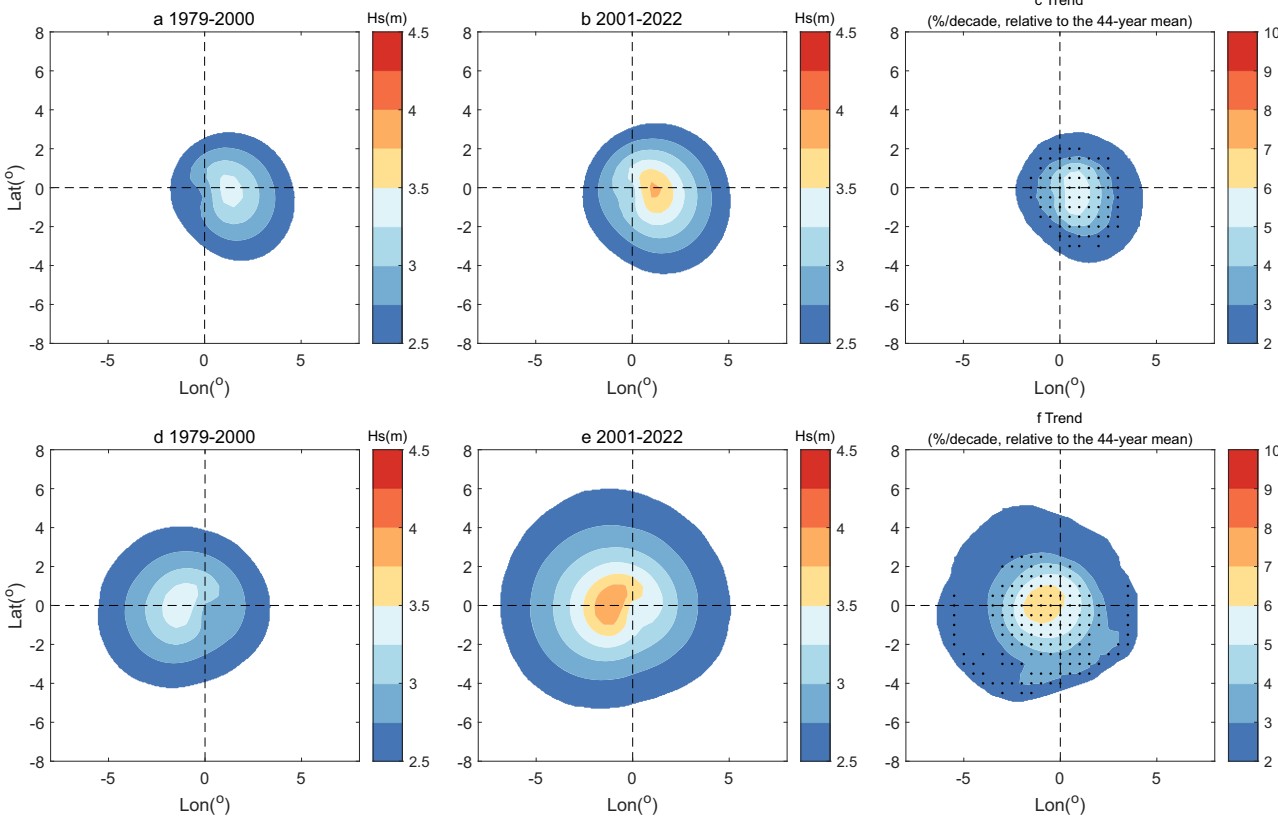

**Fig. 1 | Composite mean and linear trends of the height of tropical cyclone (TC) wave footprint. a–b** Composite mean of the 6-h wave height (m) around the TC track position in the Northern Hemisphere (NH), for the two epochs 1979–2000 and 2001–2022. **c** Relative trend (%/decade) of the annual 6-h wave height around the TC track position in the NH, over 1979–2022. The relative trend is relative to the 44-year mean; and dotted areas pass the 95% confidence level. **d–f** As (**a–c**) but for the Southern Hemisphere (SH). In (**a, b, d, e**), the 6-h wave height centred at the TC track position is averaged over the two epochs, and only the average

values above 2.5 m are plotted. In (**c, f**) the 6-h wave height centred at the TC track position is averaged in each year, and then the linear relative trend over the 44 years is computed at each grid point; only relative trends at grid points where the 44-year average above 2.5 m are plotted. The wave fields have been rotated in the TC orientation. Only TC waves for which the TC track points are between 40 °N and 40 °S are considered. Wave height is represented by significant wave height (Hs). Source data are provided as a Source Data file.

NH. In the SH, related to the lack of land, the effective wind fetch is much longer, resulting in larger waves[8]. In contrast, in the NH, because the fetch is shorter, the waves are relatively smaller. Figure 1c, f shows the linear relative trend in the 6-h height within the TC wave footprint over the whole period 1979–2022, relative to the long-term mean. The largest trends of about 6–7%/decade are seen to the right of the centre in the NH and to the left in the SH.

We find that the maximum height of TCWs has significantly increased both globally and in individual basins. The maximum height is defined as the maximum value of Hs within the 6-h footprint. The TCWs are further partitioned to swell and wind sea waves depending on the relationship between the local wind forcing and wave direction. Details on the wave metrics are provided in "ERA5 wave reanalysis" in the Methods section. The global average of the maximum height of the 6-h TC wave footprint for each year is shown in Fig. 2a. Relative to the 44-year mean, the maximum height has significantly increased by $3.2 \pm 1.3$%/decade, with $2.8 \pm 1.2$ and $3.5 \pm 1.4$%/decade for swell and wind sea waves, respectively (Supplementary Table 1). The maximum height has a larger trend value in the SH ($4.4 \pm 1.9$%/decade) than in the NH ($2.9 \pm 1.6$%/decade) (Fig. 2b, c, Supplementary Table 1). We note that the trends of the maximum height are much larger than the growth rates of TC intensity, which are only $2.3 \pm 0.8$, $2.3 \pm 0.9$ and $2.5 \pm 1.2$%/decade for the globe, NH and SH, respectively (Supplementary Table 1). This discrepancy suggests that the change of storm intensity cannot fully explain the large increase of the maximum height of TCWs and that other factors may also play a role. These factors include changes of TC translation speed and the nonlinear wind-wave interaction (this will be discussed later). To test the first factor, we calculated the linear trends of TC translation speed at global and hemispheric scales (Supplementary Fig. 2). The decreasing trends of the translation speed are noted but not significant. The decreasing trend of TC translation speed has been reported in several previous studies[28,29], but the trend has low confidence related to the analysed period and sources of track data[34,35], and this is consistent with our analysis. We also calculated the interannual (detrended) correlation between the maximum height of TCWs and the TC translation speed. The correlation is weak and not significant at global and hemispheric scales ($r = -0.18$, $-0.25$ and $-0.10$, $p > 0.05$, for the global, NH and SH averages, respectively), thus only hinting at a small contribution from the reduced translation speed to the large increase of TC wave height.

The basin-wide trends in the maximum height of TCWs and TC intensity are provided in Supplementary Fig. 3 and Supplementary Table 1. The upward trend of the maximum height is significant in all ocean basins, in the range 3.2–5.0%/decade, with the largest values in the NA, WNP and South Indian Ocean (SI). We notice that the WNP basin has the largest change of TC intensity, while the NA basin has the largest change of TC wave height, confirming that other factors affect the trends of the maximum height (this will be discussed in the discussion section).

We highlight that the area of TCWs has significantly increased in the satellite era (Fig. 1). We further quantify the global increase of the area for different thresholds of TC wave heights (Fig. 3a). The area trends in individual basins are provided in Supplementary Table 2. For the TCWs with Hs ≥ 2.5 m, the area increases by $5.7 \pm 3.8$%/decade, with respect to the 44-year mean. When raising the threshold from 2.5 m to 4.5 m, the relative trends of the area become slightly smaller, e.g., with $5.5 \pm 5.7$%/decade for the threshold of 4.5 m. However, the larger interannual variability with the larger height threshold tends to weaken the robustness of the trend (this is shown by the larger error bars, or lower confidence levels, of the trend values in Supplementary Table 2). We notice that these relative trends in TC wave area are 2–3 times larger than the relative trends of TC intensity (surface maximum wind speed, at $2.3 \pm 0.8$%/decade). The area trends are bigger in the SH than in the NH, especially for larger thresholds (Fig. 3b, c). The SI is an important contributor to the SH signal and there the relative trends of the area are very sensitive to the height threshold (i.e., the trend values are larger when using larger height thresholds), while in the NA and WNP the relative trends of the area are only weakly dependent on the threshold (Supplementary Table 2). This basin difference may be due to the smaller areas and the strong TC intensity in the NA and WNP compared to other basins (Supplementary Discussion and Supplementary Fig. 1) so that the proportion of high waves in the footprint in these two basins is more likely to be saturated with increased storm intensity.

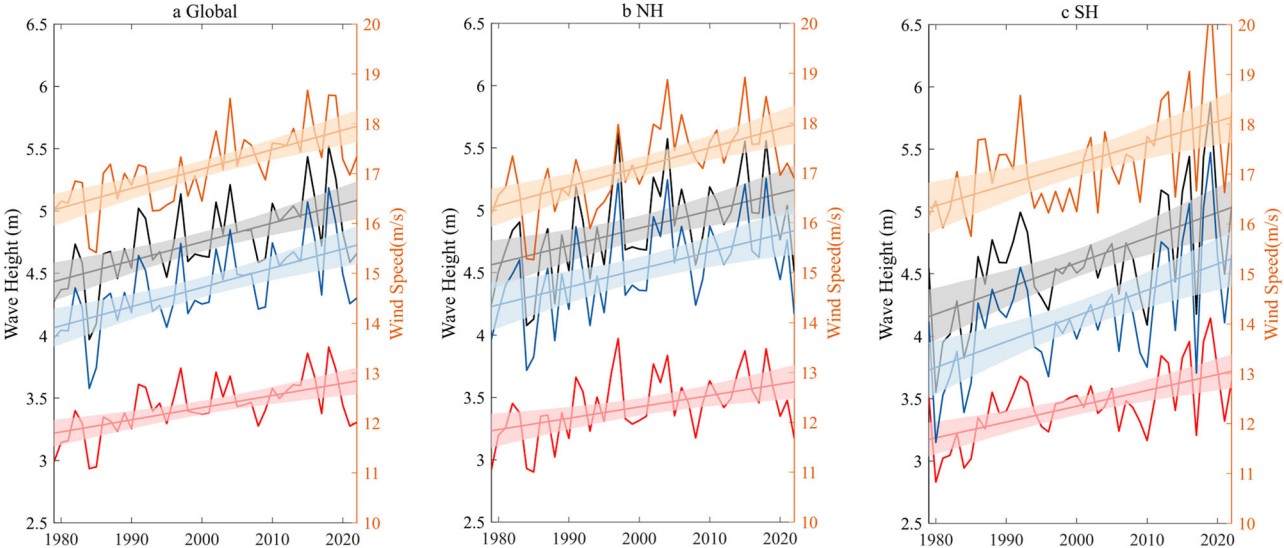

**Fig. 2 | Time series and linear trends of the maximum height of tropical cyclone (TC) wave footprint, and surface maximum wind speed. a** Global trend (straight line) and time series (solid line) of the annual 6-h maximum height for mixed (black), swell (red) and wind (blue) sea waves over 1979–2022. Annual 6-h surface maximum wind speed of TCs (orange) is also provided. **b, c** As (**a**), but for the Northern Hemisphere (NH) and Southern Hemisphere (SH). Shading shows the 95% confidence interval for the significant trend. Wave height is represented by significant wave height (Hs). Maximum height is defined by the maximum value of Hs within the 6-h TC wave footprint; the annual 6-h maximum height is the maximum height averaged over each year. Source data are provided as a Source Data file.

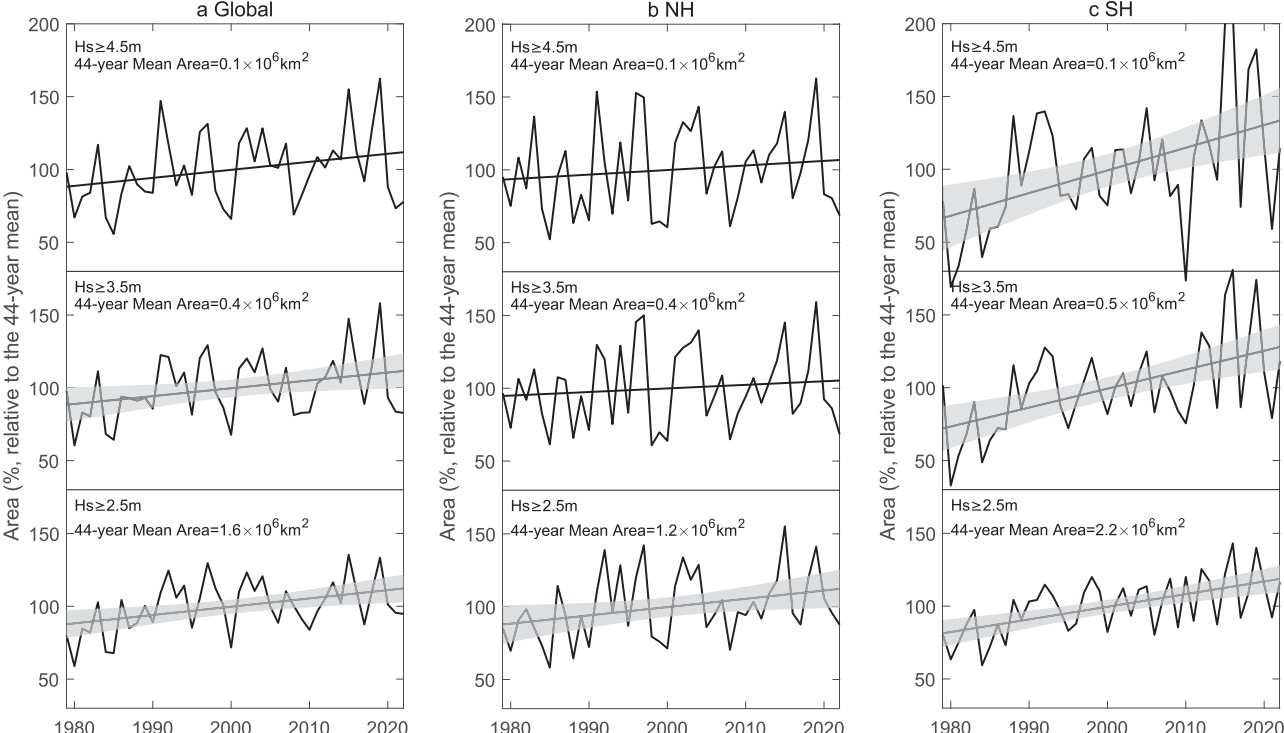

**Fig. 3 | Time series and linear trends of the area of tropical cyclone (TC) wave footprint. a** Global relative trend (straight line) and time series (solid line) of the annual mean of the wave area over 1979–2022. The area is specified for three different thresholds of TC waves (wave height above 2.5, 3.5 and 4.5 m). Relative values are relative to the 44-year mean. **b, c** As (**a**), but for the Northern Hemisphere (NH) and Southern Hemisphere (SH). Shading shows the 95% confidence interval for the significant trend; in (**b**) (top), the straight line with no shading indicates that the trend is not significant at the 95% confidence level, but significant at the 90% confidence level. Wave height is represented by significant wave height (Hs). Source data are provided as a Source Data file.

## Varying increase of TC wave energy

TCs play an important role in maintaining the energy balance at the air-sea interface[36,37]. TCs take heat energy from the ocean surface to fuel their development, and on the other hand, they also dissipate kinetic energy into the ocean by waves. The TC wave energy, defined as the TC energy accumulated annually in the ocean surface waves, indicates the amount of kinetic energy transferred from the atmosphere to the ocean by TCs. Precisely, the TC wave energy integrates the height and the area of the 6-h TC wave footprint over the annual accumulated TC duration (see Methods). In the following, the height, area and duration are denoted as three contributing components of the TC wave energy.

Figure 4a shows the trend of the globally integrated TC wave energy. Over the last 44 years, the energy has significantly increased by $8.9 \pm 7.3\%$/decade, relative to the long-term mean. Unlike the maximum height of TCWs, the upward trend in the global TC wave energy is dominated by swells, which account for 67% of the trend in the overall waves, while wind sea waves contribute only 33% (Supplementary Table 2). We decompose the energy into three linear terms of the three contributing components (the height, area and duration) by holding two components constant with time and by allowing the other one component to vary with time, and into four nonlinear terms by retaining two or three components varying with time simultaneously (see "Decompose the annual TC wave energy" in the Methods section). Here, we only report the results of the linear terms because the nonlinear terms have very little impact on the energy trends. We find that for the linear terms, the global increases of the area, duration (Fig. 4a), and wave height contribute to 72%, 35%, and −2.0% of the energy trend, respectively, and for the nonlinear terms, they overall contribute to 5% of the energy trend. However, the upward trends of the global duration and wave height are not significant at the 95% confidence level. Thus, the global energy trend is largely driven by the significant trend in the area.

The linear trend of the energy over the last 44 years varies greatly with the basin (Supplementary Figs. 3b, 4a and Supplementary Table 2). Significant upward trends are found in the NA ($29.3 \pm 14.4\%$/decade), EP ($28.0 \pm 21.7\%$/decade), NI ($16.9 \pm 18.2\%$/decade, significant in 90% confidence) and SI ($11.9 \pm 9.7\%$/decade). Small and insignificant upward trends are seen in the other two basins (WNP and SP). As shown in the global trend, swells dominate the basin-wide trends of the energy. Supplementary Fig. 4b shows relative contributions of the linear effects of the three contributing components to the basin-wide trends of the energy. Corresponding to the large variations in the basin-wide trends of the energy, the contributing components play very different roles in different basins. In the SP and WNP, the trends of TC wave energy are insignificant, but the causes are different. In the SP, related to the large interannual variability, none of the three components shows a significant trend, yielding no significant trend in the energy. In the WNP, however, the decrease of the annual duration, which is related to a reduction of TC frequency associated with the strengthening of the Pacific Walker Circulation[24], counteracts the effect of the increase of the area related to the strengthening of storm intensity. Consequently, in the WNP, the energy has no trend. In the NI, the energy trend is solely caused by the increase of the annual duration of storms, with small and insignificant changes in other two contributing components. In contrast, in the SI, the energy change is due to a significant increase in both the height and the area of TCWs. In the EP and NA, about 90% of the energy trend is due to the significant increase in the area and the duration, which agrees with previous studies on TC intensity[38] and frequency[39,40]. Figure 4b, c demonstrates the mixed picture in the energy trend in the WNP and NA due to different changes in the storm duration. Therefore, over the last 44 years, significant growth of the area is commonly seen in all basins, but the increase or decrease of other two contributing terms can superimpose or

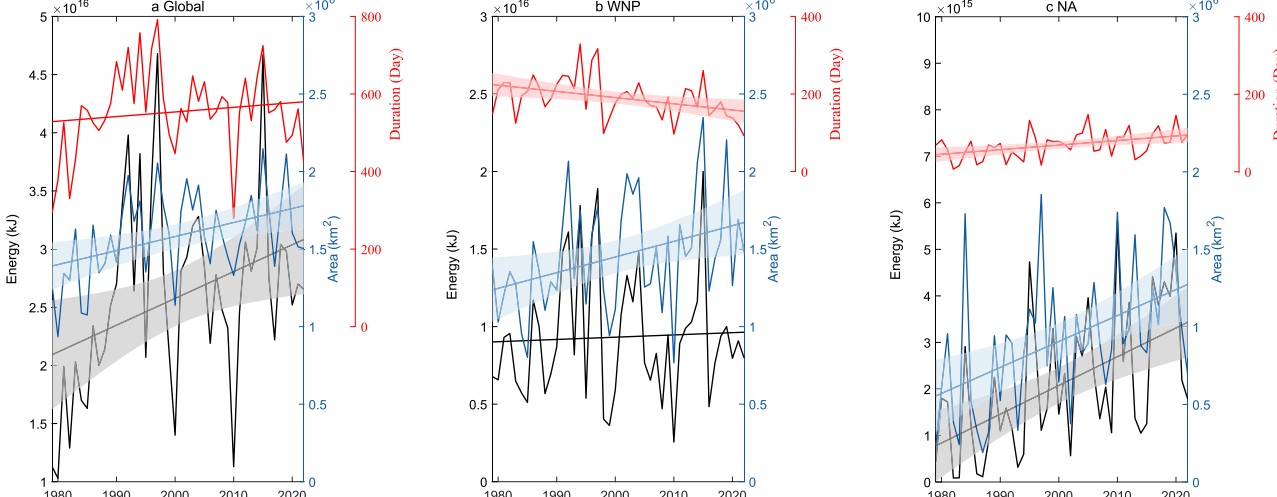

**Fig. 4 | Time series and linear trends of tropical cyclone (TC) wave energy, TC wave area, and TC wave duration. a** Global trend (straight line) and time series (solid line) of the TC wave energy (black; annually accumulated), the TC wave area (blue; annually averaged), and the TC wave duration (red; annually accumulated) over 1979–2022. **b, c** As (**a**), but for the western North Pacific (WNP) and the North Atlantic (NA). Shading shows the 95% confidence interval for the significant trend; in (**a, b**), straight lines without shading indicate that the trends are not significant at the 95% confidence level. Source data are provided as a Source Data file.

counteract this effect from the area, largely altering the final trend in the energy.

## Discussion

Here, we have quantified the global long-term historical changes in TC-associated surface ocean waves by analysing the TC wave footprints. This is based on a dataset that synthesises the latest ocean wave reanalysis and TC observations. We find that over the past 44 years, in the global average, the maximum height and the area of TCWs have increased by $3.2 \pm 1.3$%/decade and $5.7 \pm 3.8$%/decade (relative to the 44-year mean), respectively. Upward trends of the height and the area of TCWs are seen in all ocean basins, with the largest trends in the NA ($5.0 \pm 4.2$ and $18.3 \pm 10.2$ %/decade for the height and the area, respectively), although the trends in the NI and SP are not significant. The surface area exceeding a threshold of wave height is of importance when considering the risk of offshore and coastal structures encountering a damaging wave.

We also show that the global TC wave energy, indicating the TC energy transferred at the interface from the atmosphere to the ocean, has increased even faster by $8.9 \pm 7.3$%/decade. This global trend in the TC wave energy is about two times larger than that estimated for the global all-year wave energy[23], pointing to the important role of TCs in the global wave climate. The trend of the TC wave energy is mainly driven by changes in the area but varies greatly with basin, with the largest trends of about 30%/decade in the EP and NA. The compound effects of multiple TC properties, which contribute to the energy, can largely enhance or reduce the trends of the wave energy at the basin scale. We confirm that the above trends in TCWs are not dependent on the track source of ERA5 and the ENSO effect (subsection "Due to the ENSO effect" the in Methods section). But we notice that there is strong interannual variability in the wave metrics, and this can reduce the detectability of long-term trends over the period, especially at regional scales. Longer periods of wave data and climate simulations would help to reduce such uncertainty.

We emphasise that the upward trends of the TC wave metrics are much larger than the trend in the TC surface maximum winds ($2.3 \pm 0.8$%/decade globally) that drive the waves. An idealised atmosphere-wave coupled model study with ocean surface warming showed that TC wave height increases more rapidly than the surface winds[41]. The model also showed that the surface area of the TC wave footprint increases more rapidly than both the maximum wave height

and the surface winds. These nonlinear relationships are caused by the drag coefficient increasing with moderate wind speed. These predictions are supported by the trends of the height and the area of TCWs presented here. Furthermore, related to this point, we find that the maximum trends of the 6-h height within the footprint (6–7%/decade, Fig. 1c, f) are larger than the trends of 6-h maxima (3%/decade, Fig. 2b, c, Supplementary Table 1). Note that the wave fields in Fig. 1 have been rotated in the TC orientation. The maxima of the composite field average, based on fixed locations relative to storm centre, is smaller than the average of 6-h spatial maxima because the former may include smaller waves compared to the latter. This means that the 6-h spatial maxima is less sensitive to the surface winds than the storm wave maxima with fixed location.

Here, we also calculated the trends in the maximum height of TCWs during the storm lifetime, named the lifetime-maximum wave height (Supplementary Fig. 5 and Supplementary Table 1). We found that the relative trends of the lifetime-maximum wave height are smaller and less detectable than the relative trends of 6-h maximum height. The relative trends of the lifetime-maximum wave height are only significant in the globe, SH and SI, with rates of $1.7 \pm 1.4$%, $4.2 \pm 2.1$% and $5.4 \pm 2.6$%, respectively. This contrasts with the significant trends in 6-h maximum height in all basins. The weak and small trends in lifetime-maximum wave height resemble the less robust trends in the lifetime-maximum intensity. This may be also related to the saturation of the drag coefficient at the highest wind speed found in the modelling[41]. In short, our study sheds light on the rapid increase of overall ocean waves associated with TCs as a major hazard in the past 44 years. The relative changes in ocean waves presented here are much larger than the relative changes in TC surface maximum wind speed. This finding is consistent with previous observation-based studies in which the natural impacts of TC (such as total TC rainfall) over the satellite era have more robust trends than TC intensity likely related to uncertainty in observing storm intensity[42–44].

For future projections, a recent global study reported future decreases in the extreme wave heights driven by their modelled decrease in TC frequency[5], despite a strengthening of TC intensity. Another study also projected a future decrease in the extreme wave heights in the WNP associated with a decrease in TC frequency[45]. However, these studies should not be interpreted that the threat of the TC-associated waves will decrease in the future. Instead, we

anticipate that the risk of TCWs defined by the overall height and the areal footprint, which are independent of TC frequency, is likely to increase substantially in the future. Any total damage caused by TCWs is sensitive to the areal footprint. For future projections, the research community has more confidence in the effect of climate change on the TC intensity than on the other TC metrics (e.g., frequency and translation speed)[46,47]. More attention needs to be paid to the multiple drivers of future changes in the TC wave climate as a major peril.

## Methods

### TC tracks

TCs used in our study are firstly identified in the ECMWF fifth generation climate reanalysis (ERA5[48]) from 6-h atmospheric data, during 1979–2022, using the method described in refs. 49,50. The method tracks maximum vorticity centres in the spectrally filtered vorticity fields and matches the TC features based on observed tracks. First, the vertical average of the relative vorticity between 850 and 600 hPa is obtained. This is then spatially filtered using spherical harmonics to T63 resolution; the large-scale background with total wavenumbers $n \leq 5$ is removed. Vorticity maxima in the NH and vorticity minima in the SH are determined on the T63 grid and then used as starting points to obtain the off-grid locations using B-spline interpolation and maximisation methods[49]. In the first instance, all vorticity centres that exceed $0.5 \times 10^{-5}\,\text{s}^{-1}$ in the NH and that are below $-0.5 \times 10^{-5}\,\text{s}^{-1}$ in the SH are identified through the data time series. The tracking is performed by first initialising a set of tracks using a nearest neighbour method and then refining them by minimising a cost function for track smoothness subject to adaptive constraints on track smoothness and displacement distance in a time step. The detailed processes in tracking can be seen in refs. 48–50. After the tracking process, the maximum surface wind speed and its locations, in the unfiltered data are added to the ERA5 tracks. This is done by searching for the maximum winds at 10 m height within a 6° geodesic radius around the TC track point using the B-splines and minimisation method[51].

Because the identification of TCs in ERA5 is different from the TC identification in observations[52], a mismatch may occur between these two datasets, e.g., some observed TCs are not identified in ERA5 tracks or some ERA5 tracks are not in observations. To avoid the inconsistency, a matching procedure is applied to match the ERA5 tracks to the verification tracks from the International Best Track Archive for Climate Stewardship (IBTrACS)[52,53]. Details in the matching process can be seen in ref. 54,55. Briefly, an ERA5 track is matched to a verification track if the mean spatial separation is ≤5° over the corresponding paired track points and it is the track with the smallest separation. In the matching, we only include the tropical storms in IBTrACS with lifetime maximum intensity ≥17.5 m/s (i.e. several tropical storms). The matching process ensures that the ERA5 tracks are those TCs that are also observed in IBTrACS.

The following truncating process is finally applied. Each of the matched full TC tracks from ERA5 is truncated to the time length of the same track from IBTrACS, to ensure that the two datasets have the same lifetime for the same storms. We further restrict the 6-h track points within 40 °N and 40 °S. This criterion applied here is to minimise the uncertainty in analysing the TCWs. First, IBTrACS has the uncertainty in including extratropical cyclones and post-tropical cyclones (this then translates into the truncated ERA5 tracks)[55,56]. Second, swell waves in the middle-to-high latitudes driven by extratropical cyclones and strong westlies are usually in a large area and persist for a long time. This can largely contaminate the TC wave signal[57,58].

We use the ERA5 truncated tracks as the main TC dataset in the TC wave identification (see section "TC wave metrics" below). This is because the ERA5 tracks are dynamically associated with the ERA5 wave data through the data assimilation system and the wave model in ERA5. IBTrACS is not assimilated in ERA5. Although the ERA5 tracks are

matched with IBTrACS, the storm locations in the two datasets could slightly differ (difference is up to a 5° distance as described above). Thus, using ERA5 truncated tracks has less uncertainty in TC wave identification.

Despite the ERA5 data being extended back to 1940, in this study we focus on the post-satellite period (1979–2022). This is because of low confidence of IBTrACS data prior to 1979 that could be used in the matching and truncating processes.

### ERA5 wave reanalysis

We use the 6-h wave data (including the significant wave heigt of mixed (overall) waves, swell or wind sea waves) archived from ERA5, at an output resolution of 0.5 deg × 0.5 deg, from 1979 to 2022. The ocean wave model used in ERA5 is the WAve Modelling (WAM) model[59]. The horizontal grid size in WAM is 28 km. The wave spectrum is discretized in 36 directions and 36 frequencies. In ERA5, the ocean wave data assimilation is based on the optimal interpolation (OI) with a window length of 12 h. The wave height data from space-borne altimeters are assimilated in the wave fields from WAM. In the data assimilation, the two-dimensional spectra of significant wave heights are corrected by observations via the OI scheme, and the analysed field is used as the initial condition for the next step of model integration.

Here, we briefly describe wave spectra in the ERA5 wave data because they are in the centre of defining the TC wave energy in the subsection 3. In the ERA5 wave reanalysis, the spectral components of wind sea and swells are separated using the spectral partitioning techniques which account for the relationship between the local winds and wave direction[60]. In the partitioning, the spectral components are wind sea components when

$$1.2 \times 28 \frac{u_*}{c} \cos(\theta - \phi) > 1 \tag{1}$$

where $u_*$ is the friction velocity induced by winds (in the units of m/s), $\phi$ is the wind direction, and $c$ and $\theta$ are the phase speed (in the units of m/s) and the direction (in the units of degree) of wave components. For those components which do not match the above criterion, they are defined as swell components.

The significant wave heights (Hs, in the units of m) of either mixed (overall) waves, swell or wind sea waves are defined as four times the square root of the zeroth-order moment of wave spectrum energy, as follows

$$\text{Hs} = 4\sqrt{m_0} \tag{2}$$

where $m_O$ is the zeroth-order moment (in the units of m²), which is calculated from the integral of the two-dimensional wave spectral energy density function $F(f, \theta)$:

$$m_o = \int_{f=0}^{\infty} \int_{\theta=0}^{2\pi} F(f, \theta)\, df\, d\theta \tag{3}$$

where $f$ is the frequency (in the units of Hz) and $\theta$ represents the wave direction, for each spectrum component.

### TC wave metrics

The TC-associated surface ocean waves are identified by associating the TC tracks with wave data in ERA5. For each 6-h track point, a circle centred at the TC position with a geodesic radius of 15-degree is first drawn from the field of significant wave height (Hs) (Supplementary Fig. 6a), i.e., the first guess of TCWs. Within the 15-degree geodesic circle, the waves at the contiguous grid points where Hs ≥ 2.5 m are defined as the TCW footprint. The threshold of Hs = 2.5 m is chosen based on a recent evaluation of the size of ERA5 TCs[61]. This study found that compared to other measures of TC size, the outer size of TCs with

a surface wind speed of 9 m/s is best represented in ERA5[61]. This wind speed (9 m/s) produces waves with Hs = 2–3 m according to a wind-wave relationship[62] and a parametric wave model[63] under various TC conditions. The average value of Hs = 2.5 m is thus selected as the threshold in defining the TCW footprint. The 6-h TC wave footprint is further used to determine the statistical metrics for the area and the height of TCWs, and the TCW energy. Please note that we tested the sensitivity of the TCW footprint to the geodesic radius used in the first guess of TCWs. Supplementary Fig. 6b shows the distribution of the area of the 6-h TCW (i.g., where Hs ≥ 2.5 m) as a function of geodesic radius in the first guess. For small radius (i.e., 5 degree), the identified TCW footprint has a small area around the TC centre, and none of the footprint exceeds $10^6 km^2$. When the radius in the first guess becomes bigger, the sample size of small footprints reduces and sample size of large footprints increases. The distribution of the footprint area becomes stable when the radius is larger than 15-degree. Thus, we conclude that 15-degree of radius in the first guess is appropriate for the TCW footprint defined by Hs ≥ 2.5 m.

The 6-h area within the closed contour of the Hs value threshold is named as the TCW area ($A$, $km^2$), which can be expressed as follows:

$$A = \sum_{i=1}^{Ng} S(i) \text{ with Hs} \geq \text{Hs*} \tag{4}$$

where $N_g$ is the number of the contiguous grid points within the contour of the Hs threshold (Hs* = 2.5 m), and $i$ is the index of the grid cell. $S(i)$ is the area ($km^2$) associated with grid point $i$.

The maximum value of Hs within the 6-h TCW area is defined as the maximum height of TCWs. The maximum value of Hs among all the 6-h TCW areas during the TC lifetime is defined as the lifetime-maximum height of TCWs.

The TCW energy ($e$, $J/m^2$) for mixed waves (wind sea and swell waves), per unit of horizontal area, is calculated by integrating two-dimensional wave spectral energy in Eq. (3), expressed as:

$$e = \rho g \int_{f=0}^{\infty} \int_{\theta=0}^{2\pi} F(f, \theta)\, df\, d\theta = \rho g m_0 \tag{5}$$

where $\rho$ and $g$ are the density of seawater (1025 $kg/m^3$) and the gravitational acceleration (9.81 $m/s^2$). From Eqs. (2) and (3), e is rewritten as:

$$e = \frac{\rho g}{16} \text{Hs}^2 \tag{6}$$

The energies of swell and wind sea waves can be separated following ref. 57. This is defined as:

$$e_{\text{swell}} = \left( \frac{\text{Hs}_{\text{swell}}}{\text{Hs}} \right)^2 e \tag{7}$$

$$e_{\text{windsea}} = e - e_{\text{swell}} \tag{8}$$

where $\text{Hs}_{\text{swell}}$ is defined in Eq. (2) using the partitioned spectral components of swells. The wind sea component is calculated as the difference between the mixed wave energy and swell energy.

The annual accumulated TCW energy ($E$) for mixed waves is finally calculated by integrating the 6-h energy ($e$) of TCWs within the area ($A$) throughout the accumulated TCW duration in a year (i.e., combining Eq. (4) and Eq. (6) in a yearly base):

$$E(y) = \frac{\rho g}{16} \sum_{j=1}^{D(y)} \sum_{i=1}^{Ng(j,y)} \text{Hs}^2(i, j) S(i, j) \tag{9}$$

where $i$ is the index of the grid point in each 6-h TCW area ($A$), and $j$ is the index of the TCW duration in a given year $y$. $D(y)$ is the TCW duration (in the units of days), which is denoted by the duration of the 6-h TCW footprint in a given year. The TCW duration can be equally expressed as $D(y) = \sum_{k=1}^{TCN(y)} d(k)$, where TCN is the annual TC frequency, $d(k)$ is the length of individual TC lifetime. $S(i, j)$ represents the grid area ($km^2$) at grid index $i$ and duration index $j$.

For simplicity, in the TCW energy definition, the mean square of TCW height averaged over the area ($A$) is termed as the height term ($h$) of the energy, which can be expressed as:

$$h = \frac{\sum_{i=1}^{Ng} \text{Hs}(i)^2 S(i)}{A} \tag{10}$$

Substituting Eq. (10) into Eq. (9), the TCW energy $E(y)$ accumulated from the 6-h wave fields can be simply written as:

$$E(y) = \frac{\rho g}{16} \sum_{j=1}^{D(y)} h(j)A(j) \tag{11}$$

**Decompose the annual TC wave energy**

The annual TCW energy in Eq. (11) can be expressed by the yearly averaged height, area and duration:

$$E(y) = \frac{\rho g}{16} H(y)A(y)D(y) \tag{12}$$

where $H(y)$ is the annual average of the height term ($h$), obtained by averaging $h$ over the storm duration ($D$), as follows:

$$H(y) = \frac{\sum_{j=1}^{D(y)} h(j)A(j)}{A(y)D(y)} \tag{13}$$

Similarly, $A(y)$ is the annual average of TCW area:

$$A(y) = \frac{\sum_{j=1}^{D(y)} A(j)}{D(y)} \tag{14}$$

In each year $y$, $H(y)$, $A(y)$ and $D(y)$ can be denoted as the combination of their interannual departures ($H'(y)$, $A'(y)$ and $D'(y)$) and long-term means ($\overline{H}$, $\overline{A}$ and $\overline{D}$):

$$\begin{aligned} H(y) &= H'(y) + \overline{H} \\ A(y) &= A'(y) + \overline{A} \\ D(y) &= D'(y) + \overline{D} \end{aligned} \tag{15}$$

We then decompose the annual values of $E(y)$ into long-term time means and anomaly terms departing from the time means in a given year $y$, using the three contributing components of $H(y)$, $A(y)$ and $D(y)$ in Eq. (15). $E(y)$ in Eq. (12) is expressed as:

$$\begin{aligned} E(y) &= \frac{\rho g}{16} \left( \overline{H} + H'(y) \right) \left( \overline{A} + A'(y) \right) \left( \overline{D} + D'(y) \right) \\ &= \frac{\rho g}{16} \sum_{k=1}^{8} \delta_k \end{aligned} \tag{16}$$

$$\delta_1 = H'(y) * \overline{A} * \overline{D}$$
$$\delta_2 = \overline{H} * A'(y) * \overline{D}$$
$$\delta_3 = \overline{H} * \overline{A} * D'(y)$$
$$\delta_4 = \overline{H} * A'(y) * D'(y)$$
$$\delta_5 = H'(y) * \overline{A} * D'(y) \qquad (17)$$
$$\delta_6 = H'(y) * A'(y) * \overline{D}$$
$$\delta_7 = H'(y) * A'(y) * D'(y)$$
$$\delta_8 = \overline{H} * \overline{A} * \overline{D}$$

Now, annual $E(y)$ consists of long-term time means and anomaly terms. The long-term climatic average ($\delta_8$) is $\delta_8 = \overline{H} * \overline{A} * \overline{D}$, which represents the wave energy produced by the long-term means of height ($H$), area ($A$) and duration ($D$). This term is constant. The anomaly terms include first-order, second-order and third-order effects of the three contributing terms ($H$, $A$, $D$) on the annual wave energy. There are three first-order (linear) terms, denoted as $\delta_1$, $\delta_2$ and $\delta_3$, which explicitly quantify the deviation of energy solely due to interannual anomalies of height ($H$), area ($A$), and duration ($D$), respectively. There are three second-order terms $\delta_4$-$\delta_6$ and one third-order term $\delta_7$ in Eq. (17), which represent the nonlinear effect (covariability) of interannual anomalies of height ($H$), area ($A$), and duration ($D$). These high-order terms have small values and make neglectable contributions to the final energy trends, and they thus are excluded in our trend analysis.

## Validation of ERA5 TC waves

Here, we evaluate how well the ERA5 wave reanalysis represents the TC-associated ocean waves. We validated the ERA5 wave data against wave observations from in-situ buoys and satellite altimeters under TC conditions, and also compared against another wave reanalysis provided by the WAVe ReanalYSis (WAVERYS)[64].

Wave buoys from the National Data Buoy Centre are first used to validate the ERA5 wave reanalysis. These wave buoy data are independent of ERA5 because they are not assimilated in ERA5. Observations from 22 buoys during 1979–2018 are used in our validation (Supplementary Fig. 7a). These buoys are in the North Pacific and NA basins. Detailed information about location, valid period, and the number of validated TCs for each buoy is provided in Supplementary Table 3.

The validation process is as follows. First, at the time of a 6-h interval of ERA5 TC track, if the buoy is within the 15-degree geodesic circle of the track point, the Hs values from both the buoy and ERA5 wave data at the buoy location are extracted. For each validated TC, two time series of Hs, with one from buoy and the other from ERA5, are obtained (Supplementary Fig. 7b). Then, a TCW duration is calculated as the duration when the 6-h Hs exceeds 2.5 m. In principle, this process estimates the time when both the buoy data and ERA5 wave data are above the Hs threshold (2.5 m) at the same location during the same TC passage. Because the buoy data and ERA5 wave data are independent, the pair of TCW durations are calculated separately. During the observational period 1979–2018, there are 582 TCs satisfying the above criteria. Finally, we compare the TCW duration in the two wave datasets.

The comparison of the TCW duration between observations and ERA5 for the same TCs is shown in Supplementary Fig. 8. The observed TCW duration is well captured by the ERA5 wave data, with $r = 0.79$ for all buoys. The correlation is higher in the NA ($r = 0.84$), and slightly lower in the North Pacific basins where the sample sizes are small (less than 30% of total samples, Supplementary Table 3). The ERA5 wave data tend to underestimate the duration mean, with mean relative error (MRE, relative to the observations) = −8.3%. The Quantile-Quantile plots of TCW duration between observations and

ERA5, as another measure of validation, are shown in Supplementary Fig. 9. The ERA5 wave data well capture the TCW duration at most of percentiles although it tends to underestimate the values at highest percentiles in the WNP and EP where the sample sizes are small. Thus, ERA5 well represents the duration of the above defined TCW in observations.

The TCW duration is related to both the maximum Hs values and the storm translation speed. The scatter plots of the maximum Hs during the identified duration for the 582 TC cases from observations and ERA5 wave data are shown in Supplementary Fig. 10. The ERA5 wave data tend to slightly underestimate the observed maximum height of TCWs, but they agree well in general including the extreme values of Hs (>8 m). The correlation between observation and ERA5 data is $r = 0.77$, 0.68, 0.67 and 0.83 for the globe, the WNP, EP and NA, respectively. The mean relative error (MRE) is −10.7% and the RMSE is 1.0 m for the total 582 TCs. This result aligns with other conclusions that ERA5 has skill in capturing extreme wave heights observed by buoys and satellite altimeter[65,66].

For the 582 TCs which are seen in both wave buoy observations and ERA5 wave data, their TC translation speeds calculated from IBTrACS and ERA5 TC data are compared in Supplementary Fig. 11. For each TC, the translation speed is calculated by using the great-circle distance between two consecutive 6-h locations and averaged over the storm lifetime. The observed TC and ERA5 TC have a good agreement in the storm translation speed, with $r = 0.98$. The correlation does not vary with ocean basins. The MRE is about 1% and RMSE is less than 1.5 km/h. Thus, we conclude that the ERA5 well reproduces the observed TCW durations due to a good estimate in both the Hs values (≥2.5 m) and the TC translation speed.

Satellite remote sensing provides wave estimates at global scale, and it has been regarded as useful observational data in studying ocean waves[17,19,22]. Here, Hs observations from the Jason-2 satellite by a Ku-band altimeter[67] are also used to validate the TC wave data from ERA5. The validation period is from July 2008 to December 2018 when Jason-2 data are available. We use the satellite observations in the TCW first guess circle (15-degree geodesic circle) when ERA5 TCs are at their lifetime maximum intensity (Supplementary Fig. 12a). We take Jason-2 along-track Hs (about 3–4 km wide) within the TCW first guess area, and then bin them into the ERA5 grid (0.5 × 0.5 deg). A total of 1373 gridded values for 51 TCs are finally found over the validation period. Here, we only used the satellite data at the time of TC lifetime maximum intensity because waves are mostly largest when a storm reaches lifetime maximum intensity, and thus the proportion of TCWs (Hs ≥ 2.5 m) along the satellite track is larger than any other time. This reduces the proportions of small waves in the validation and makes the validation more desirable for TCWs.

The scatter plots of matched Hs values from Jason-2 and ERA5 are shown in Supplementary Fig. 12b–j for global and basin scales. Satellite altimetry can rarely observe the maximum value of Hs > 8 m (Supplementary Fig. 12b–j). This is related to limited satellite data recorded at fixed locations, compared to in-situ data, leading to large uncertainty of satellite altimetry in representing extreme waves[68]. We find that the ERA5 wave data can well capture the observed height of TCWs in Jason-2, including large Hs ≥ 6 m. The correlation between Jason-2 and ERA5 data is $r = 0.86$, 0.82 and 0.92 for the globe, the NH and SH, respectively. The MRE is −3.1% and the RMSE is 0.38 m, for the globe. The smallest errors are in the SI and SP basins. Jason-2 satellite altimetry is assimilated in ERA5 through its data assimilation system. However, our validation is still useful to prove the good quality of ERA5 for TCWs.

We also include WAVERYS[64] as an additional source of wave dataset to verify ERA5 wave data for TCWs. WAVERYS is a global wave reanalysis covering the period from 1993 onwards. The wave model used is the version 4 of the model Meteo France WAve Model. The altimeter wave data and directional wave spectra provided by Sentinel-

1 satellite are assimilated using the OI method with an assimilation window of 3 h. Compared to ERA5, WAVERYS produces the global wave data at a finer model resolution of about 0.2°. The wave-current interaction is considered in WAVERYS, but it is not included in ERA5. Source terms of white capping in the wave models of the two reanalyses are also different.

Using the same definitions of wave metrics (section 3 above), we compared the maximum height of 6-h TCWs between WAVERYS and ERA5 over the common period 1993–2018. The long-term means from WAVERYS are slightly larger than those from ERA5 both globally and in most basins (Supplementary Fig. 13), likely related to a finer model resolution of WAVERYS. At global and basin scales, the linear trends over 1993–2018 are very close between the two datasets (Supplementary Table 1). For example, the global trend of the maximum height of TCWs is 4.2 ± 2.5%/decade and 4.6 ± 2.5%/decade for ERA5 and WAVERYS, respectively. Upward trends of the maximum height of TCWs are found in all the basins when using WAVERYS, consistent with the results using ERA5. The small discrepancies in the TCW trends between the two datasets might be related to different model physics and resolutions. Nevertheless, we conclude that the two datasets produce consistent trends in TCWs over the period 1993–2018.

Lastly, we compare the ERA5 TCW footprint against a parametric wave model[63] that has been developed for TC conditions, for the area and maximum Hs. In the statistical wave model, the JONSWAP fetch-limited wind-wave growth relationship is used:

$$\frac{g\text{Hs}}{U_{10}^2} = 0.0016 \left( \frac{gF_L}{U_{10}^2} \right)^{0.5} \qquad (18)$$

Where $U_{10}$ is the 10 m wind speed, $F_L$ is the fetch length and $g$ is the gravity acceleration. Within the first guess area (15-degree geodesic circle), Hs is solved conditional on the wind speed $U_{10}$ and fetch length $F_L$ at each grid point. $U_{10}$ is obtained from the ERA5, and $F_L$ is a function of maximum wind speed ($V_{max}$) and TC translation speed ($V_{fm}$). Young (1988) defined an equivalent fetch $F_L$ for TC conditions:

$$\frac{F_L}{R'} = \psi \left( a_1 V_{max}^2 + a_2 V_{max} V_{fm} + a_3 V_{fm}^2 + a_4 V_{max} + a_5 V_{fm} + a_6 \right) \qquad (19)$$

where $a_1$-$a_6$ are coefficients pre-defined in ref. [69]. The term $\psi$ is a scaling factor[70], which is defined by

$$\psi = -0.015 V_{max} + 0.0431 V_{fm} + 1.30 \qquad (20)$$

The term $R'$ is defined by

$$R' = 22.5 \times 10^3 \log_{10} R - 70.8 \times 10^3 \qquad (21)$$

where $R$ is the radius of the maximum wind speed, which is obtained in the ERA5 TC track (see subsection "TC tracks" in this section). An instantaneous wave field produced by Eq. (18) is demonstrated in Supplementary Fig. 14a. Over the period 1979–2022, there are 809 TCs based on the selecting criteria in ref. [69] (i.e. 30 m/s < $V_{max}$ < 70 m/s, $V_{fm}$ < 15 m/s and 10 km < $R$ < 100 km).

We validate the lifetime average area, and the lifetime average maximum Hs within the footprint. By average, the TCW area is $1.79 \times 10^6$ and $1.72 \times 10^6$ km² for the ERA5 and the parametric model[63], respectively. The scatter plot of the TCW area averaged in the lifetime of individual TCs is shown in Supplementary Fig. 14b. ERA5 slightly overestimates the area with *MRE* of only 4% and the *RMSE* of $9 \times 10^5$ km². The scatter plot of the 6-h maximum height averaged over the storm lifetime in Eq. (18) and ERA5 is shown in Supplementary Fig. 14c. Compared with the parametric model, ERA5 tends to underestimate

the low-to-medium values of Hs (Hs < 6 m), and tends to overestimate the medium-to-high values of Hs (Hs > 6 m). But the overall error is small with *MRE* of 3% and *RMSE* of 0.9 m. This result is consistent with other findings that ERA5 tends to underestimate the extreme wave height[65,66].

Finally, we compared the composite of the TCW footprint in the two epochs (1979–2000 and 2001–2022) in the two datasets (Supplementary Figs. 15, 16). Only the common time when both datasets have valid values is considered. We find that the asymmetric distribution and the values of Hs within the footprint in ERA5 well resemble those in Eq. (18). An increase of Hs values and area between the two epochs is seen in both datasets. These agreements are better in the NH than in the SH, related to more samples of footprint in the NH. Compared to Eq. (18), the ERA5 wave data have less clear features in the storm centre, likely related to less representation of the eyewall contrast in the ERA5 wave model due to the low resolution. We note the difference of the composite values in Supplementary Figs. 15, 16 and Fig. 1. This is due to the different TCW sample selections when calculating Eq. (18), which is valid only for the time when TC is strong, i.e., 30 m/s < $V_{max}$ < 70 m/s. Thus, the TCW footprints produced by Eq. (18) are only used in validating the ERA5 wave data, and they are not ideal for trend analysis due to lower representability of TC conditions.

In all, we conclude that the ERA5 wave data are able to reproduce the duration, the area and the maximum Hs of TCW, including the Hs values within the footprint, compared to wave buoy, satellite altimetry, WAVERYS, and an independent parametric wave model. However, there is still a large uncertainty in the extremely high values of TCW (Hs > 8 m) due to very limited observations.

### Advantages and caveats of ERA5 wave data in representing TC waves

In this section, we summarise the advantages and caveats of the ERA5 wave data in analysing the long-term trends in TCWs. Advantages of ERA5 wave data include:

- Dynamical consistency between TC tracks and wave data. Because TC tracks and TCWs are consistently and objectively identified from the atmosphere-wave coupled reanalysis (see subsections 1 and 2 in the Methods section), they are well matched in terms of the timing and location. This is important for extreme analysis, including attributing wave trends to atmospheric forcing.
- Wave data assimilation. In ERA5, wave height data (Hs) from space-borne altimeters are assimilated in the wave fields from WAM twice a day with a 12-hour window (see subsection 2 above). The wave data assimilation corrects the initial wave fields used for the next step of model integration, and the atmosphere data assimilation corrects the atmospheric fields. This atmosphere-wave coupled analysis procedure improves both the quality of wave data and the coupling level of waves with the atmosphere.
- Good quality of TC wave metrics in the trend analysis. We defined the wave metrics based on the TC footprint which is tailored for ERA5 wave data. We further showed that ERA5 is adequate in systematically representing these TC wave metrics when compared to observations and other sources of wave data (see subsection 5).

Caveats of the ERA5 wave data, including how to minimise the impact, are:

- Uncertainty in extremely large waves. Global wave models have uncertainty in simulating extreme waves under TC conditions partially because the wave model resolution is not fine enough. This is also related to inaccurate wind forcing. Another source of uncertainty is the lack of extreme waves in

observations. Satellite altimetry can rarely observe Hs > 8 m (see "Comparison to satellite altimetry" in subsection 5 of the Methods section). Although the ERA5 wave data can capture some extreme events (Hs > 8 m) observed by wave buoys, the sample size is still limited (see "Comparison to wave buoy" in subsection 5 of the Methods section). Because validation for extremely large waves is challenging, it is still uncertain how well the ERA5 wave data can capture the most extreme events. To mitigate these uncertainties in the ERA5 TC wave data, we focus on the metrics of TC wave footprint that are designed to be less sensitive to extremely high values (see subsection 3). Briefly, the TC wave footprint is defined by moderate size of waves (Hs ≥ 2.5 m). Through a systematic validation, we show that ERA5 wave data well represent these metrics of TC wave footprint (i.e., the area and duration). Furthermore, we also analyse the relative changes in these TC wave metrics (including the maximum Hs). We anticipate that any identified relative trends in TCWs are more trustworthy than the absolute trends as the relative trends are less dependent on the mean values of extremely high waves.

- Interference of remote waves caused by other weather systems. The ERA5 wave data are driven by wind forcing from multiple types of weather, such as TCs, extratropical cyclones and large-scale persistent winds. It becomes challenging to distinguish TCWs from the waves driven by other weather systems at the middle-to-high latitudes. This becomes more difficult for swells as they can propagate in a long distance. To mitigate this challenge, we focus on the TCWs between 40 °N and 40 °S (see subsection 1) and leverage this with the TC wave footprint around the TC centre (see subsection 3).

- Potential underestimation of TCWs in coastal regions. A latest study[71] compared ERA5 and WAVERYS with in-situ measurements globally. They find that both wave reanalyses consistently underestimate extreme wave heights in most coastal locations. Caution must be taken for the feasibility of ERA5 wave data being used for coastal regions. However, we re-calculated the trends of the TCW area and maximum wave height after excluding TCW values in water depths less than 100 m, and the trend value remains essentially almost unchanged (Supplementary Tables 1, 2). For example, the global relative trend in maximum wave height changes from 3.2 ± 1.3%/decade to 3.3 ± 1.3%/decade, and the global relative trend in the area changes from 5.7 ± 3.8%/decade to 5.7 ± 3.9%/decade. Relatedly, recent studies showed that increasing horizontal resolution of wave models has little impact on TCWs in the open sea[72]. Thus, we anticipate that the large-scale (global and basin-wide) trends in this paper are dominated by waves in the deep ocean, and that the inaccuracy of ERA5 in representing coastal extreme waves has little impact on our conclusions.

All these suggest that the ERA5 wave data are validated in the trend analysis of TCWs.

## Uncertainty in the trend analysis

To consider the impact of the uncertainty of TC track identification on our results, we estimated the trends in the height of TCWs using different TC data. Supplementary Table 1 shows the trends of the TCWs identified using the full tracks of ERA5 TCs (i.e., matched to observed tracks but with the full lifetimes). The relative trends of the maximum height with the full tracks (2.1 ± 1.2, 2.3 ± 1.7 and 2.0 ± 1.8%/decade for the globe, NH and SH) are smaller than those with the observed tracks (3.2 ± 1.3%/decade), but the signs and significance levels for the global and hemispherical scales remain unchanged. Thus, the trends of the maximum height of TCWs are not strongly dependent on the TC track data.

The basin-wide trends in energy and their decompositions based on the full track of TCs were also estimated (Supplementary Fig. 4c and Table 2 (T2)). Although the absolute trends in energy become larger due to the longer lifetime of storms, the relative trends in energy are close to the above results. Relative contributions of the contributing components to the energy trends also mostly retain the same signs and significance levels when using the full track of TCs.

We evaluated the effect of ENSO on the trend detection of TCWs. To do so, the effect of ENSO is removed from the timeseries of TC and TCW metrics, as suggested in refs. 24,73. A multiple linear regression on the yearly ENSO index with a linear least-squares method is first estimated for the timeseries. Yearly ENSO index is the average of the monthly Niño 3.4 indices over a year. The Niño 3.4 index is defined by the standardised sea surface temperature anomalies in the Niño 3.4 region (5° S–5° N, 170° W–120° W)[74]. The regression values determined by ENSO index are then removed from the timeseries of TC and TCW metrics to obtain the timeseries residuals. These residuals are then used in the trend and correlation analyses. These processes are done for the timeseries of yearly values of TC and TCW metrics.

After removing the ENSO effect, we found slight changes in the trends, with the global trend value changing from 3.2 ± 1.3%/decade to 3.7 ± 1.0%/decade for the maximum height of TCWs, from 5.7 ± 3.8% to 6.9 ± 3.2%%/decade for the TCW area, from 8.9 ± 7.3%/decade to 10.6 ± 6.6%/decade for the TC wave energy, from 2.3 ± 0.8%/decade to 2.5 ± 0.6%/decade for the TC intensity, and from 2.1 ± 4.8%/decade to 2.9 ± 4.6%/decade for the TC wave duration (Supplementary Table 4). This means the ENSO only slightly impacts the trend detection in the TC metrics by altering the year-to-year variability, and it does not significantly change the trend values and significance level of the trends, either in global or basin scales.

## Statistical analyses

For the trend analysis, we estimate the trend (denoted as $b$) using a linear least-squares regression, and we also estimate the error bars (denoted as $err$) of the trend by a two-tailed 95% confidence interval under the assumption that the residuals of the regression follow a normal distribution. $b \pm err$ represents the 95% confidence estimate of the trend value. The trend is tested for statistical significance for a null hypothesis that the trend is zero (i.e., a significant trend means that the interval of trend ($b \pm err$) does not include zero). The Pearson correlation coefficient (denoted as $r$) is used to measure the correlation between the timeseries of two variables. A two-tailed $t$-test with a $p$ value of 0.05 is used to test significance, with a null hypothesis of a zero correlation. In our paper, for simplicity, we only provide the $b$ or $r$ values if they pass the significance test. In other words, the statistically significant values ($b$ or $r$) are at the 95% confidence level, unless stated otherwise.

## Data availability

The ERA5 climate reanalysis was generated and distributed by the European Centre for Medium-Range Weather Forecasts (ECMWF; https://www.ecmwf.int/en/forecasts/datasets/ reanalysis-datasets/ era5). The WAVERYS wave reanalysis was distributed by the Copernicus Marine Service (https://www.copernicus.eu/en/access-data/copernicus-services-catalogue/ global ocean-aves-reanalysis-waverys). The best track data were taken from the International Best Track Archive for Climate Stewardship (IBTrACS; https://www.ncdc.noaa.gov/ibtracs/). Monthly ENSO indices are retrieved from the NOAA's National Centres for Environmental Information (www.ncdc.noaa.gov/teleconnections/). The ocean wave buoy locations and observations were obtained from the National Data Buoy Centre (NDBC; https://www.ndbc.noaa.gov/). The ocean wave height observations from Jason-2 satellite were taken from National Centres for

Environmental Information (NCEI; https://www.ncei.noaa.gov/ data/oceans/jason2/). Source data are provided with this paper.

## Code availability
The code for tropical cyclone identification is available from https://gitlab.act.reading.ac.uk/track. The main scripts for data processing and plotting are available at zenodo (https://zenodo.org/records/10024056). Other source codes are available from Jian Shi (jian-shi@hhu.edu.cn) upon request.

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

## Acknowledgements

J.S., C.Z., A.T., W.Z. and J.Z. were supported by the National Key R&D Program of China (2023YFC3007900) and the National Natural Science Foundation of China (41930538, U2040203). X.F. and R.T. were supported by the Singapore Green Finance Centre. X.F. and K.H. were supported by the UK Met Office Weather and Climate Science for Service Partnership for Southeast Asia, as part of the Newton Fund. X.F. was also supported by "the Belt and Road Special Foundation of the State Key Laboratory of Hydrology-Water Resources and Hydraulic Engineering" (Nos. 2018490111), foundation of Key Laboratory of Ministry of Education for Coastal Disaster and Protection, Hohai University (J202203), and the UK National Centre for Atmospheric Science through the NERC National Capability International Programmes Award (NE/X006263/1).

## Author contributions

J.S., X.F., R.T. and J.Z. designed and performed the research; K.H. tracked the storm tracks in the climate reanalysis; C.Z., A.T. and W.Z. assisted with interpretation of the results. J.S., X.F., R.T., K.H. and J.Z. discussed the results and wrote the manuscript.

## Competing interests

The authors declare no competing interests.
