## [Peer Review File · Nature Communications]

Global increase in tropical cyclone ocean surface wavesREVIEWER COMMENTS

Reviewer #1 (Remarks to the Author):

This article analyzes long-term trends of tropical cyclone (TC) induced wind-waves based on the ERA5 wave results. The focus is timely given the need to better understand the links between climate change and coastal hazards in tropical cyclone areas, as surges and waves induced by tropical cyclones are a major natural hazard. However, I keep strong caveats about using ERA5 for TC induced waves. The article does not provide sufficient context or demonstration of how results from a global wave model can be used for TC applications, which seems to contradict previous wave climate analyses and results. There is also great uncertainty and lack of description in the methods, including the data, approach and modeling used to study TC induced wave activity.

In general, there is limited context given on (1) TC changes in each basin, which should be related to some of the results here presented, and (2) wave climatology in TC areas and limitations of global wave datasets to study these areas, including extreme wave heights.

On these grounds, I cannot support the publication of this article.

Other general comments include: more context and relevant references on TC and waves needed in the introduction; improve the description of the methods; better connect results with a summary of the method used to obtain them.

Detailed comments:

Line 47. The approach and focus on wave energy requires more elaboration. Reference 13 proposed a global metric for characterizing global changes in wave climate. However, existing global wave climate analyses present key limitations for TC areas. The authors may elaborate here that they propose to investigate whether a similar global change can be detected from tropical cyclone waves, using wave energy as proposed for wave climate in Reference 13, or will focus on other parameters too.

Line 62. New wave data set that synthesizes previous wave reanalysis? Or a new model that is specific for tropical cyclones only?

L62 to 76 – I don't think this paragraph needs to present the main results in this form.

L65 – is this the increase in wave height? Based on the intro, it seems one contribution of the article could be the focus on wave energy and ocean basins trends, instead of heights.

Also, note that previous works found that the largest changes in wave heights (e.g. references 8,9) and wave energy (reference 13) occur in high latitudes. More context should be provided in this regard.

L78. This line talks about extreme waves. The authors should clarify if the results refer to (1) regular wave heights, (2) wave energy and/or (3) extreme wave heights. Each parameter requires specific treatment.

L89. What storm position? The figure is not very clear, nor is it easy to understand what it represents, or how it is calculated. This requires much more elaboration.

Fig1. If the figure represents the 'average' or composite wave field for cyclones, the authors should describe if this is a result of wider TC or how this is explained. The article requires providing more

grounds on changes in TC, as wave fields have a more local genesis than wave parameters for extra-tropical storms.

L108. How are the trends and maximum heights calculated? This may require a specific extreme value analysis, but no indication is provided of how it has been determined.

L114-115. What other factors? This seems speculative.

Fig 2. What parameter? Significant wave height? Is this the maximum significant wave height over each TC season? The Results section requires far more explanation.

L135. Where are the changes in TC intensity described/provided?

L141. What is 'conservative threshold'?

L172. How are swells versus TC waves represented in the dataset?

L173. How is the energy decomposed into three principal components and why the three parameters provided represent each PC?

L337. Is ERA5 representing TC adequately compared to specific TC track datasets? The authors should elaborate why a TC specific analysis is not using TC tracks, wind field models, and simulation of wave fields in the analysis. See also L375, where it is stated that IBTrACS is not assimilated in ERA5.

L454. What is H6? Should not TCW have a σ^2 too? What is Hs respective to TCH height and Hs(l,j,y)?

L476. This is an interesting approach to study the different factors contributing to changes in E, but it is not sufficiently described.

Section in L495. How many of these waves are in TC areas. Note that global wave models do not capture TC induced activity due to (1) a lack of precision in the wind forcing and (2) lack of numerical resolution in the wave models. For example, see publication on global wave models and ensembles, such as: Morim et al A global ensemble of ocean wave climate statistics from contemporary wave reanalysis and hindcasts <https://www.nature.com/articles/s41597-022-01459-3>

See also previous analyses of TC in ERA5:

Dulac et al How Realistic are Tropical Cyclones in the ERA5 Reanalysis?

<https://meetingorganizer.copernicus.org/EGU22/EGU22-5755.html>

Bourdin et al Intercomparison of Four Tropical Cyclones Detection Algorithms on ERA5

<https://egusphere.copernicus.org/preprints/2022/egusphere-2022-179/egusphere-2022-179.pdf>

Hodges et al How Well Are Tropical Cyclones Represented in Reanalysis Datasets?

<https://journals.ametsoc.org/view/journals/clim/30/14/jcli-d-16-0557.1.xml>

Section in L517. The validation with satellite is only a validation of ERA5 wave parameters, which has been done before. The article should provide more references and context on this regard. It should also elaborate on the specific application for TC. See for ex: Cagigal et al Wind wave footprint of tropical cyclones from satellite data <https://rmets.onlinelibrary.wiley.com/doi/full/10.1002/joc.7764>, which states 'underestimation of the most extreme events is observed due to the relatively small number of observations recorded'.

I would also advise the authors to check this article for further context on wave climate in TC areas.

Reviewer #2 (Remarks to the Author):

The article provides a global assessment of tropical cyclone-derived ocean waves using the ERA5 reanalysis data from 1979 to 2018. The manuscript is well-written and clear. The methods followed are sound, and the conclusions are backed by the results. I believe the article can be accepted almost as is, but I do have an important concern and a few comments:

1. My main concern is the values reported for maximum significant wave heights. The maximum values shown are near 10 m, and we know there have been much higher waves measured under tropical cyclones. Please explain the values; why are there no waves above 10m? The validation vs altimeter data may not show higher waves because it depends on the time of the satellite measuring the waves, but there are buoy registers above the shown values; why are those not reflected? Are you really assessing the ERA5 based on tropical cyclone waves? Please try to be clearer in this issue, and please show the comparison between buoys and ERA5 values clearer. I am concern not only with the validation but if the ERA5 is still underestimating tropical cyclone waves, as previous reanalyses, then this is an important thing that needs to be mentioned.
2. The abstract and introduction mention that this is the first global assessment of tropical cyclone waves, but there is a published paper by Timmermans et al. (2017) that does present a global assessment (<https://doi.org/10.1002/2016GL071681>). While that paper is different as it used a global model and not reanalysis data, I think it should be mentioned.
3. Extended table 1 shows the same values for swell in the NH and the SH; please check that this is not an error.

Reviewer #3 (Remarks to the Author):

This manuscript reports trends of tropical cyclone (TC) wave activity on a global scale over the past four decades, based on state-of-the-art reanalysis data and additional information from best-track data, moored wave buoys and satellite altimetry. The authors find that while TC winds have increased by ~3%/decade, TC wave metrics exhibit higher, statistically significant trends for TC wave heights (4%/decade), surface area (7%/decade) and energy (11%/decade). These results are certainly novel and original, they fill a knowledge gap in the sense that such information was lacking for historical TC trends that have focused on atmospheric parameters (winds, rainfall, tracks, etc.) and was only available in the context of future projections from a handful of recent studies. The marine response to TCs, including both waves and storm surge, is now recognized as a major hazard for society, infrastructure and ecosystems, implying that the expect impact of this manuscript could be broad and multidisciplinary. However, the manuscript in its current form suffers from a number of flaws. The most important are that it is vague in numerous instances, using ambiguous writing and formalism that may be deceptive and do not allow reproducing the study; several additional analyses are also needed to better support (or even simply support) some of the authors' claims; some interpretations being made are questionable; some methodological aspects lack proper justification; addressing the numerous minor comments will also

require significant work. Detailed major and minor comments are provided below. For these reasons, I recommended publication in Nature Communications after major revision.

Detailed Comments

Major Comments

- Absolute vs. relative trend values. In general, I find that expressing trends as percentages is ambiguous and I prefer using dimensional units. It is understandable that the authors made this choice to be able to compare trends in different atmospheric and wave variables, both from the results they report and from the published literature (e.g. TC rain rates). However, in such case the reference value has to be specified clearly from the start, i.e. in the abstract, the introduction and wherever needed including the discussion&conclusion section. It is not the case here: as far as I know, the first instance is in the caption of Figure 1, i.e. only in the results section. Thus, those readers who will only browse quickly through the article (abstract, introduction, conclusion, perhaps the figures and not necessarily the full captions) may make a wrong interpretation. For those readers who read the paper more carefully like myself, they may be left with interrogations for several pages until the answer finally shows up, which is undesirable. For instance, an increase of 4% per decade may be interpreted as 4% of the baseline value (e.g. from 1979), but also as 4% of a running reference value that would change (increase) every decade, therefore with a nonlinear trend. This may sound trivial but dimensional units are unambiguous, while relative changes are particularly appropriate for integrated values over long period (e.g., the max height of TC waves has increased by 16% over the last four decades). They are perhaps more appropriate for the abstract, introduction and conclusion section, especially if the 1979 value is used (not the interannual yearly value but the value from the regression line). A clear definition of the reference value used in the reported relative trends would greatly improve clarity of the results for a diverse readership and the wider audience.

- 4th paragraph of the discussion section, L250-251: I am not fully convinced: the authors' interpretation is based on the assumption that lifetime max winds are increasing (just like the 6-hourly max winds), whereas based on ED Table 1, this is not true for the WNP, NI and SP (and less robust for the EP compared to the increase in the 6-hourly max winds), leading to non-significant NH trends. The basins where lifetime max wind trends are/are not robust tend to match those where lifetime TC wave height trends are/are not robust very well (only exceptions are NA and, to a lesser extent, EP). The authors should not ignore this result and rephrase their statement.

- Same paragraph, L253-254 (see also L28-30): This sentence is ambiguous for various reasons:

1. Increases in TC wave parameters are in the 4-11%/decade range. Ref. 28 reports a 12%/decade decrease in inner-core TC rainfall, which is a much larger value.
2. But maybe the authors actually referred to total TC rainfall including both the inner core and the outer region? Total TC rainfall has been shown to increase by 4%/decade according to Ref. 28 and by 13%/decade according to Ref. 29, which are values of the order of those reported here for ocean waves. The latter are thus not larger, unless the authors meant something else that they would need to clarify.
3. A word of caution here though: the trends reported here are for the past four decades, instead of the past two decades for TC rainfall. Have you tried computing the trends over the 1999-2018 period? This would be useful, both to allow better comparison with TC rainfall changes, and to assess if the TC wave increase may be changing over time.
4. Last, those readers unfamiliar with the works of refs. 28 & 29 may think that these lead to opposite

conclusions, judging solely from the paper titles. Yet, both studies have similar conclusions. For this reason also, it is absolutely necessary that the authors clarify this sentence and what part of the TC they refer to when mentioning changes in TC rainfall (i.e., not the inner core).

- Method reproducibility: several instances of vague descriptions that do not allow reproducing the analyses. e.g. L342-344 (unclear: please expand or at least provide a reference - see also L369-370); L346-349 (please provide more details and/or refer to published works describing the method(s)); L577-578 (Which index is being used, Nino3.4 ? What are the dates considered for the TC season?)

- Insufficiently supported choice of methods. e.g. L358-359 Please elaborate a little more: why are TC positions located between 40° and 60° latitude being excluded, after being considered in the first place (i.e. why not use the 40° threshold from the start)? Also, this methodology will not only exclude extratropical cyclones, but also post-TCs after their extratropical transition. Why has this choice been made? It may have been possible to separate between these and purely extratropical cyclones by also considering the initial position latitude. Related to L364: I don't understand: how is a comprehensive analysis including the post-TC stages possible if positions beyond 40° latitude are being excluded? Another example is found L522-523: Why? The analyses in this paper are mostly not restricted to the time of TC lifetime intensity.

- Equation (8): I find this equation a bit ambiguous, particularly the definition of TCW duration, in the sense that the recurrent case of simultaneous multiple storms is not explicitly mentioned. I find that it would be clearer to include an index over each TC track lifetime (i.e., 6-hourly time steps) and another one over each TC track in a given year. Perhaps the authors have rigorously considered all TCW footprints, even when several footprints occur at the same time, and found it practically (e.g. computationally) more convenient to frame the calculation this way. Is that so? Anyway, this has to be clearly specified in the text.

Related to this comment, ED Fig 6b illustrates the calculation of TCW duration for a single storm. This is very ambiguous and somewhat contradictory with the layout of Eq (8), which does not consider TC wave duration for individual storms, only that accumulated over a year. This point definitely requires clarification.

Regardless, another related concern is regarding the treatment/interpretation of duration and associated trends in the main text. Because it is accumulated over a year (as specified in the Fig. 4 caption), it depends on both individual storm duration and storm counts (frequency). Yet, this term is ambiguous as many readers are likely to interpret it as average storm duration, which is misleading. The authors need to address this issue by either rephrasing this term or by also including a more detailed analysis of TC frequency and average duration trends.

- Interpretation of the buoy comparison section as validation of TC wave area (L513-515): This claim is way too strong and I am afraid I must disagree. Even in the simplest case where we neglect TCW area variations in the cross-track direction and only focus on the along-track axis (reducing the spatial dimension to one), event duration at a particular point will depend not only on footprint spatial scale, but also on TC translation speed. To maintain this claim, the authors need to validate ERA5 TC translation speeds e.g. with IBTrACS data, using the 656 events (i.e. track portions) that relate to the duration analysis.

- Conclusion of the WAVERYS analysis regarding the robustness of ERA5 trends (L549-551): I am not convinced by the claim that trends are similar: for example, in the EP and NA (and in the NH), the upward trend is significant for ERA5 but not for WAVERYS; in the SP both trends are positive and non-significant but it is much larger in WAVERYS. In fact, since the purpose of this figure is to compare ERA5

and WAVERYS, only the common 1993-2018 period should be shown and considered for computations. How does the comparison change when the ERA5 period is reduced to after 1993? How do the results impact the paper conclusions and the overall robustness of the reported increasing trends of TCW activity? The authors should also provide the numbers here (trends in ED table) and discuss the results of this section in much more detail.

Minor Comments

- L35: Ref. 1 is about projections of extreme waves but has limitations for TC waves due to model resolution. Ref. 2 is about the role of coral reefs for coastal protection against ocean wave hazards, but is only marginally related to TC waves (at least, not explicitly in the paper). Ref. 3 is a case study for a particular TC but is still relevant here. I would suggest the authors also include their Ref. 30 here, because it overcomes the aforementioned limitations of Ref. 1 with high-resolution model forcing. Other references that the authors may wish to consider in this respect are Shimura et al. 2015 (methodologies used to derive TC- and non-TC wave fields are described in detail, unlike Ref. 30 which still uses the same methods), Timmermans et al. 2017 (earlier global-scale study with high-resolution forcing) and Belmadani et al. 2021 (high-resolution forcing for the North Atlantic). In addition, I suggest the authors also include a reference to Rappaport 2014 in order to support these introductory statements more directly.

Belmadani A, Dalphinnet A, Chauvin F et al (2021) Projected future changes in tropical cyclone-related wave climate in the North Atlantic. *Clim Dyn* 56:3687–3708. <https://doi.org/10.1007/s00382-021-05664-5>

Rappaport EN (2014) Fatalities in the United States from Atlantic tropical cyclones: new data and interpretation. *Bull Am Meteorol Soc* 95:341–346. <https://doi.org/10.1175/BAMS-D-12-00074.1>

Shimura T, Mori N, Mase H (2015) Future projections of extreme ocean wave climates and the relation to tropical cyclones: ensemble experiments of MRI-AGCM3.2H. *J Clim* 28:9838–9856

Timmermans B, Stone D, Wehner M, Krishnan H (2017) Impact of tropical cyclones on modeled extreme wind-wave climate. *Geophys Res Lett* 44:1393–1401. <https://doi.org/10.1002/2016G>

L0716 81

- L37: Rappaport 2014 also relevant here.

- L41: These references are all for oceanic WAVE extremes. Other oceanic extremes may include extreme sea levels, but also e.g. marine heatwaves. The authors should either change the terminology being used, or include additional relevant references (e.g. Vousdoukas et al. 2018, Frölicher et al. 2018).

Frölicher, T.L., Fischer, E.M. & Gruber, N. Marine heatwaves under global warming. *Nature* 560, 360–364 (2018). <https://doi.org/10.1038/s41586-018-0383-9>

Vousdoukas, M.I., Mentaschi, L., Voukouvalas, E. et al. Global probabilistic projections of extreme sea levels show intensification of coastal flood hazard. *Nat Commun* 9, 2360 (2018).

<https://doi.org/10.1038/s41467-018-04692-w>

- L46: Also consider including a reference to Timmermans et al. 2020 where the sensitivity of wave height trends to the chosen datasets is assessed using data from Ref. 12 and other recent datasets. Timmermans, B. W., Gommenginger, C. P., Dodet, G., & Bidlot, J.-R. (2020). Global wave height trends and variability from new multimission satellite altimeter products, reanalyses, and wave buoys. *Geophysical Research Letters*, 47, e2019GL086880. <https://doi.org/10.1029/2019GL086880>

- L48-49: long-term HISTORICAL changes. Studies of future projections are available (see above) but not of observed changes, indeed.

- L56: The term 'global warming' is a bit deceptive for Ref. 22, which is a process study of idealized TCs and associated wind & waves under varying background ocean temperature conditions, rather than a study of future projections. As far as I know, Ref. 24 does NOT deal with TC waves. Ref. 23 is about future projections of wave extremes including TC waves. Therefore, and because the next sentence is about historical changes, the authors should refer to 'the FUTURE impact of global warming on TC waves'.
- L58-59: Yes, but the authors could be a bit more specific, stating how these influence wind fetches and thus wave generation and propagation, preferably with some reference(s), e.g. Ref. 25. The authors may also specify the difference in TC wind sea and TC swell.
- L67: the uncertainty is quite large with err ~80% of b, which also seems to be the case for the other TC basins. This may be worth mentioning. See also L69, L71. Also, unit is cm for the absolute trend values.
- L69: Large uncertainty here as well => misleading use of 'even more quickly'
- L71-72: include the associated dimensional quantities.
- L96-97: Could you please discuss or at least mention the differences between the NH and SH values? Why are the latter clearly larger?
- L108: Extended Data (ED) Table 1 should be cited here (i.e. from the start).
- L108-109: Do not repeat the absolute trend values here, they are already in the table. It will help improve the reading flow. Also L133-135 for the relative trend values.
- L111: These relative trends. Also L147.
- L116: "nonlinear wind-wave interaction": Please elaborate more on this possibility. "decrease of TC translation speed": Could you comment on the non-significant (at the 95% level) negative trends shown on ED Fig. 1? Please discuss these results in the context of Refs. 17 & 18 but also Moon et al. 2019 and Lanzante 2019 who questioned the results of Ref. 18.
Lanzante, J.R. Uncertainties in tropical-cyclone translation speed. *Nature* 570, E6–E15 (2019).
<https://doi.org/10.1038/s41586-019-1223-2>
Moon, J.J., Kim, S.H. & Chan, J.C.L. Climate change and tropical cyclone trend. *Nature* 570, E3–E5 (2019).
<https://doi.org/10.1038/s41586-019-1222-3>
- L119: $p > 0.05$. What about the 0.1 threshold? It is used elsewhere in the paper for confidence intervals
- L120: only hinting at a small effect (if any) from the reduced translation speed
- Fig. 2: Why are the NH values larger than SH values here (also the case on ED Fig 4), whereas it is the opposite on Fig. 1?
- L131: The basin-wide trends in the maximum height of TC waves and surface maximum wind speed
- L132-133: Only significant at the 90% level for the weaker NI and SP trends.
- L135-137: Right, but the NA is the basin with the second largest change in TC intensity (3rd is SI and is notably lower), while the WNP is the basin with the second largest change in TC wave height (3rd again is SI). So yes, other factors affect the height trends, but TC intensity is apparently dominant.
- L140: for different thresholds of TC wave heights. Extended Data Table 2 should also be cited here to quantitatively supports several of the statements made in this paragraph.
- L142: become slightly larger
- L143-144: I assume this is true, but it is not evident from the figure which only shows that all three trends are significant at the 95% level. Or did you simply infer that from the relative increase in the error levels (compared to the trend values) ? Please clarify.
- L145: Remind the 2.6% value or refer to ED Table 1.
- L146: Please rephrase as "The area trends are much bigger in the SH than in the NH, especially for large waves"

- L146-147: "The SI is an important contributor to the SH signal": Why is that so? How many tracks in the SI vs SP?
- L147-148: sensitive to the height threshold (larger with larger thresholds, as with global averages)
- L148-149: So what? What does it suggest?
- L153-154: The 40-year mean value should be indicated on each subplot
- L155-156: The only one in this case is however significant at the 90% level, see ED Table 2
- L160-162: please start the sentence with "Among other processes,"
- L172: Here these values are relative to the global trend expressed in energy units (kJ/decade), whereas the paper systematically focuses on relative trends (in %), including in the previous sentence. This is confusing and the authors should fix that.
- L173: Better rephrase as main, major or dominant components. PCs have another meaning (see EOFs) and this term is therefore ambiguous.
- L174-176: I don't understand these values. From ED Table 2, TC wave area increases by 6.7%/decade and TC wave energy by 11.1%/decade (I don't know about duration, see my next comment). 6.7 is not the half of 11.1. Or maybe this relates to Eq.s (15-16) somehow, where trends in E and in Δ^2 , both expressed in kJ/decade, are compared, but then the reader will likely wish to see the corresponding values (see further comment regarding wave height trends).
- L176-177: Duration trend values should be included in ED Table 2 (or in a separate table) to better support this claim and e.g. indicate if the trend is significant at the 90% level, as for other trends.
- L177: I don't understand. A small negative value is provided here, whereas trends in TC wave heights have been previously shown as positive (Fig. 2a, ED Table 1). I understand that this analysis follows Equation 16 and probably refers to the term named Δ^1 , which is proportional to H' . How can the trend in H' be negative and that in TC wave heights be positive at the same time? This is very confusing.
- L182-183: I think it would make more sense to divide these values by four in order to have these estimates expressed in days
- L183: WNP and NA: It would be useful to include duration trend values for all basins on ED Table 2 (see previous comment regarding duration global trend)
- L187-188: Extended Data Fig. 3a may also be referred to here
- L189: NI: Trend does not appear as significant (at the 95% level) on ED Fig. 2b. In fact, it is only significant at the 90% level according to ED Table 2. Strikingly, the associated confidence interval crosses the zero-line: how can the trend then be significant considering the definition provided L595-597 (same for the swell component)? Or maybe the 90% CI is smaller than the trend value? The authors should check and elaborate on this (or correct in case of any errors).
- L195-196: Again, I am concerned with the significance of the energy trend in the NI basin.
- L198-199: I must be missing something: how come TC wave heights have a trend significant at the 90% level but not Δ^1 ? See related comments on the previous page.
- L199: Over 90%: $64+25=89 < 90$
However, adding up signed % contributions seems deceptive to me because negative percentages are possible, leading e.g. to $57+49=106 > 100$ in the NA.
- L200-201: Yes, but this decrease does not appear significant. In general, the results of this analysis for the WNP appear similar to those for the SP: they should be grouped together and this should be made explicit (even though there are slight differences such as that commented here by the authors).
- L215-216: Not entirely accurate: trends in TC wave area are insignificant in the NI and SP
- L239-241: Please rephrase as "the maximum trends of the 6-hourly height in the spatial fields, about 6-

8%/decade (Fig.1c,f), are larger than the trends of 6-hourly maxima, ~4%/decade". Fig. 2 alone is useless to support this sentence because trend values are not apparent. The authors should also quote ED Table 1 here. Also, it is Fig. 2b,c which should be quoted because Fig. 1c,f are for hemisphere-scale patterns and values.

- L241-244: These two sentences are very unclear. What spatial average? What do you mean by fixed locations? Etc. Please rephrase.
- L246: Please rephrase as "Based on the idealised model, Ref 22 also argued"
- L248: The authors should also mention that they calculated the trend in the lifetime max TC intensity since they mention 'increases of already very high winds' in the previous sentence.
- L250: Rephrase as "are usually smaller and less detectable". In the SH, relative trends are very similar and absolute trends are actually larger with lifetime-maxima, which seems to originate from the SI (ED Table 1).
- L258-259: I would not insist much on this study because of its limitations for TCs (see first minor comment).
- L271, 297, 316, 322, 641: Avoid using et al. here, there are only 6-8 co-authors.
- L338-339: Ref 33 is for ERA5 data, not for TC identification in such data, as the sentence suggests.
- L354: Please provide the following reference for IBTrACS data:
Knapp, K. R., Kruk, M. C., Levinson, D. H., Diamond, H. J., & Neumann, C. J. (2010). The International Best Track Archive for Climate Stewardship (IBTrACS), Bulletin of the American Meteorological Society, 91(3), 363-376.
- L357-358: i) State that this corresponds to the threshold for tropical storms
- L363: Actually, reference 35 was published before ERA5 was issued and only considers the older ERA-Interim. Can the authors include a more recent reference specific to ERA5 data? Or rephrase
- L367-368: I found no indication that MSLP has been used in the paper. Please remove. Also L369.
- L368-369: Did you mean the TC track position here? This makes for 6-hourly max wind time series for each track, from which one value of lifetime max wind speed may be extracted.
- L377-378: rephrase as "could be in slightly different locations."
- L383-384: Redundant with first sentence of previous subsection, please remove.
- L385-386: While this may be true to a certain extent, I think that the authors should be more careful with this statement: see Timmermans et al. 2020.
- L398: Wind waves usually refer to surface ocean gravity waves because they originate from a wind forcing, even if a part of the energy propagates as free waves (i.e. swell). The authors should rather call them wind sea, which is more standard. Here and across the manuscript.
- L401: Please clarify what you mean by mixed waves: is it the total wave field (adding up wind sea and swell components?)
- L423-424: please use a consistent notation, preferably A_6h (subindex). Also L452-454: H_6h
- L426: rephrase as "S(i) is the area (km²) associated with grid point i".
- L433: I cannot see how Ref. 33 may be relevant here. As far as I know, it does not present the method for computing wave energy. If the authors would just like to remind the data being processed, they should simply refer to ERA5.
- L466: Similarly to Eq(9). Mathematically, Eq(12) could also be derived from Eq (10) and Eq (11), but I believe this is not what the authors mean here.
- L485-486: Have you checked this thoroughly for all the TC basins?
- L503: How many for each buoy? For each of the 3 basins? See comment on ED Fig. 6a below.

- L508: smaller sample size. Yes but numbers are not being provided.
- L510: mostly in the small values of duration. This is practically impossible to tell from the figures (except for the WNP) given the sample sizes and use of the same color for data points and the x=y straight line. A Q-Q plot would be much better to support the authors' claim. See also L529: A Q-Q plot may also be useful here, although not as critical as for ED Fig. 7.
- 510-511: 'Probably' seems too strong unless the authors can rely on some published reference and/or other more solid evidence.
- L511-512: Again it is impossible to see from the plots. The 3-duration threshold is not even indicated.
- L524: ED Fig. 8 caption mentions 3-4 km
- L525: How many values & events for each basin (and hemisphere) ?
- L532: Vague. Do you refer here to the few points with severely underestimated Hs in ERA5, particularly in the NH? Did you explore these individual cases to check if they relate to e.g. underestimated wind speeds in ERA5?
- L548-549: No. They are very similar indeed, but not identical. In addition they differ more in the SP basin: any explanation for such discrepancy?
- L558: area of TCWs: this is not shown in the tables. I assume the authors meant energy rather than area.
- L558-559: This paragraph only deals with height. References to energy and ED Table 2 should be moved to the next paragraph.
- L562-564: Change to 'are not strongly dependent' as the agreement is only qualitative. Also, I noticed from ED Table 1 that this is not true on a basin scale: compare the trend significance levels for the WNP, NI and SP (the agreement is better for the other basins). This should not be eluded and the authors should therefore limit their claim to large spatial scales and/or discuss more precisely the discrepancies found for these basins.
- L570: rephrase as "also mostly retain the same signs"
- L583-586: What about the energy trend? Are the significance levels affected? Also, it would be interesting to see if this result holds for the 2 hemisphere and the 6 basins. In this respect, detailed results in table format, including also absolute trends would be useful.
- L600-601: I disagree with this statement, non-significant trend values are provided in the ED Tables. The authors should be clear that this statement does not apply to Extended Data.
- ED Fig. 2: How many TC tracks for each basin? How many more TCs in the NH compared to the SH? Can it influence the comparison between hemispheres (and basins) ? Please use the same y-axis ranges for all the basins to allow easier comparison. It would also help comparing with hemispheric averages from Figure 2.
- L676: 10m maximum wind speed.
- L680-681: Most of them are however significant at the 90% level, see ED Table 1
- L701-702: Not entirely sure of what this represents since values are higher than those shown on Fig. 2. I assume it may be the surface max wind speed occurring at the time of TC lifetime max wave heights, averaged annually, but the caption is not entirely clear.
- ED Fig 5: It would be useful to show the Hs field outside the 15° radius to allow the reader to fully judge the relevance of this distance in capturing the TC wave field. Further, in this respect, an individual case such as the one shown on this figure is an illustration but not a demonstration. Nothing is said as to how strong TC Jebi was in general and on 2018-09-01 12 UTC in particular. I suggest the authors compute the average Hs along the 15° circle for each position of each TC track and then show the histogram of these

average values. Then repeat for different radii. This should provide convincing evidence that the 15° radius is an appropriate choice.

- ED Fig6a: Additionally, the authors should provide more detailed information in an ED Table with the precise buoy coordinates, basin, IDs, time periods, % of missing data (at least during the TC seasons), sampling frequency, etc. Also include the number of TC events being considered for each buoy.

- ED Fig 6b: Elsewhere in the manuscript TCW duration relates to a TCW footprint over one or several grid points. Here we are at a fixed location, which will tend to underestimate TCW duration since it may be found outside the 2.5-m contour while such contour may still be found elsewhere within the 15°-radius. The authors need to discuss that. Also, the use of shading combined with contour lines connecting the dots is deceptive because it implicitly suggests that the values are linearly interpolated to compute start and end dates/times more accurately. Histograms would be more appropriate in this context (staircase-like graphs). Finally, if this is a demonstration, then you should provide the results for this particular example: is it 6 for ERA5 and 9 for the buoy? I am not sure this is a very good example because the start and end points lie very close to the 2.5m-threshold (except for the ERA5 end point), making it difficult for the reader to see whether these points are included or not.

- L720-721: rephrase as "The number of 6-hourly time steps with TC wave $H_s \geq 2.5\text{m}$ (shaded area) represents the TC wave duration"

- ED Fig 8: In panels b-d please use the same tick intervals for the y and x axes (every 2 m everywhere). It would also be useful to break the information into the different basins

- L733: rephrase as "(footprint about 3-4 km wide, expanded here for visual clarity)"

- L734-735: I don't understand what is shown on ED Fig8a. Obviously NH and SH TCs are seasonally out of phase so the 6 TCW footprints cannot occur at the same time. Please provide dates, times and names of the 6 TCs shown in this example.

- ED Table 1: Some lines in the table are not being used in the manuscript and should therefore be removed.

We have worked to address all of the comments carefully in this revised version. Those improvements are highlighted in the manuscript. Please see below, in blue, for a point-by-point response to the reviewers' comments. All page numbers refer to the revised manuscript with tracked changes.

Best regards,

Jinhai Zheng

Corresponding author, on behalf of all authors (jhzheng@hhu.edu.cn)

Responds to the reviewers' comments:

Reviewer #1:

This article analyzes long-term trends of tropical cyclone (TC) induced wind-waves based on the ERA5 wave results. The focus is timely given the need to better understand the links between climate change and coastal hazards in tropical cyclone areas, as surges and waves induced by tropical cyclones are a major natural hazard. However, I keep strong caveats about using ERA5 for TC induced waves. The article does not provide sufficient context or demonstration of how results from a global wave model can be used for TC applications, which seems to contradict previous wave climate analyses and results. There is also great uncertainty and lack of description in the methods, including the data, approach and modeling used to study TC induced wave activity.

In general, there is limited context given on (1) TC changes in each basin, which should be related to some of the results here presented, and (2) wave climatology in TC areas and limitations of global wave datasets to study these areas, including extreme wave heights.

Response: We thank the reviewer for these comments which are very useful to improve the quality of our paper.

- With respect to 'TC changes in each basin, which should be related to some of the results here presented', we improved the evaluations on the changes of TC intensity, duration and translation speed at globally and basin-wide scales, and explicitly linked these changes to TC wave changes (please see Lines 163-170,184-192 and 189-192, Figures 2-4 and, Supplementary Figs. 2-5 and Supplementary Tables 1-2). At globally and basin-wide scales, we also decomposed TC wave energy to three contributing terms (wave height, area and duration), which are related to TC intensity, frequency and lifetime. Please see Supplementary Fig. 4 and Supplementary Tables 1-2. We included clarifications for these in Lines 277-296.
- With respect to 'wave climatology in TC areas', we also added a new figure and new section to describe climatology of TC wave areas (Supplementary Information Lines 30-53, Supplementary Fig. 1). We also described climatology of TC wave height and energy in this section.
- With respect to 'limitations of global wave datasets to study these areas', We added the Methods subsection 6 to summarize advantages and caveats of ERA5 wave data(Lines 721-762) . We also provided an evaluation on extreme wave heights under TC conditions, which is presented in Supplementary Fig. 10 and Lines 635-642.
- In the Methods subsections 7, we further discussed the uncertainty in the trend analysis. Please see Lines 764-808.

- Finally, the whole Methods section has been improved in the revised version of the manuscript, including adding more clarifications in the Methods subsections 3&5 about TC wave metrics and validation. Please see Lines 485-548 and 586-719.

We hope these substantial improvements will address the reviewer's concern on the feasibility of using the ERA5 wave reanalysis in TC wave trends.

On these grounds, I cannot support the publication of this article.

Other general comments include: more context and relevant references on TC and waves needed in the introduction; improve the description of the methods; better connect results with a summary of the method used to obtain them.

Response: We thank the reviewer for these comments. In this revised version of the manuscript, the reviewer's comments have been carefully considered and addressed as follows.

- Briefly, with respect to 'more context and relevant references on TC and waves needed in the introduction', we have extended the description on ocean wave characteristics and added 10 more references related to TC and TC waves in the introduction part (Lines 36-71).
- With respect to 'improve the description of the methods', we have expanded the Methods section by providing more details on how those methods are developed and constrained. This includes a better description on the TC tracking and matching processes (Lines 382-445), a better description on the TC wave metrics (Lines 485-548), and more detailed validation results against observations (Lines 586-719);
- With respect to 'better connect results with a summary of the method used to obtain them', we have added a brief description on the TC wave metrics when they are first mentioned in the main text, and then explicitly referenced this to the method part (Lines 77, 86-87, and 232-233).

Detailed comments:

Line 47. The approach and focus on wave energy requires more elaboration. Reference 13 proposed a global metric for characterizing global changes in wave climate. However, existing global wave climate analyses present key limitations for TC areas. The authors may elaborate here that they propose to investigate whether a similar global change can be detected from tropical cyclone waves, using wave energy as proposed for wave climate in Reference 13, or will focus on other parameters too.

Response: Thanks for this useful comment. We agree that the global wave metrics developed for the general waves may not be ideal for TC waves. Based on this comment, we re-defined the research objectives in the revised version of the manuscript in Lines 51-59.

Line 62. New wave data set that synthesizes previous wave reanalysis? Or a new model that is specific for tropical cyclones only?

Response: Thanks for this comment. We clarified this in Lines 73-75 in the revised version.

L62 to 76 – I don't think this paragraph needs to present the main results in this form.

Response: Thanks for pointing this out. We understand the reviewer's concern. However, this is an editorial requirement for Nature Communications articles (<https://www.nature.com/ncomms/submit/article>). From previous submissions, it says “*The results of this study should only be discussed in the final paragraph of the Introduction. This paragraph should provide a short summary of the main findings of your study. Technical details that are necessary to follow the main text should be provided at the beginning of the Results section, while a more detailed discussion should be placed in the Methods section at the end of the*

article. Please focus more on what you find, not what you do in the introduction.”

L65 – is this the increase in wave height? Based on the intro, it seems one contribution of the article could be the focus on wave energy and ocean basins trends, instead of heights.

Response: Thank you for pointing this out. Yes, this is the *relative* trend in wave height. In this paper, we looked at various metrics for TC waves (e.g., height, area, duration, and energy) both globally and at basin scale. We highlight the value of this study in the relative trends of TC waves, which depend less on how well the ERA5 simulates the extreme wave height. We clarified this comment in Lines 75-77.

Also, note that previous works found that the largest changes in wave heights (e.g. references 8,9) and wave energy (reference 13) occur in high latitudes. More context should be provided in this regard.

Response: Thank you for pointing this out. We added a sentence to highlight this point (Lines 51-55), which mirrors the challenges and lack of studies for TC waves.

L78. This line talks about extreme waves. The authors should clarify if the results refer to (1) regular wave heights, (2) wave energy and/or (3) extreme wave heights. Each parameter requires specific treatment.

Response: Thank you for pointing this out. We have removed this sentence to avoid confusion. In the paper, we later introduced which wave variables are analyzed (Lines 75-82).

L89. What storm position? The figure is not very clear, nor is it easy to understand what it represents, or how it is calculated. This requires much more elaboration.

Response: Thank you for pointing this out. The storm position means the TC track location. We improved this sentence in Lines 111-112, and we also added a clarification on how this figure is calculated in Lines 113-114. More details on the Fig. 1 caption (Lines 129-141).

Fig1. If the figure represents the ‘average’ or composite wave field for cyclones, the authors should describe if this is a result of wider TC or how this is explained. The article requires providing more grounds on changes in TC, as wave fields have a more local genesis than wave parameters for extra-tropical storms.

Response: Thank you for pointing this out.

- In this paper, we define the TC waves as the waves with $H_s \geq 2.5\text{m}$ around the TC track position (see the Methods Lines 487-492). Fig. 1 thus only plots the average H_s values in (a,b,d,e) where $H_s \geq 2.5\text{m}$, and the trend in (c,f) where the 40-yr mean $H_s \geq 2.5\text{m}$. $H_s = 2.5\text{m}$ is chosen as the threshold because the ERA5 has the best estimation of outer size of TCs with surface wind speed of $\sim 9\text{m/s}$, which is equivalent to $H_s = 2.5$ by the classic wind-wave relationship (more details in the Method section. Please see Lines 492-496). We added a description in Fig. 1 caption to address this point (see Lines 129-141).
- The trend in TC intensity is shown in Fig. 2 (green Lines), and relevant description is in Lines 154-157. We also estimated the trends in other TC metrics, such as translation speed and 6-hourly maximum wind speed (Supplementary Fig. 1 and Supplementary Table 1). We found the trends in TCs can only explain part of the trends in TC waves. We further linked the inconsistent trends in TCs and TC waves with the nonlinear interactions of strong winds and high waves (Lines 330-334). All these point to the complicity of TC waves (as we firstly expected in the intro).
- In this paper, we only focus on the waves related to TCs (satisfying the warm core criterion) within 40°N and 40°S , to avoid the influence of extratropical cyclones. This has been clarified in Lines 423-434.

L108. How are the trends and maximum heights calculated? This may require a specific extreme value analysis, but no indication is provided of how it has been determined.

Response: Thank you for pointing this out. We have added a brief clarification on the definition of the maximum height of TC waves (Lines 148-149 and Lines 179-181 in Fig. 2 caption). Due to the limited space, in the main text, we cannot provide details on the definition and the process of computing linear trends. Instead, they are all provided in the Methods part (subsection 3&7) and explicitly referred to the Methods subsections.

L114-115. What other factors? This seems speculative.

Response: Thanks. The other factors are mentioned in L159-160. We then discussed and tested the contributions of these factors (nonlinear wind-wave interaction and translation speed) in L330-333 and Lines 161-170.

Fig 2. What parameter? Significant wave height? Is this the maximum significant wave height over each TC season? The Results section requires far more explanation.

Response: We are sorry for the confusion. This parameter is the annual mean of 6-hourly maximum height of TC waves, and the maximum height is defined by the maximum value of H_s within the 6-hourly TC wave footprint. This has been clarified in Fig. 2 caption. Please see Lines 179-181.

L135. Where are the changes in TC intensity described/provided?

Response: We are sorry for the confusion. We were supposed to refer this paragraph to Supplementary Fig. 3 and Supplementary Table 1. We have corrected this in the revision. Please see Line 184.

L141. What is 'conservative threshold'?

Response: We are sorry for the confusion. We have rephrased this sentence (Line 197) to make this point clearer.

L172. How are swells versus TC waves represented in the dataset?

Response: In ERA5 wave model, the two-dimensional wave spectrum is divided to swell and wind sea wave components using spectral partitioning techniques accounting for the local wind speed, wind direction and wave direction. For each component, H_s is estimated by four times the square root of the wave spectrum energy. We have clarified briefly the main text (Lines 146-147) and added a detailed definition in the Method section (Lines 465-474).

L173. How is the energy decomposed into three principal components and why the three parameters provided represent each PC?

Response: Thanks for pointing this out. In the revised version, we had a description on these three terms before showing the results (Lines 230-234). To make this point more explicit, we improved the sentence in Lines 241-245. Detailed descriptions are provided in the Methods section (Lines 550-584). We hope this will address the reviewer's concern.

L337. Is ERA5 representing TC adequately compared to specific TC track datasets? The authors should elaborate why a TC specific analysis is not using TC tracks, wind field models, and simulation of wave fields in the analysis. See also L375, where it is stated that IBTrACS is not assimilated in ERA5.

Response: Thanks for pointing this out. There are several reasons why we think the TC tracks and

associated wave data in ERA5 are adequate in TC wave trend analysis. :

- First, for TC track itself, previous studies showed that TC tracks in climate reanalysis can well capture the observed tracks (Best Track) after applying appropriate criteria in vortex tracking (Hodges et al., 2017) . Second, as we mentioned in the Methods section (Lines 406-418), in this study the ERA5 tracks are matched to IBTrACS to make sure that those identified tracks in ERA5 are equally observed in IBTrACS. Furthermore, in the revised manuscript, we compared TC translation speed between the ERA5 tracks and IBTrACS, and they are found to have a good agreement (Lines 644-655). Thus, we are confident that the ERA5 TCs are adequate to represent observed tracks.
- We didn't use IBTrACS in interpreting the trends found in ERA5 TC wave data, because the ERA5 TCs have small mismatch with the wave data (although the mismatch is small due to the matching process mentioned in above) and also because IBTrACS have a lot of missing data in storm intensity. Thus, using IBTrACS in an ERA5 wave analysis will introduce more uncertainty than using the ERA5 tracks. We have included clarifications on this point in the Methods Lines 430-434.
- Producing a separate long-term and high-resolution TC wave dataset by forcing a wave model with statistically estimated TC wind fields is expensive. More importantly, from our experience, this will include multiple sources of uncertainty, which could significantly degrade the feasibility of such data used in a trend analysis. The first caveat is the lack of intensity observations in some tracks, including in the early and late life stages of storms, which reduces the sample size of TC wave data (both TC number and duration will be reduced). Second, the statistical model used in reconstructing TC wind fields has large uncertainty in the outer size and wind speed, related to lifecycle, regions, and impacts of large-scale background conditions. This will have a significant impact on the trends of TC wave area and energy. Third, there are no wave field corrections/constraints during the free model run. The model output will be drifted and biased towards its own wind-wave equilibrium between the source and sink terms.
- Instead, the benefits of ERA5 wave data include consistency between the timing and location of TC tracks and wave data, wave data assimilation, and good quality for analysed TC wave metrics in trend analysis. We have added a separate subsection (the Methods subsection 6, Lines 721-760) in the new version of the manuscript to explicitly clarify this point.
- In the abstract and introduction of the revised manuscript, to avoid any overestimation of the results, we have explicitly specified that the analysis results are genuinely based on the 6-hourly climate reanalysis ERA5.

We think all these improvements will address the reviewer's concern on the usage of ERA5 data.

L454. What is H6? Should not TCW have a \hat{h}^2 too? What is Hs respective to TCH height and Hs(L,j,y)?

Response: We are sorry for the confusion. We have modified these equations for TC wave metrics. In the new version of the manuscript, we used h to represent the mean square of TCW height over the 6-hourly TCW area only in the energy definition, and H to represent the annual mean of the height term (h)(Lines 544). We replaced Hs(i,j,y) with Hs(i,j) in Eq(9), which is the Hs in the grid i and TCW duration index j (Line 533).

L476. This is an interesting approach to study the different factors contributing to changes in E, but it is not sufficiently described.

Response: In the new version of the manuscript, detailed descriptions on this approach are provided in the Methods section (Lines 550-584). we have added a brief description on this approach in the main text before showing the results (Lines 241-245). We hope these improvements will address this comment.

Section in L495. How many of these waves are in TC areas. Note that global wave models do not capture TC induced activity due to (1) a lack of precision in the wind forcing and (2) lack of

numerical resolution in the wave models. For example, see publication on global wave models and ensembles, such as:

Morim et al A global ensemble of ocean wave climate statistics from contemporary wave reanalysis and hindcasts <https://www.nature.com/articles/s41597-022-01459-3>

See also previous analyses of TC in ERA5:

Dulac et al How Realistic are Tropical Cyclones in the ERA5 Reanalysis?

<https://meetingorganizer.copernicus.org/EGU22/EGU22-5755.html>

Bourdin et al Intercomparison of Four Tropical Cyclones Detection Algorithms on

ERA5 <https://egusphere.copernicus.org/preprints/2022/egusphere-2022-179/egusphere-2022-179.pdf>

Hodges et al How Well Are Tropical Cyclones Represented in Reanalysis

Datasets? <https://journals.ametsoc.org/view/journals/clim/30/14/jcli-d-16-0557.1.xml>

Response: Thank you for pointing this out. We have compared ERA5 TC wave data with moored wave buoys and satellite altimetry data (Supplementary Figs. 7-12). The results showed that ERA5 wave data have a good quality in representing TC wave footprint (i.e., TC wave area and TC wave duration), which is the main focus in this paper. Even for the height, in the new version of the manuscript, we showed that TC Hs is in a good agreement between ERA5 and Jason-2 satellite altimeter data (Lines 681-692 and Supplementary Fig. 12), and between ERA5 and buoy observations (Lines 635-642 and Supplementary Fig. 10). In this paper, we looked at various wave metrics for TC conditions (height, area, and energy) both globally and at basin scale. We note that we focus on the relative trends of these variables of the TC wave footprint, which depends less on how well the ERA5 simulates the extreme wave height. Please see the new added subsection 6 in the Methods (Lines 721-760) for a summary of the above points.

Section in L517. The validation with satellite is only a validation of ERA5 wave parameters, which has been done before. The article should provide more references and context on this regard. It should also elaborate on the specific application for TC. See for ex: Cagigal et al Wind wave footprint of tropical cyclones from satellite data <https://rmets.onlinelibrary.wiley.com/doi/full/10.1002/joc.7764>, which states ‘underestimation of the most extreme events is observed due to the relatively small number of observations recorded’.

Response: Thanks for this useful comment. To address this,

- We have added more references relevant to the validation of ERA5 waves. We also clearly stated that our validation was focused on TC wave data, and improved the description on the validation methods to compare wave data between satellite data and ERA5 data. Please see Lines 661-692.
- Thanks for pointing out this very useful reference (Cagigal et al., 2022). For extreme waves ‘the underestimation ... probably because the number of data to populate those bins associated with extreme events is not large enough’. This means that the wave retrieval algorithm used in satellite data has large uncertainty in estimating the largest waves. We have added a clarification on the uncertainty of satellite data in representing extreme waves (Lines 682-685).
- We agree with the reviewer that the uncertainty of ERA5 wave data for the largest waves is high, which is the case for observations as well (e.g., buoys could be not working properly or damaged, and sea state retrieval algorithms for satellite altimeters have low confidence in estimating waves, under TC conditions). We recall that our study focuses on the wave metrics tailored for ERA5 (e.g., area, duration) and their relative trends, which are less sensitive to the largest absolute wave heights. We added a clarification in the main text to emphasize this important point, which should have been included in the first version of the manuscript. Please see Lines 75-80.
- To more explicitly address the reviewer’s concern on the feasibility of ERA5 wave data for TC wave analysis, we added a separate subsection in the Methods to summarize the advantages and caveats of ERA5 data, including how to minimize the effect of these caveats on our analysis. Please see Lines 721-760.

I would also advice the authors to check this article for further context on wave climate in TC

areas.

Response: Thanks for your suggestion. We have carefully read the reference and found there are two main differences between Cagigal et al., (2022) and our study.

- Cagigal et al., (2022) proposed a new model to estimate the wind wave footprint by using satellite data. The model results were validated against buoy data to confirm its usefulness. They found that for extreme wave conditions, the model cannot well estimate the footprint. The reason is due to the relatively small number of observations recorded. In our study, we used a wave reanalysis dataset to determine the TC wave footprint, which has consistent data quality for all the TC events. The validation of ERA5 waves with buoy and satellite data showed that ERA5 wave data have a good quality in representing TC wave footprint (Supplementary Figs. 7-12).
- The definitions of wind wave footprint are different. In Cagigal et al., (2022), the wave footprint is defined as a circle with radius of 500-km from each track point. In our study, the TC wave footprint(area) is defined as the 6-hourly area within the closed contour of the Hs value threshold ($H_s=2.5\text{m}$). The trend analysis of TC areas was not included in Cagigal et al., (2022), but is one of the key points in our study.

We have referenced Cagigal et al., (2022) in clarification on the uncertainty of satellite data in Lines 682-685. For the wave climate in TC areas, as mentioned in Cagigal et al., (2022) “ although the spatial distribution of SWH may correctly reflect the waves generated by a TC and its mean value, very extreme events are likely to be under-predicted ”, we focus on the relative changes in the metrics of TC wave footprint that are designed to be less sensitive to extreme values(Methods section 3).

Reviewer #2:

The article provides a global assessment of tropical cyclone-derived ocean waves using the ERA5 reanalysis data from 1979 to 2018. The manuscript is well-written and clear. The methods followed are sound, and the conclusions are backed by the results. I believe the article can be accepted almost as is, but I do have an important concern and a few comments:

Response: Thank you for your positive comments and valuable suggestions to improve the quality of our manuscript. According to your suggestions, we have made extensive corrections to our previous draft, the detailed corrections are listed below.

1. My main concern is the values reported for maximum significant wave heights. The maximum values shown are near 10 m, and we know there have been much higher waves measured under tropical cyclones. Please explain the values; why are there no waves above 10m? The validation vs altimeter data may not show higher waves because it depends on the time of the satellite measuring the waves, but there are buoy registers above the shown values; why are those not reflected? Are you really assessing the ERA5 based on tropical cyclone waves? Please try to be clearer in this issue, and please show the comparison between buoys and ERA5 values clearer. I am concern not only with the validation but if the ERA5 is still underestimating tropical cyclone waves, as previous reanalysis, then this is an important thing that needs to be mentioned.

Response:

- In the revised version of the manuscript, we have added more validation of the ERA5 data, including the comparison of the maximum Hs between buoy observations and ERA5 in Supplementary Fig. 10 as the reviewer suggested. In the north Atlantic, buoys observed the maximum values of Hs over 10m. In the comparison, it can be seen that although ERA5 underestimates some of such extreme events, it can still well capture some others. The mean relative error of 10.7% globally. However, part of underestimation for the maximum Hs values is related to a very small number of observations recorded (as the first reviewer pointed out). We have clarified this point in the revised version (Lines 682-685).
- To mitigate this uncertainty in the largest waves, we focus on relative changes in the TC wave footprint that are less sensitive to the extreme values. The TC wave footprint is defined by a minimum threshold of significant wave height (Hs) that the wave reanalysis can well estimate under TC conditions. This definition of TC wave footprint also allows a large sample size of TC wave data for detecting a robust trend. To express this issue clearly, we also added a subsection on the advantages and caveats of ERA5 wave data in Lines 721-760.
- In all, we conclude that ERA5 reasonably captures the maximum Hs during the TC periods, while there is a larger uncertainty in the extreme high values (Hs>8m) due to very limited observations. We have added this in the Methods subsection 6 (Lines 742-752).

2. The abstract and introduction mention that this is the first global assessment of tropical cyclone waves, but there is a published paper by Timmermans et al. (2017) that does present a global assessment (<https://doi.org/10.1002/2016GL071681>). While that paper is different as it used a global model and not reanalysis data, I think it should be mentioned.

Response: Thank you for pointing this out. We have probably read and cited the paper by Timmermans et al. (2017) in the intro. Timmermans et al. (2017) evaluated the TC wave changes under climate scenarios, while our study focuses on the historic trends of TC waves. We also rephrased the sentences in Lines 22-23, to reflect the distinction of our study vs previous studies.

3. Extended table 1 shows the same values for swell in the NH and the SH; please check that this is not an error.

Response: We are sorry for the error. This has been corrected in the revised manuscript.

Reviewer #3:

This manuscript reports trends of tropical cyclone (TC) wave activity on a global scale over the past four decades, based on state-of-the-art reanalysis data and additional information from best-track data, moored wave buoys and satellite altimetry. The authors find that while TC winds have increased by ~3%/decade, TC wave metrics exhibit higher, statistically significant trends for TC wave heights (4%/decade), surface area (7%/decade) and energy (11%/decade). These results are certainly novel and original, they fill a knowledge gap in the sense that such information was lacking for historical TC trends that have focused on atmospheric parameters (winds, rainfall, tracks, etc.) and was only available in the context of future projections from a handful of recent studies. The marine response to TCs, including both waves and storm surge, is now recognized as a major hazard for society, infrastructure and ecosystems, implying that the expected impact of this manuscript could be broad and multidisciplinary.

However, the manuscript in its current form suffers from a number of flaws. The most important are that it is vague in numerous instances, using ambiguous writing and formalism that may be deceptive and do not allow reproducing the study; several additional analyses are also needed to better support (or even simply support) some of the authors' claims; some interpretations being made are questionable; some methodological aspects lack proper justification; addressing the numerous minor comments will also require significant work. Detailed major and minor comments are provided below. For these reasons, I recommended publication in Nature Communications after major revision.

Response: We thank the reviewer for useful comments. In this revised version of the manuscript, these comments have been carefully considered and addressed as follows.

Detailed Comments

Major Comments

- Absolute vs. relative trend values. In general, I find that expressing trends as percentages is ambiguous and I prefer using dimensional units. It is understandable that the authors made this choice to be able to compare trends in different atmospheric and wave variables, both from the results they report and from the published literature (e.g. TC rain rates). However, in such case the reference value has to be specified clearly from the start, i.e. in the abstract, the introduction and wherever needed including the discussion&conclusion section. It is not the case here: as far as I know, the first instance is in the caption of Figure 1, i.e. only in the results section. Thus, those readers who will only browse quickly through the article (abstract, introduction, conclusion, perhaps the figures and not necessarily the full captions) may make a wrong interpretation. For those readers who read the paper more carefully like myself, they may be left with interrogations for several pages until the answer finally shows up, which is undesirable. For instance, an increase of 4% per decade may be interpreted as 4% of the baseline value (e.g. from 1979), but also as 4% of a running reference value that would change (increase) every decade, therefore with a nonlinear trend. This may sound trivial but dimensional units are unambiguous, while relative changes are particularly appropriate for integrated values over long period (e.g., the max height of TC waves has increased by 16% over the last four decades). They are perhaps more appropriate for the abstract, introduction and conclusion section, especially if the 1979 value is used (not the interannual yearly value but the value from the regression line). A clear definition of the reference value used in the reported relative trends would greatly improve clarity of the results for a diverse readership and the wider audience.

Response: Thank you for this very good point. As the reviewer mentioned, we preferred relative trends over absolute trends partially because they are used to compare with trends in other TC variables (e.g. intensity) and because they are less sensitive to the extreme large waves which are more uncertain in ERA5 (Lines 75-77). In the revised version, we added a clear clarification on relative trends, including explicitly specifying the reference values, in the introduction (Line 86),

the main text (before describing the relative trends) (Lines 119,149-150), conclusions/discussion (Lines 303-305), and figure captions (Lines 133 and 218). In the main text, we used relative trends in the main figures, and included absolute trends in tables in the supplementary materials. We hope these improvements significantly increase the readability of this manuscript.

- 4th paragraph of the discussion section, L250-251: I am not fully convinced: the authors' interpretation is based on the assumption that lifetime max winds are increasing (just like the 6-hourly max winds), whereas based on ED Table 1, this is not true for the WNP, NI and SP (and less robust for the EP compared to the increase in the 6-hourly max winds), leading to non-significant NH trends. The basins where lifetime max wind trends are/are not robust tend to match those where lifetime TC wave height trends are/are not robust very well (only exceptions are NA and, to a lesser extent, EP). The authors should not ignore this result and rephrase their statement.

Response: Thanks for this useful comment. We have clarified this point clearly in the revised version. Please see Lines 347-357.

- Same paragraph, L253-254 (see also L28-30): This sentence is ambiguous for various reasons:

1. Increases in TC wave parameters are in the 4-11%/decade range. Ref. 28 reports a 12%/decade decrease in inner-core TC rainfall, which is a much larger value.
2. But maybe the authors actually referred to total TC rainfall including both the inner core and the outer region? Total TC rainfall has been shown to increase by 4%/decade according to Ref. 28 and by 13%/decade according to Ref. 29, which are values of the order of those reported here for ocean waves. The latter are thus not larger, unless the authors meant something else that they would need to clarify.
3. A word of caution here though: the trends reported here are for the past four decades, instead of the past two decades for TC rainfall. Have you tried computing the trends over the 1999-2018 period? This would be useful, both to allow better comparison with TC rainfall changes, and to assess if the TC wave increase may be changing over time.
4. Last, those readers unfamiliar with the works of refs. 28 & 29 may think that these lead to opposite conclusions, judging solely from the paper titles. Yet, both studies have similar conclusions. For this reason also, it is absolutely necessary that the authors clarify this sentence and what part of the TC they refer to when mentioning changes in TC rainfall (i.e., not the inner core).

Response: We apologize for the error.

- We referred to the total TC rainfall. As you mentioned above, our wave height trend value is bigger than the total rainfall trend in ref 28, while smaller than the value in ref 29. We further found that it is not straightforward to compare our results with those in these references as we used different definitions of “relative trends”. So, in the revised version of the manuscript, we focused on the trend comparisons in TC waves and TC intensity.
- More interestingly, We found that these references and our analysis made a common conclusion, which is “the physical impacts of TC (such as total TC rainfall) over the satellite era are found to have more robust trends than TC intensity itself due partially to uncertainty in observing storm intensity”. This is clarified in Lines 361-364.

We hope this will address the reviewer's comment.

- Method reproducibility: several instances of vague descriptions that do not allow reproducing the analyses. e.g. L342-344 (unclear: please expand or at least provide a reference - see also L369-370); L346-349 (please provide more details and/or refer to published works describing the method(s)); L577-578 (Which index is being used, Nino3.4 ? What are the dates considered for the TC season?)

Response: Thanks for your suggestions. We meant to briefly describe the TC tracking method as it has been described in detail in previous publications, and it is not new here. But, we take this point as we agree that clearer reference citations are needed. Please see the improvement in Lines 388-404.

We have clarified how the yearly ENSO index is calculated. Please see Lines 806-808.

- Insufficiently supported choice of methods. e.g. L358-359 Please elaborate a little more: why are TC positions located between 40° and 60° latitude being excluded, after being considered in the first place (i.e. why not use the 40° threshold from the start)? Also, this methodology will not only exclude extratropical cyclones, but also post-TCs after their extratropical transition. Why has this choice been made? It may have been possible to separate between these and purely extratropical cyclones by also considering the initial position latitude. Related to L364: I don't understand: how is a comprehensive analysis including the post-TC stages possible if positions beyond 40° latitude are being excluded?

Response: We are sorry for the confusion.

- The 40°N-40°S cut off is used for the truncating process after the matching process. The truncated processes are applied to make sure the ERA5 TCs have the same lifetime as observed and to reduce the uncertainty in high-latitude waves which are not related to TCs.
- The main results are based on the truncated tracks. We meant to say that we also use the full tracks (matched but not truncated) to support the main conclusions made based on the truncated tracks.

We have improved the methodology description in this subsection and clarified why and how we matched and truncated the ERA5 TC tracks using IBTrACS. Please see Lines 406-434. We hope this will address the reviewer's comment.

Another example is found L522-523: Why? The analyses in this paper are mostly not restricted to the time of TC lifetime intensity.

Response: Thanks for this comment. This analysis with satellite data aims to validate the quality of ERA5 wave data under TC conditions. The comparison of wave data when TC at the lifetime maximum intensity is because waves are mostly largest at the time of TC lifetime maximum intensity, and thus the proportion of TC waves ($H_s > 2.5\text{m}$) along the satellite track is larger than any other time. We could have included the satellite data in the whole TC lifetime. But this will require a large effort and produce a very large dataset. This will also put more weight on small H_s values when TCs are in a relatively weak stage. We think this will not provide significant extra benefits to our study. We should have specified this point in the previous version (but obviously we did not). In this new version, we have added a clarification for this. Please see Lines 674-679.

- Equation (8): I find this equation a bit ambiguous, particularly the definition of TCW duration, in the sense that the recurrent case of simultaneous multiple storms is not explicitly mentioned. I find that it would be clearer to include an index over each TC track lifetime (i.e., 6-hourly time steps) and another one over each TC track in a given year. Perhaps the authors have rigorously considered all TCW footprints, even when several footprints occur at the same time, and found it practically (e.g. computationally) more convenient to frame the calculation this way. Is that so? Anyway, this has to be clearly specified in the text.

Response: Thanks for your suggestions. To make the definition of TC wave duration more clearly, we defined two durations. One is duration for individual tracks, and the other is an annual duration for accumulated tracks (Lines 536-539).

Related to this comment, ED Fig 6b illustrates the calculation of TCW duration for a single storm. This is very ambiguous and somewhat contradictory with the layout of Eq (8), which does not

consider TC wave duration for individual storms, only that accumulated over a year. This point definitely requires clarification.

Response: In the caption of Supplementary Fig. 7, we pointed out it is wave duration for individual TCs, and also we added the expression between individual TC duration and annual accumulated duration in L536-539.

Regardless, another related concern is regarding the treatment/interpretation of duration and associated trends in the main text. Because it is accumulated over a year (as specified in the Fig. 4 caption), it depends on both individual storm duration and storm counts (frequency). Yet, this term is ambiguous as many readers are likely to interpret it as average storm duration, which is misleading. The authors need to address this issue by either rephrasing this term or by also including a more detailed analysis of TC frequency and average duration trends.

Response: As mentioned above, to make the definition of TC wave duration more clearly, we defined two durations. One is duration for individual tracks, and the other is an annual duration for accumulated tracks (L536-539). We also defined the annual accumulated duration in units of days to reduce misunderstanding of it. All these have been clearly clarified in the results, figure captions and tables.

- Interpretation of the buoy comparison section as validation of TC wave area (L513-515): This claim is way too strong and I am afraid I must disagree. Even in the simplest case where we neglect TCW area variations in the cross-track direction and only focus on the along-track axis (reducing the spatial dimension to one), event duration at a particular point will depend not only on footprint spatial scale, but also on TC translation speed. To maintain this claim, the authors need to validate ERA5 TC translation speeds e.g. with IBTrACS data, using the 656 events (i.e. track portions) that relate to the duration analysis.

Response: Thanks. This is a good point. Following your suggestion, we compared the TC translation speed in ERA5 TCs and IBTrACS (Supplementary Fig. 11 in the revised version). We found that the ERA5 TC translation speed agrees well with the values calculated from IBTrACS tracks, with $r=0.98$ for all the 6-hourly TC track points. The correlation is similar in the three ocean basins. The ERA5 estimates the average translation speed well with mean relative error (MRE) $<0.71\%$, and RMSE <1.2 km/h. Thus, we think this strongly supports our statement on the duration. The TC translation speed comparison is described in L644-655.

- Conclusion of the WEVERYS analysis regarding the robustness of ERA5 trends (L549-551): I am not convinced by the claim that trends are similar: for example, in the EP and NA (and in the NH), the upward trend is significant for ERA5 but not for WEVERYS; in the SP both trends are positive and non-significant but it is much larger in WEVERYS. In fact, since the purpose of this figure is to compare ERA5 and WEVERYS, only the common 1993-2018 period should be shown and considered for computations. How does the comparison change when the ERA5 period is reduced to after 1993? How do the results impact the paper conclusions and the overall robustness of the reported increasing trends of TCW activity? The authors should also provide the numbers here (trends in ED table) and discuss the results of this section in much more detail.

Response: Thank you for your suggestion. Following this suggestion, we restricted the comparison between ERA5 and WEVERYS to 1993-2018 (Supplementary Fig. 13 in the revised manuscript). The absolute and relative trend values are provided in Supplementary Table 1. We confirm that the TCW trends are identical during 1993-2018 in these two datasets. We also extended the discussion about comparisons between ERA5 and WEVERYS in Lines 706-719. We hope these improvements will address this comment. Thanks!

Minor Comments

- L35: Ref. 1 is about projections of extreme waves but has limitations for TC waves due to model resolution. Ref. 2 is about the role of coral reefs for coastal protection against ocean wave hazards, but is only marginally related to TC waves (at least, not explicitly in the paper). Ref. 3 is a case study for a particular TC but is still relevant here. I would suggest the authors also include their Ref. 30 here, because it overcomes the aforementioned limitations of Ref. 1 with high-resolution model forcing. Other references that the authors may wish to consider in this respect are Shimura et al. 2015 (methodologies used to derive TC- and non-TC wave fields are described in detail, unlike Ref. 30 which still uses the same methods), Timmermans et al. 2017 (earlier global-scale study with high-resolution forcing) and Belmadani et al. 2021 (high-resolution forcing for the North Atlantic). In addition, I suggest the authors also include a reference to Rappaport 2014 in order to support these introductory statements more directly.

Belmadani A, Dalphiné A, Chauvin F et al (2021) Projected future changes in tropical cyclone-related wave climate in the North Atlantic. *Clim Dyn* 56:3687–3708. <https://doi.org/10.1007/s00382-021-05664-5>

Rappaport EN (2014) Fatalities in the United States from Atlantic tropical cyclones: new data and interpretation. *Bull Am Meteorol Soc* 95:341–346. <https://doi.org/10.1175/BAMS-D-12-00074.1>

Shimura T, Mori N, Mase H (2015) Future projections of extreme ocean wave climates and the relation to tropical cyclones: ensemble experiments of MRI-AGCM3.2H. *J Clim* 28:9838–9856

Timmermans B, Stone D, Wehner M, Krishnan H (2017) Impact of tropical cyclones on modeled extreme wind-wave climate. *Geophys Res Lett* 44:1393–1401. <https://doi.org/10.1002/2016GL071681>

- L37: Rappaport 2014 also relevant here.

Response: Thanks for your above suggestions. We have carefully read and properly cited these references in the revised version of the manuscript. Please see Line 38.

- L41: These references are all for oceanic WAVE extremes. Other oceanic extremes may include extreme sea levels, but also e.g. marine heatwaves. The authors should either change the terminology being used, or include additional relevant references (e.g. Vousdoukas et al. 2018, Frölicher et al. 2018).

Frölicher, T.L., Fischer, E.M. & Gruber, N. Marine heatwaves under global warming. *Nature* 560, 360–364 (2018). <https://doi.org/10.1038/s41586-018-0383-9>

Vousdoukas, M.I., Mentaschi, L., Voukouvalas, E. et al. Global probabilistic projections of extreme sea levels show intensification of coastal flood hazard. *Nat Commun* 9, 2360 (2018). <https://doi.org/10.1038/s41467-018-04692-w>

Response: Thanks for this useful suggestion. We have included references for other oceanic extremes (e.g. extreme sea levels and heatwaves) in L42-44.

- L46: Also consider including a reference to Timmermans et al. 2020 where the sensitivity of wave height trends to the chosen datasets is assessed using data from Ref. 12 and other recent datasets.

Timmermans, B. W., Gommenginger, C. P., Dodet, G., & Bidlot, J.-R. (2020). Global wave height trends and variability from new multimission satellite altimeter products, reanalyses, and wave buoys. *Geophysical Research Letters*, 47, e2019GL086880. <https://doi.org/10.1029/2019GL086880>

Response: Thanks for your suggestion. We have included the references in L51.

- L48-49: long-term HISTORICAL changes. Studies of future projections are available (see above) but not of observed changes, indeed.

Response: Thanks for your suggestion. We have improved this sentence (L73).

- L56: The term 'global warming' is a bit deceptive for Ref. 22, which is a process study of idealized TCs and associated wind & waves under varying background ocean temperature conditions, rather than a study of future projections. As far as I know, Ref. 24 does NOT deal with TC waves. Ref. 23 is about future projections of wave extremes including TC waves. Therefore, and because the next sentence is about historical changes, the authors should refer to 'the FUTURE impact of global warming on TC waves'.

Response: Thanks for your suggestion. We have improved the citation of these references (L66-67).

- L58-59: Yes, but the authors could be a bit more specific, stating how these influence wind fetches and thus wave generation and propagation, preferably with some reference(s), e.g. Ref. 25. The authors may also specify the difference in TC wind sea and TC swell.

Response: Thanks for your suggestion. The interpretation of how the TC properties impact wave generation and propagation was included in the Results part in L120-123. For the difference in TC wind sea and TC swell, we added the definition and equation of identifying wind sea and swell in the Method part in L465-474.

- L67: the uncertainty is quite large with err ~80% of b, which also seems to be the case for the other TC basins. This may be worth mentioning. See also L69, L71. Also, unit is cm for the absolute trend values.

Response: Thanks. We are sorry for the mistake in the unit. The absolute trend has been removed in the revised main text. As for the error bar, we showed this values in the main text because the values were the indicator for uncertainty. We have added statement of the error bar in the main text L86.

- L69: Large uncertainty here as well => misleading use of 'even more quickly'

Response: Thank you for pointing this out. We have rephased the sentence in L91-93.

- L96-97: Could you please discuss or at least mention the differences between the NH and SH values? Why are the latter clearly larger?

Response: Thanks for your suggestion. We have included the interpretation in L119-123.

- L108: Extended Data (ED) Table 1 should be cited here (i.e. from the start).

Response: Thanks for your suggestion. We have added the citation in L156-157.

- L108-109: Do not repeat the absolute trend values here, they are already in the table. It will help improve the reading flow. Also L133-135 for the relative trend values.

Response: Thanks for your suggestion. We have deleted the absolute trend values in the main text, and the relative trend values in L188-189.

- L111: These relative trends. Also L147.

Response: Thanks for your suggestion. We have deleted the values.

- L116: "nonlinear wind-wave interaction": Please elaborate more on this possibility. "decrease of TC translation speed": Could you comment on the non-significant (at the 95% level) negative trends shown on ED Fig. 1? Please discuss these results in the context of Refs. 17 & 18 but also Moon et al. 2019 and Lanzante 2019 who questioned the results of Ref. 18.

Lanzante, J.R. Uncertainties in tropical-cyclone translation speed. *Nature* 570, E6–E15 (2019). <https://doi.org/10.1038/s41586-019-1223-2>

Moon, IJ., Kim, SH. & Chan, J.C.L. Climate change and tropical cyclone trend. *Nature* 570, E3–E5 (2019). <https://doi.org/10.1038/s41586-019-1222-3>

Response: Thanks for your suggestions. We have extend the illustration of "nonlinear wind-wave interaction" in Discussion part (L330-334). We also discussed the non-significant trend of translation speed, and mentioned the uncertainty in global decrease of translation speed in L161-165.

- L119: $p > 0.05$. What about the 0.1 threshold? It is used elsewhere in the paper for confidence intervals

Response: For the threshold of 0.1, the increasing trend of global TC translation speed is significant with $p = 0.08$, but for the NH and SH, the trends are still insignificant.

- L120: only hinting at a small effect (if any) from the reduced translation speed

Response: The relationship between the translation speed and the TC wave height is complex. The larger the TC translation velocity, the longer will be the waves subject to group velocity resonance. The energy of TC-generated waves will then also be larger. But if the TC translation velocity exceeds a threshold value, TC-generated waves cannot reach group velocity resonance, and in their course of development, will travel backwards. Therefore, we cannot find a significant correlation between the increase of wave height and decrease of the translation speed (Kudryavtsev et al., 2021). In this sentence, we changed the 'effect' to 'contribution', only state the small contribution from the reduced translation speed on the large increase of TC wave height.

Kudryavtsev, V., Yurovskaya, M., & Chapron, B. (2021). Self-Similarity of Surface Wave Developments Under Tropical Cyclones. *Journal of Geophysical Research: Oceans*, 126(4), e2020JC016916.

- Fig. 2: Why are the NH values larger than SH values here (also the case on ED Fig 4), whereas it is the opposite on Fig. 1?

Response: In Fig. 2 and Supplementary Fig. 5, the wave heights are the the annual mean of 6-hourly maximum height and the annual mean of lifetime maximum height. The maximum wave heights highly depend on the TC intensity. As the the overall TCs in SH is less intense than the NH, it is reasonable that the NH values in Fig. 2 and Supplementary Fig 4 are larger.

The wave heights in Fig. 1 are values by mean of the 6-hourly height (m) in the NH and SH, for the two epochs 1979–1998 and 1999–2018. In the calculating of 20-year mean values, the wave fields in Fig. 1 have been rotated in the TC orientation. Related to the lack of land boundary in the SH, the effective wind fetch is much longer, resulting in larger waves. In contrast, in the NH, the fetch is much shorter, the extreme waves generate locally with relative smaller spatial scale close to the eyewall (Line 120-123).

- L131: The basin-wide trends in the maximum height of TC waves and surface maximum wind speed

Response: Thanks for your suggestion. We have rewritten the sentence. Please see L184.

- L132-133: Only significant at the 90% level for the weaker NI and SP trends.

Response: Thank you for pointing this out. We have pointed out that the trends in the NI and SP are only significant at the 90% confidence in L188-189.

- L135-137: Right, but the NA is the basin with the second largest change in TC intensity (3rd is SI and is notably lower), while the WNP is the basin with the second largest change in TC wave height (3rd again is SI). So yes, other factors affect the height trends, but TC intensity is apparently dominant.

Response: Thank you for pointing this out. Yes, TC intensity dominant the change of TC waves. But other factors include changes of TC translation speed, and the nonlinear wind-wave interaction also play a role. We have mentioned this in L157-159. We also discussed the nonlinear wind-wave interaction in L330-334. That explained the reason why largest changes in TC intensity cannot induce largest TC wave change.

- L140: for different thresholds of TC wave heights. Extended Data Table 2 should also be cited here to quantitatively supports several of the statements made in this paragraph.

Response: Thanks for your suggestion. We have included the citation here. Please see L196.

- L142: become slightly larger

Response: Thanks. We have improved the sentence.

- L143-144: I assume this is true, but it is not evident from the figure which only shows that all three trends are significant at the 95% level. Or did you simply infer that from the relative increase in the error levels (compared to the trend values) ? Please clarify.

Response: Thank you for pointing this out. We have included citation to the Supplementary Table 2, and mentioned larger error level compared to the trend values with the larger height threshold. Please see L201-202.

- L145: Remind the 2.6% value or refer to ED Table 1.

Response: Thanks for your suggestion. We have included "2.6±0.8%/decade" in the revise manuscript. Please see L204.

- L146: Please rephrase as "The area trends are much bigger in the SH than in the NH, especially for large waves"

Response: Thanks for your suggestion. We have rephrased this sentence. Please see L205.

- L146-147: "The SI is an important contributor to the SH signal": Why is that so? How many tracks in the SI vs SP?

Response: Yes, The SI is an important contributor to the SH signal, because more TC events occur in the SI. For the annual mean TC number, there is 13 TCs in the SI, while 9 TCs in the SP. Besides, the increasing trend of TC wave area in the SI is significant, but the values in the SP is insignificant. The increasing trend of SH is mainly impacted by the trend in SI. We have added the 40-year mean of TC number for each basin in the caption of Supplementary Fig. 1.

- L147-148: sensitive to the height threshold (larger with larger thresholds, as with global averages)

Response: Thanks for your suggestion. We have rephrased this sentence. Please see L207-208.

- L148-149: So what? What does it suggest?

Response: Thank you for pointing this out. We have extended the explanation about the reason why the relative trends of area are larger with larger thresholds in the SI in L210-213.

- L153-154: The 40-year mean value should be indicated on each subplot

Response: Thanks for your suggestion. We have added 'relative to the 40-year mean' on each subplot.

- L155-156: The only one in this case is however significant at the 90% level, see ED Table 2

Response: Thanks for your suggestion. We have rephrased this sentence. Please see L221.

- L160-162: please start the sentence with "Among other processes,"

Response: Thanks. Because we did not mention other processes in maintaining the energy balance at the air-sea interface, "Among other processes," is ambiguous here.

- L172: Here these values are relative to the global trend expressed in energy units (kJ/decade), whereas the paper systematically focuses on relative trends (in %), including in the previous sentence. This is confusing and the authors should fix that.

Response: Thank you for pointing this out. We have added the definition of these percentage in L240-241.

- L173: Better rephrase as main, major or dominant components. PCs have another meaning (see EOFs) and this term is therefore ambiguous.

Response: Thanks for your suggestion. We have rephrase as contributing components. Please see L242.

- L174-176: I don't understand these values. From ED Table 2, TC wave area increases by 6.7%/decade and TC wave energy by 11.1%/decade (I don't know about duration, see my next comment). 6.7 is not the half of 11.1. Or maybe this relates to Eq.s (15-16) somehow, where trends in E and in Δ^2 , both expressed in kJ/decade, are compared, but then the reader will likely wish to see the corresponding values (see further comment regarding wave height trends).

Response: Thank you for pointing this out. The values is related to Eq.s (16-17), and the contributions of the area and duration can be calculated by the relative values of Δ^2 and Δ^3 to the trend values in wave energy. We have corrected the values in L248-249

- L176-177: Duration trend values should be included in ED Table 2 (or in a separate table) to better support this claim and e.g. indicate if the trend is significant at the 90% level, as for other trends.

Response: Thanks for your suggestion. We have included the duration trend values in Supplementary Table 2.

- L177: I don't understand. A small negative value is provided here, whereas trends in TC wave heights have been previously shown as positive (Fig. 2a, ED Table 1). I understand that this analysis follows Equation 16 and probably refers to the term named Δ^1 , which is proportional to H'. How can the trend in H' be negative and that in TC wave heights be positive at the same time? This is very confusing.

Response: The wave height term in Eq 17 is proportional to H' . The expression of H can be found in Eq 13, which is the annual mean value of H_s over the annual TC area and duration. The maximum TC wave height in Fig. 2a and Supplementary Table 1 are the annual mean of 6-hourly maximum height. Therefore, the annual mean of 6-hourly maximum height increases, while the annual mean of 6-hourly maximum height decreases. The reason why the annual mean of 6-hourly maximum height decreases is the TC wave area expansion with increasing TC intensity. Therefore, there are two different trends of the mean and maximum wave height as the increasing of TC intensity. The first one is the maximum wave height increases. The other one is the TC area increases, resulting more areas with $H_s > 2.5\text{m}$ and proportion of small magnitudes of H_s increases. Thus, the trend of H' is negative.

- L182-183: I think it would make more sense to divide these values by four in order to have these estimates expressed in days

Response: Thanks for your suggestion. We have replotted the figure, and expressed duration in days. See Fig. 4.

- L183: WNP and NA: It would be useful to include duration trend values for all basins on ED Table 2 (see previous comment regarding duration global trend)

Response: Thanks for your suggestion. We have included the duration trend values for all basins on Supplementary Table 2.

- L187-188: Extended Data Fig. 3a may also be referred to here

Response: Thanks for your suggestion. The Supplementary Fig. 4a in the revised manuscript has been referred here. Please see L268.

- L189: NI: Trend does not appear as significant (at the 95% level) on ED Fig. 2b. In fact, it is only significant at the 90% level according to ED Table 2. Strikingly, the associated confidence interval crosses the zero-line: how can the trend then be significant considering the definition provided L595-597 (same for the swell component)? Or maybe the 90% CI is smaller than the trend value? The authors should check and elaborate on this (or correct in case of any errors).

Response: Thank you for pointing this out. We are very sorry for our incorrect writing. Because the relative trend value in the NI is large than the WNP and SP. We did not notice the significance level. We have included illustration of the significance level in the NI in L270.

- L195-196: Again, I am concerned with the significance of the energy trend in the NI basin.

Response: Thank you for pointing this out. We have included illustration of the significance level in the NI in L270.

- L198-199: I must be missing something: how come TC wave heights have a trend significant at the 90% level but not $\Delta 1$? See related comments on the previous page.

Response: Thank you for pointing this out. The TC wave heights in Supplementary Table 1 related to the annual mean of 6-hourly maximum height, while the H' in $\Delta 1$ related to the annual mean value of H_s over the annual TC area and duration. The corresponding trends of these two wave height terms to the increase of TC intensity are different.

- L199: Over 90%: $64+25=89 < 90$

Response: Thank you for pointing this out. We have changed “over 90%” to “about 90%” in Line 290.

However, adding up signed % contributions seems deceptive to me because negative percentages are possible, leading e.g. to $57+49=106 > 100$ in the NA.

Response: Thank you for pointing this out. As shown in Eq. 17, the wave energy can be divided into 8 terms. The trend of wave energy is induced by the 8 terms. For simplicity, we only analyzed the first three contributing terms. Therefore, the summary of the contributions of the three terms are not equal to 100%. Because the other 5 terms also play a role in different basins. Detail description of the method can be found in Methods section 4.

- L200-201: Yes, but this decrease does not appear significant. In general, the results of this analysis for the WNP appear similar to those for the SP: they should be grouped together and this should be made explicit (even though there are slight differences such as that commented here by the authors).

Response: Thank you for pointing this out. We have rephrased the sentences, and grouped together the results of SP and WNP in L280-288.

- L215-216: Not entirely accurate: trends in TC wave area are insignificant in the NI and SP

Response: Thanks. We have mentioned the insignificant trend in NI and SP in L308-309 of the revised manuscript.

- L239-241: Please rephrase as "the maximum trends of the 6-hourly height in the spatial fields, about 6-8%/decade (Fig. 1c,f), are larger than the trends of 6-hourly maxima, ~4%/decade". Fig. 2 alone is useless to support this sentence because trend values are not apparent. The authors should also quote ED Table 1 here. Also, it is Fig. 2b,c which should be quoted because Fig. 1c,f are for hemisphere-scale patterns and values.

Response: Thanks for your suggestion. We have rephrased the sentences, which can be found in L334-336 of the revised manuscript.

- L241-244: These two sentences are very unclear. What spatial average? What do you mean by fixed locations? Etc. Please rephrase.

Response: Thank you for pointing this out. We have rephrased the sentences in L336-343 of the revised manuscript.

- L246: Please rephrase as "Based on the idealized model, Ref 22 also argued"

Response: Thanks for your suggestion. We have rephrased the sentence. Please see L328-333.

- L248: The authors should also mention that they calculated the trend in the lifetime max TC intensity since they mention 'increases of already very high winds' in the previous sentence.

Response: Thanks for your suggestion. We have rephrased the sentence. Please see L347-349.

- L250: Rephrase as "are usually smaller and less detectable". In the SH, relative trends are very similar and absolute trends are actually larger with lifetime-maxima, which seems to originate from the SI (ED Table 1).

Response: Thanks. We have rephrased the sentence. Please see L350.

- L258-259: I would not insist much on this study because of its limitations for TCs (see first minor comment).

Response: Thanks for your suggestion. We realized the limitation of the reference, and changed to cite a new article of Shimura et al., (2017). Please see L369.

Shimura, T., Mori, N., & Hemer, M. A. (2017). Projection of tropical cyclone-generated extreme wave climate based on CMIP5 multi-model ensemble in the Western North Pacific. *Climate Dynamics*, 49(4), 1449-1462.

- L271, 297, 316, 322, 641: Avoid using et al. here, there are only 6-8 co-authors.

Response: Thanks for your suggestion. We have includes all the authors. Please see in the References part.

- L338-339: Ref 33 is for ERA5 data, not for TC identification in such data, as the sentence suggests.

Response: Thank you for pointing this out. We have included references related to TC identification in L386.

- L354: Please provide the following reference for IBTrACS data:

Knapp, K. R., Kruk, M. C., Levinson, D. H., Diamond, H. J., & Neumann, C. J. (2010). The International Best Track Archive for Climate Stewardship (IBTrACS), *Bulletin of the American Meteorological Society*, 91(3), 363-376.

Response: Thanks for your suggestion. We have cited the above reference.

- L357-358: i) State that this corresponds to the threshold for tropical storms

Response: Thanks for your suggestion. We have added the statement. Please see L413-416.

- L363: Actually, reference 35 was published before ERA5 was issued and only considers the older ERA-Interim. Can the authors include a more recent reference specific to ERA5 data? Or rephrase

Response: Thanks for your suggestion. We have rephrased the sentence. Please see L411.

- L367-368: I found no indication that MSLP has been used in the paper. Please remove. Also L369.

Response: Thanks. We have deleted MSLP.

- L368-369: Did you mean the TC track position here? This makes for 6-hourly max wind time series for each track, from which one value of lifetime max wind speed may be extracted.

Response: We are sorry for the confusion. We have rephrased the sentence. Please see L400-404.

- L377-378: rephrase as "could be in slightly different locations."

Response: Thanks for your suggestion. The sentence has been rephrased. Please see L441-443.

- L383-384: Redundant with first sentence of previous subsection, please remove.

Response: Thanks. We have deleted the first sentence.

- L385-386: While this may be true to a certain extent, I think that the authors should be more careful with this statement: see Timmermans et al. 2020.

Response: Thanks. We have deleted the sentence.

- L398: Wind waves usually refer to surface ocean gravity waves because they originate from a wind forcing, even if a part of the energy propagates as free waves (i.e. swell). The authors should rather call them wind sea, which is more standard. Here and across the manuscript.

Response: Thanks. We have changed “wind waves” into “wind sea” in the revised version. Please see L465.

- L401: Please clarify what you mean by mixed waves: is it the total wave field (adding up wind sea and swell components?)

Response: Thanks for your suggestion. We have included the definition of mixed waves in L476.

- L423-424: please use a consistent notation, preferably A_{6h} (subindex). Also L452-454: H_{6h}

Response: Thanks for your suggestion. We have used consistent notation in the revised manuscript. Please see L508-513.

- L426: rephrase as "S(i) is the area (km²) associated with grid point i".

Response: Thanks. The sentence has been rephrased. Please see L512-513.

- L433: I cannot see how Ref. 33 may be relevant here. As far as I know, it does not present the method for computing wave energy. If the authors would just like to remind the data being processed, they should simply refer to ERA5.

Response: Thanks for your suggestion. We have referred to ERA5 here. Please see L526.

- L466: Similarly to Eq(9). Mathematically, Eq(12) could also be derived from Eq (10) and Eq (11), but I believe this is not what the authors mean here.

Response: Thank you for pointing this out. We have rephrased the sentence in L555-556.

- L485-486: Have you checked this thoroughly for all the TC basins?

Response: Yes, we have checked for all the TC basins. The first-order terms are the main contributing terms.

- L503: How many for each buoy? For each of the 3 basins? See comment on ED Fig. 6a below.

- L508: smaller sample size. Yes but numbers are not being provided.

Response: The detailed buoy information have been included in the new Supplementary Table 3.

- L510: mostly in the small values of duration. This is practically impossible to tell from the figures (except for the WNP) given the sample sizes and use of the same color for data points and the $x=y$ straight line. A Q-Q plot would be much better to support the authors' claim. See also L529: A Q-Q plot may also be useful here, although not as critical as for Supplementary Fig. 7.

Response: Thanks for your suggestion. we have included a Q-Q plot in the new Supplementary figure 9. From the figure, it can be seen that the underestimation of duration is mainly induced by the values at small percentiles(Supplementary Fig. 9, L626-633).

- 510-511: 'Probably' seems too strong unless the authors can rely on some published reference and/or other more solid evidence.

Response: Thank you for pointing this out. We have rephrased the sentence based on the Q-Q plot and scatter plot between buoy and ERA5 data. Please see L626-633.

- L511-512: Again it is impossible to see from the plots. The 3-duration threshold is not even indicated.

Response: Thanks. We have removed this sentence.

- L524: ED Fig. 8 caption mentions 3-4 km

Response: Thanks. We have corrected it. Now it is consistent with the figure (Supplementary Fig. 12 in the revised version).

- L525: How many values & events for each basin (and hemisphere) ?

Response: Thank you for pointing this out. We have included the sample sizes in the figure caption (Supplementary Fig. 12 in the revised version).

- L532: Vague. Do you refer here to the few points with severely underestimated Hs in ERA5, particularly in the NH? Did you explore these individual cases to check if they relate to e.g. underestimated wind speeds in ERA5?

Response: Thank you for pointing this out. We have removed the sentence.

- L548-549: No. They are very similar indeed, but not identical. In addition they differ more in the SP basin: any explanation for such discrepancy?

Response: Thanks. We have improved this sentence. Please see L709-712.

- L558: area of TCWs: this is not shown in the tables. I assume the authors meant energy rather than area.

Response: Thanks. We have corrected this error. Please see Supplementary Table 2.

- L558-559: This paragraph only deals with height. References to energy and ED Table 2 should be moved to the next paragraph.

Response: Thanks. We have deleted the reference to Supplementary Table 2 here. Please see L769-770.

- L562-564: Change to 'are not strongly dependent' as the agreement is only qualitative. Also, I noticed from ED Table 1 that this is not true on a basin scale: compare the trend significance levels for the WNP, NI and SP (the agreement is better for the other basins). This should not be eluded and the authors should therefore limit their claim to large spatial scales and/or discuss more precisely the discrepancies found for these basins.

Response: Thanks. We have improved the sentence. Please see L776.

- L570: rephrase as "also mostly retain the same signs"

Response: Thanks. We have rephrased the sentence. Please see L782-783.

- L583-586: What about the energy trend? Are the significance levels affected? Also, it would be interesting to see if this result holds for the 2 hemisphere and the 6 basins. In this respect, detailed

results in table format, including also absolute trends would be useful.

Response: Thanks for your suggestion. We have included an additional Supplementary Table 4 for the trends after removing ENSO effect (Supplementary Table 4 in this revised version). We find that the ENSO does not significantly change the trend values and significance of the trend values in both global and basin scales. We have added a clarification in L801-804.

- L600-601: I disagree with this statement, non-significant trend values are provided in the ED Tables. The authors should be clear that this statement does not apply to Extended Data.

Response: Sorry for the confusion. We have included clarification in the captions of Supplementary Table 1, 2 and 4.

- ED Fig. 2: How many TC tracks for each basin? How many more TCs in the NH compared to the SH? Can it influence the comparison between hemispheres (and basins) ? Please use the same y-axis ranges for all the basins to allow easier comparison. It would also help comparing with hemispheric averages from Figure 2.

Response: Thanks for your suggestion. We have improved the Supplementary Fig. 3 in the revised version. The 40 year mean of TC count for the six basins (WNP, EP, NA, NI, SI and SP) are 21, 8, 8, 5, 13 and 9. Because the yearly wave energy is defined as the annual accumulated value, yes, the magnitudes of TC wave energy is related to the annual TC number. We have added the average TC number for each basin in the caption of Supplementary Fig. 1.

- L676: 10m maximum wind speed.

Response: Thanks. We have corrected it. Please see caption of Supplementary Fig. 3 in the revised version.

- L680-681: Most of them are however significant at the 90% level, see ED Table 1

Response: Thanks. We have improved the figure caption to reflect this point. Please see caption of Supplementary Fig. 3 in the revised version.

- L701-702: Not entirely sure of what this represents since values are higher than those shown on Fig. 2. I assume it may be the surface max wind speed occurring at the time of TC lifetime max wave heights, averaged annually, but the caption is not entirely clear.

Response: Thanks. The wave height is the annual mean of TC lifetime-maximum TC waves. The wind speed is the annual mean of TC lifetime-maximum 10m wind speed. Because the wave height and wind speed are only averaged for the TC lifetime maximum values, the values are larger than values in Fig. 2. We have rephrased the figure caption to clarify this. Please see caption of Supplementary Fig. 5 in the revised version.

- ED Fig 5: It would be useful to show the Hs field outside the 15° radius to allow the reader to fully judge the relevance of this distance in capturing the TC wave field. Further, in this respect, an individual case such as the one shown on this figure is an illustration but not a demonstration. Nothing is said as to how strong TC Jebi was in general and on 2018-09-01 12 UTC in particular. I suggest the authors compute the average Hs along the 15° circle for each position of each TC track and then show the histogram of these average values. Then repeat for different radii. This should provide convincing evidence that the 15° radius is an appropriate choice.

Response: Thanks for your suggestion. We have improved the figure (Supplementary Fig. 6 in the new version). Supplementary Fig. 6b shows the distribution of the area of the 6-hourly TCW (i.g., where $H_s \geq 2.5\text{m}$) as a function of geodesic radius in the first guess. For small radius (i.e., 5 degree), the identified TCW footprint has a small area around the TC centre, and none of the footprint

exceeds 10^6km^2 . When the radius of the first guess becomes bigger, the sample size of small footprints reduces and sample size of large footprints increases. The distribution of the footprint area becomes stable when the radius is larger than 15 degree. Thus, we conclude that 15 degree of radius in the first guess is appropriate for the TCW footprint defined by $H_s \geq 2.5\text{m}$. Please see L498-506.

- ED Fig6a: Additionally, the authors should provide more detailed information in an ED Table with the precise buoy coordinates, basin, IDs, time periods, % of missing data (at least during the TC seasons), sampling frequency, etc. Also include the number of TC events being considered for each buoy.

Response: Thanks for your suggestion. We have added detailed information about buoy data and associated TCs in Supplementary Table 3 in the revised version.

- ED Fig 6b: Elsewhere in the manuscript TCW duration relates to a TCW footprint over one or several grid points. Here we are at a fixed location, which will tend to underestimate TCW duration since it may be found outside the 2.5-m contour while such contour may still be found elsewhere within the 15° -radius. The authors need to discuss that.

Response: We should have clarified this in the previous version of the manuscript. In the validation of duration against buoy data, we only considered the TC cases in which both ERA5 waves and buoy data have at least one 6-hourly data within the 15° -radius. This is to exclude the cases in which buoy data satisfy the criteria of $H_s > 2.5\text{m}$ and within the 15° -radius, but ERA5 data have no contour overlapping with buoy location. In these cases, ERA5 is more likely to underestimate wave values. This also excludes the cases in which ERA5 data have $H_s > 2.5\text{m}$ contour overlapping with buoy location, but buoy data are $< 2.5\text{m}$. In these cases, ERA5 is more likely to overestimate wave values. We found that with these exclusions, the durations from two datasets are matching well with each other (Supplementary Fig. 8). We found there are 74 TC cases in which ERA5 duration = 0h and buoy duration > 0h, likely related to the underestimation. However, these cases were found to not significantly affect the verification statistics. We have added clarification in the revised version in L620-633.

Also, the use of shading combined with contour Lines connecting the dots is deceptive because it implicitly suggests that the values are linearly interpolated to compute start and end dates/times more accurately. Histograms would be more appropriate in this context (staircase-like graphs). Finally, if this is a demonstration, then you should provide the results for this particular example: is it 6 for ERA5 and 9 for the buoy? I am not sure this is a very good example because the start and end points lie very close to the 2.5m-threshold (except for the ERA5 end point), making it difficult for the reader to see whether these points are included or not.

Response: Thank you for pointing this out. Following this suggestion, we changed this figure to a histograms plot, and chose a better TC case to illustrate the duration estimate. We finally included the results in this case in the figure caption. We hope these will address this comment. Please see Supplementary Fig. 7 in the revised version.

- L720-721: rephrase as "The number of 6-hourly time steps with TC wave $H_s \geq 2.5\text{m}$ (shaded area) represents the TC wave duration"

Response: Thanks. We have rephrased the sentence. Please see caption of Supplementary Fig. 7 in the revised version.

- ED Fig 8: In panels b-d please use the same tick intervals for the y and x axes (every 2 m everywhere). It would also be useful to break the information into the different basins

Response: Thank you for pointing this out. We have replotted the figure (Supplementary Fig. 12 in the new version). We also added a sentence in the Methods to interpret the basin results. Please see this in L681-692.

- L733: rephrase as "(footprint about 3-4 km wide, expanded here for visual clarity)"

Response: Thanks. We have rephrased the sentence. Please see caption of Supplementary Fig. 12 in the revised version.

- L734-735: I don't understand what is shown on ED Fig8a. Obviously NH and SH TCs are seasonally out of phase so the 6 TCW footprints cannot occur at the same time. Please provide dates, times and names of the 6 TCs shown in this example.

Response: We are sorry for the confusion. In the revised version, we have added a date and name for each storm, to explicitly clarify that they are from different dates. Please see Supplementary Fig. 12 in the revised version.

- ED Table 1: Some Lines in the table are not being used in the manuscript and should therefore be removed.

Response: Thanks. We have removed the unused Lines in this table in the revised version of the manuscript.

REVIEWER COMMENTS

Reviewer #1 (Remarks to the Author):

GENERAL COMMENTS

I appreciate the changes and modifications to the original article. Many of the points raised have been addressed in the revised version. However, my main concern with this article remains: the adequacy of ERA5 to represent TC wave fields. Although the authors have provided more validation of ERA5 Hs against buoys and satellite observations, in my opinion, the validation of TC-induced wave fields (i.e. TCW footprint) remains unaddressed. I cannot recommend the publication of the article until actual proof of the adequacy of the dataset for TC wave fields is provided. I describe below my reasoning in more detail. I hope the authors can address this concern since I believe the paper could provide new insight into the use of ERA5 for TC conditions.

The main hypothesis that the article relies on: wave fields during TC conditions, in an area of 15 degrees from the track, are well represented by ERA5. I have not seen a direct validation of this in the paper yet. I was expecting at least a direct, spatial comparison of the TC wave field with a high resolution model of TC induced waves.

The new validation, as presented in the revised manuscript, only validates the ERA5 overall climatology, as previously done in other publications. I provide some new references and information below. However, these validations correspond to the sea+swell components, and it has been shown by different publications that ERA5 is able to reproduce wave climatology, although it underestimates the extremes. However, this is not the point of this paper, here the paper relies on the characterization in ERA5 of the (sea) waves generated by cyclones locally. Demonstration of this assumption remains unaddressed. Although I would agree that the focus of this paper is on the 'footprint', and not the actual wave heights, it still lacks validation of such footprint and the main hypothesis: can we use ERA5 to understand TC-induced wave climatology and footprint?.

One solution to address this point would be to validate individual cyclones, using high resolution wind-wave modeling of TCs and directly compare with the wave fields from ERA5, which are used in the study (e.g. as in Sup Figure 6, showing temporal evolution in ERA5 and high resolution wave models). Another possibility is using satellite derived data, for ex, does SFig 12 compare against the ERA5 wave fields too? What is SF12-a representing? I would suggest more validation is provided in the main text.

Some other relevant information is provided below:

- (1) A new publication, Fanti et al (2023) has compared ERA5 and WAVEYS with in-situ measurements globally. They find that both reanalysis, including ERA5, consistently underestimate significant wave height, 50-year return period and mean wave period in most coastal locations globally.
- (2) The main concern with the paper is that it relies on ERA5 to defined TC-induced wave fields, at 6-h resolution, but it does not provide direct demonstration that on a 0.5 grid, ERA5 is able to reproduce the

wave field generated by a TC. The validation provided in the revised version is good, but it is performed on Hs, not the wave field (i.e. TC footprint as used in the paper)

(3) Check out Young (2007) for a review of empirical models that provide an estimate of wave footprint and fields during TC conditions. This should inform the article.

(4) I would suggest direct validation against TC specific wave fields, to confirm that the statistical description of the footprint and sea components matches TC conditions.

Additional points re ERA5 data:

- The end year of this article is 2018. Should the study be updated up to the most recent TC season? (at least to 2022)
- Also, note that ERA5 is now available since the year 1940, although the study is limited by the truncation with IBTrACS, and it could be argued that this is a reason to omit the previous period, this should merit some description in the article.

SPECIFIC COMMENTS:

The intro describes the focus on the TC wave footprint, but then Line 144 onwards describes results for the maximum height of TC waves, while other studies have found that extreme Hs are underestimated.

Line 412 . 5 degrees of spatial difference in the track seem relatively large, considering the size of a typical windfield in a TC. See Young (2007), for example.

Line 451. How does the 6-h resolution influence the wave maxima? The hourly time series should have more capacity (temporal resolution) to capture the maxima footprint.

Lines 450-483 seem to describe the ERA5 model set up, but it is very detailed and has no clear connection with the analysis. Again, I lack rigorous validation of the statistical description that would allow the detection of TCW heights.

Line 496 – what is a classic relationship? Describe.

Line 498-502. The detection of waves beyond the 5-degree radius should be explained by waves that have propagated beyond the storm track. However, note that the TCWs are generated by the winds from the storm, and will be still relatively close to the wind field, which will not exceed 5 degrees; the 15-degree limit seems overly large. This is another reason why the paper requires validation with individual wave fields from storms.

Wave fields will be larger in a distance from the track ~ 2-3 times the radius, as the waves are fetch-limited outside the storm reach. See for example figure 5 in Young (2017). The detection of waves generated in previous steps of the storm will depend on whether the storm moves slower than the group velocity or not (waves can outrun the storm).

At least the authors should use a simplified wave model (e.g., parametric wave model from Young, 1988, for example) to provide some insight into the detection algorithm for wave fields. The tests for the 5 and 15 degrees search radius seem poorly based on wave climatology. Furthermore, with such large areas, other effects from high latitude swells may be reflected in the footprint.

Page 22. Clearly indicate that the study only uses the 'sea' components of the spectra for the detection, correct? (the swell components in the 15 degree area are not factored in, please confirm and describe clearly in the manuscript)

Line 533. Does this equation include the wind component only or both?

Page 26. Refer to Fanti et al, in addition to your validation. Discuss conclusions from Fanti et al and implications for this work, especially related to extreme Hs.

Line 608. Does the validation include both swell and sea components? If this is the case, note that the validation is just confirming that ERA5 is able to reproduce the wave climatology, but not the TC induced wave field only.

The problem is that the TC wave field is not validated.

Line 631. Underste

Line 681. The validation with remote sensing is affected by the same limitations that in the case of buoys. It does not provide validation of the TC induced wave field.

References

Young (2007) <https://www.mdpi.com/2073-4433/8/10/194>

Fanti et al (2023) Improved estimates of extreme wave conditions in coastal areas from calibrated global reanalyses <https://www.nature.com/articles/s43247-023-00819-0>

Reviewer #3 (Remarks to the Author):

2nd review of Global increase in tropical cyclone ocean surface waves by J. Shi, X. Feng, R. Toumi, C. Zhang, K. I. Hodges, A. Tao, W. Zhang, and J. Zheng

In this revision, the authors have greatly improved the original submission and addressed most of my comments. Some comments seem to have been overlooked, probably because they were not clear enough, leading to misunderstandings. Whenever significant, I have repeated these comments and reformulated them as necessary. There are some minor issues left, but in my opinion, no more major issues, making the manuscript nearly ready for publication, pending minor revision. Detailed comments follow.

Re : duration. The authors have evidently made efforts to clarify their definition of annual TCW duration. However, I insist that the keyword accumulated is missing in a few key instances

throughout the manuscript. The authors should consider including this word for disambiguation purposes in the following instances : abstract (L100), first appearance in the Results section (L232), first appearance in the Methods section (L531), Suppl. Fig. 4 caption (Suppl. Mat. L99), Suppl. Table 2 caption (Suppl. Mat. L213), Suppl. Table 4 caption (Suppl. Mat. L233).

Re : NH values smaller on Fig. 1 and larger on Fig. 2. From the response provided by the authors, I understand that the contribution of swell is considered larger in the SH and smaller in the NH compared to the contribution of local wind sea, based on arguments related to basin geometry. Because of swell propagation away from the source, the swell contribution is spatially smoother than the wind sea contribution, allowing perhaps for less spatio-temporal variability than the more variable wind-induced large-wave patch in the vicinity of the eyewall. Thus, rotated Lagrangian composite averages may reach larger wave height values in the SH. In contrast, more intense TCs in the NH supposedly drive larger maximum surface waves on average. If such understanding is correct, then why is the assumed larger NH TC maximum wind speed not supported by values from individual basins on Suppl. Fig. 1 (red bars)? In fact, Suppl. Discussion points out weaker winds and smaller waves in the NI and EP but not in the SH.

L66-67 : in response to a previous comment, the authors added the keyword future, which I think reads better. However, they still refer to the same published works, despite my cautionary note. Perhaps I did not express myself clearly enough, so I will try to rephrase (sorry for insisting) :
- Ref. 32 is not about future projected global warming. If the authors think this reference is really necessary here, I suggest them to rephrase the sentence as There have been theoretical and modelling studies examining [...]. Otherwise consider removing this reference.
- Ref. 34 should be removed as it does not deal with TC waves.

L77 : see the Methods section for details is too vague, which section do the authors refer to ? Section 6, L742-752 seem to partly address this issue, in that it details the uncertainty in extreme waves from reanalysis data, but refers to section 3 for details regarding why relative trends in TC wave metrics are less sensitive to extreme wave height values than absolute trends. However, while section 3 presents the methods used to compute TC wave metrics, there is nothing regarding the advantages of relative trends, which I think is lacking from the paper.

L90, 92, 95 : as already commented on the original manuscript, the uncertainty is large (~80 % of the mean) and the authors should comment on that. I feel it is not enough to show the error estimates and state how they are computed without commenting the wide error bars.

L122-123 : is ref. 8 (Timmermans et al. 2017) the correct reference for this statement ?

L195 : for different thresholds of TC wave heights.

L210 : please rephrase as smaller values of climatology of the area. The WNP value (1.4 10⁶ km²) is smaller than the SI (and SP) values but significantly larger compared to other basins.

L210-213 : I do not quite follow your interpretation. Could you please expand/clarify ?

Fig. 3 : the value of the 40-year mean should be indicated on each subplot. I already made that comment on the first round but it was apparently misunderstood.

L249 and 252 : following up on a previous comment, the authors have corrected the relative contributions to the energy trend, but the reader is left to blindly believe the authors as I could not find any table indicating the associated absolute trends (and error bars, significance levels), which I think would make a useful addition to supplementary material.

Fig. 4 : converting duration units to days. In a previous comment, I suggest the authors divide the duration values (expressed as a number of 6-hourly time steps) by four. However, from visual comparison of the duration scales between the original and revised manuscript, I have the feeling that a factor larger than four was used. For example, on panel a, the 40-year mean of TC wave duration was roughly 1800 time steps in the original manuscript, but clearly lower than 450 days in this revision. Although I may be mistaken by optical distortion made easier by the coarse sampling of the y-axis on the figures, I request that you please carefully check your computations.

L337 : did you really mean the maxima of spatial average, or rather the (spatial) maxima of the composite averaged field ? In the authors' formulation, taking the maxima of a spatially-averaged field means computing the maximum of a time series resulting from spatial averaging, which I doubt is the case here.

L347-349 : again (see related comment on the original manuscript), the authors should also mention that they calculated the trend in the lifetime max TC intensity since they mention 'increases of already very high winds' in the previous sentence.

L354-357 : I am afraid the authors have not properly addressed my previous related comment. From my understanding, the interpretation put forward by the authors (weak trends in lifetime maximum TC wave heights are due to saturating drag coefficient for strong winds) implicitly assumes that lifetime maximum winds are increasing. Such assumption is wrong in several basins, and in the northern hemisphere as a whole. I pointed out that trends in lifetime max wave heights tend to correlate with trends in lifetime max winds. Therefore, I suspect that lifetime max wave heights do not increase simply because lifetime max winds do not increase either (unlike the respective 6-hourly values). Not because of the explanation put forward (at least, not at first order).

L384-386 : sorry to insist. Ref. 48 should be moved to L385, between the parentheses right after ERA5. The extra references added by the authors (49-50) are fine.

L401 and 403 : again, please remove any mention to minimum mean sea level pressure, because SLP data does not appear to be used in the paper.

L520 : as I already stated previously, ref. 48 should be removed here, it is not relevant for describing the method used to compute ERA5 wave spectra. Please see related comment on the first round of reviews.

L526 : the authors must have missed this occurrence of wind-wave that rather means wind-sea (other instances look fine to me). Therefore ewindwave should also be ewindsea in Eq(8).

L537-538 : should be $TCN(y)$ in the formula (not just TCN). Also, rephrase as TCN is the annual TC count (or number). TCN is non-dimensional, whereas frequency is in the units of year⁻¹.

L624-625 : please cite Suppl. Table 3 when referring to sample size.

L644 : please remove commonly.

L674-675 : slightly rephrase as We only used [...] maximum intensity because waves [...]. It has already been stated that data are being used at the time of lifetime maximum intensity (L668-671).

L797-801 : since you also included TC wave duration in Suppl. Table 4, the corresponding relative trend values with (4.9 ± 5.4 %/decade**) and without ENSO (5.3 ± 5.2 %/decade*) should also be included here (note the more robust trend after removing ENSO).

L806-808 : the definition of the yearly ENSO index is clarified, but this should be moved back to the original location, right after the index was first mentioned (L790).

Suppl. Mat. L39-40 : should read ≥ 16.2 m/s and ≤ 15.4 m/s.

Suppl. Mat. L40-42 : should read 2.9-3.5 m. The end of this sentence is unclear to me.

Suppl. Mat. L48-50 : both high TC frequency and large TCW area probably contribute to the high TCW energy in both basins. SI (resp. WNP) has the second highest TC frequency (resp. third largest TCW area).

Suppl. Mat. L115-116 : please remove this last sentence, it is unnecessary (I find that L109-111 are self-explanatory as they appear now), incomplete (no mention of winds) and has a typo (maximum maximum).

Suppl. Fig. 7b : an orange (resp. a blue) bar seems to be missing in the beginning (resp. end) of the plotted time range.

Suppl. Mat. L137 : please remove (shaded area), which has no longer any meaning in the histogram plot.

Suppl. Mat. L233 : replace TC waves by TC wave height.

Responds to the reviewers' comments:

Reviewer #1:

GENERAL COMMENTS

I appreciate the changes and modifications to the original article. Many of the points raised have been addressed in the revised version. However, my main concern with this article remains: the adequacy of ERA5 to represent TC wave fields. Although the authors have provided more validation of ERA5 Hs against buoys and satellite observations, in my opinion, the validation of TC-induced wave fields (i.e. TCW footprint) remains unaddressed. I cannot recommend the publication of the article until actual proof of the adequacy of the dataset for TC wave fields is provided. I describe below my reasoning in more detail. I hope the authors can address this concern since I believe the paper could provide new insight into the use of ERA5 for TC conditions.

The main hypothesis that the article relies on: wave fields during TC conditions, in an area of 15 degrees from the track, are well represented by ERA5. I have not seen a direct validation of this in the paper yet. I was expecting at least a direct, spatial comparison of the TC wave field with a high resolution model of TC induced waves.

The new validation, as presented in the revised manuscript, only validates the ERA5 overall climatology, as previously done in other publications. I provide some new references and information below. However, these validations correspond to the sea+swell components, and it has been shown by different publications that ERA5 is able to reproduce wave climatology, although it underestimates the extremes. However, this is not the point of this paper, here the paper relies on the characterization in ERA5 of the (sea) waves generated by cyclones locally. Demonstration of this assumption remains unaddressed. Although I would agree that the focus of this paper is on the 'footprint', and not the actual wave heights, it still lacks validation of such footprint and the main hypothesis: can we use ERA5 to understand TC-induced wave climatology and footprint?.

One solution to address this point would be to validate individual cyclones, using high resolution wind-wave modeling of TCs and directly compare with the wave fields from ERA5, which are used in the study (e.g. as in Sup Figure 6, showing temporal evolution in ERA5 and high resolution wave models). Another possibility is using satellite derived data, for ex, does SF12 compare against the ERA5 wave fields too? What is SF12-a representing? I would suggest more validation is provided in the main text.

Response: We thank the reviewer for these comments which are very useful to improve the quality of our paper.

- In the new version of our manuscript, we have added a new subsection in the validation part (Lines 689-736), which is related to the validation of the TC wave footprint. We used a parametric wave model proposed by Young (1988) to calculate TC wave field, and extracted the TC wave footprint based on $H_s \geq 2.5$ m. The comparison between ERA5 and the parametric model is shown in Supplementary Fig. 14a and b. For the instantaneous wave field, although the shapes of the TCW area are different, the values of the TCW area are very similar, which are 1.79×10^6 and 1.72×10^6 km² for the ERA5 and the parametric model, respectively. We also compared the mean TCW area between the parametric model and ERA5 over the period 1979–2022. There are 809 TC events selected based on the criteria in ref (Young and Vinoth, 2013). The scatter plots of the mean TCW area from the parametric model and ERA5 are shown in Supplementary Fig. 14b. The ERA5 slightly overestimates the area with the *MRE* of 4% and the *RMSE* of 9×10^5 km². We also compared the 6-hourly maximum height from the parametric model and ERA5 in Supplementary Fig. 14c. Compared with the parametric model, ERA5 tends to underestimate the low-to-medium values of Hs ($H_s < 6$ m), and tends to overestimate the medium-to-high values of Hs ($H_s > 6$ m). But the error is relatively small with *MRE* of 3% and *RMSE* of 0.9m. We also compared the

composite mean the height of TC wave footprint for the selected 809 TC events in the NH and SH in Supplementary Fig. 15 and Supplementary Fig. 16. An increase of H_s values and area between the two epochs is seen in both datasets. These agreements are better in the Northern Hemisphere than in the Southern Hemisphere, related to more samples in the Northern Hemisphere.

- As for the underestimation of the extreme wave by ERA5, Fanti et al (2023) and many other studies have compared ERA5 with coastal buoys. The accuracy of ERA5 is lower in the coastal area since the complex wave shoaling, diffraction and refraction processes cannot be well solved by the spectral wave model with relatively coarse resolution. However, in this study, the maximum wave height is mainly located in the deep water. we re-calculated the trends of the TCW area and maximum wave height after excluding TCW values in water depths less than 100m, and the trend value remains essentially almost unchanged (Supplementary Tables 1 and 2). For example, the global relative trend in maximum wave height changes from $3.2 \pm 1.3\%/decade$ to $3.3 \pm 1.3\%/decade$, and the global relative trend in the area changes from $5.7 \pm 3.8\%/decade$ to $5.7 \pm 3.9\%/decade$.
- With respect to “using high-resolution wind-wave modeling of TCs and directly comparing with the wave fields from ERA5”, we think there are many uncertainties in the simulations and observations. For example, the differences in consideration of source terms could significantly impact the results and it would be challenging to tune an optimal choice for these parameters for each ocean basin. Relatedly, what resolution is sufficient to simulate extreme waves in all basins for different categories of TCs? How can we validate such high-resolution simulations? Nowadays, there is still limited access to reliable in-situ observations of extreme waves. We want to emphasize that in this paper we are not aiming to best simulate the most extreme waves. Instead, we investigate the long-term changes in TC waves defined by moderate wave height ($H_s \geq 2.5m$).
- With respect to “use satellite data to validate the ERA5 data”, we have shown the comparisons in Sfig. 12a. In the figure, We use the satellite observations in the TCW first guess circle (15-degree geodesic circle) when ERA5 TCs are at their lifetime maximum intensity (Supplementary Fig. 12a). Then, we take Jason-2 along-track H_s (footprint about 3-4km wide) within the TCW first guess area, and then bin them into the ERA5 grid (0.5×0.5 deg). The color along the satellite track shows the H_s observed from Jason-2 satellite, while the other color within the 15 degree circles shows the results from ERA5. Similar color near the satellite track means the ERA5 data agree well with the observations. Since the satellite altimeter can only provide H_s along the satellite tracks, it is difficult to use satellite data to validate the wave field.

References

- Young, I. R. (1988). Parametric hurricane wave prediction model. *Journal of Waterway, Port, Coastal, and Ocean Engineering*, 114(5), 637-652.
- Young, I. R., & Vinoth, J. (2013). An “extended fetch” model for the spatial distribution of tropical cyclone wind-waves as observed by altimeter. *Ocean Engineering*, 70, 14-24.
- Fanti, V., Ferreira, Ó., Kümmerer, V., & Loureiro, C. (2023). Improved estimates of extreme wave conditions in coastal areas from calibrated global reanalyses. *Commun. Earth Environ.* 4, 151.
- Li, R., Wu, K., Zhang, W., Dong, X., Lv, L., Li, S., ... & Babanin, A. V. (2023). Analysis of the 20-Year Variability of Ocean Wave Hazards in the Northwest Pacific. *Remote Sensing*, 15(11), 2768.

Some other relevant information is provided below:

- (1) A new publication, Fanti et al (2023) has compared ERA5 and WAWERYYS with in-situ measurements globally. They find that both reanalysis, including ERA5, consistently

underestimate significant wave height, 50-year return period and mean wave period in most coastal locations globally.

Response: Fanti et al (2023) compared ERA5 with 326 coastal buoys, and found the ERA5 and WAVERYS underestimated H_s around the world. We have cited this article and added an illustration of the differences between our validation and Fanti et al (2023) (Lines 795-808). The accuracy of ERA5 is lower in the coastal area since the complex wave shoaling, diffraction and refraction processes cannot be well solved by the spectral wave model with relatively coarse resolution. However, in this study, the maximum wave height is mainly located in the deep water. We have calculated the trends of maximum wave height without values in water depth less than 100m, and the trend value remains essentially almost unchanged with global trend values changing from $3.2 \pm 1.3\%$ /decade to $3.4 \pm 1.3\%$ /decade for the maximum wave height, and from $5.7 \pm 3.8\%$ /decade to $5.7 \pm 3.9\%$ /decade for the TCW area. A recent study by Li et al., (2023) also compared ERA5 extreme wave height with satellite data in the WNP. They found ERA5 can well capture the extreme waves during TC in deep water. Compared with satellite data, the ERA5 data exhibits better consistency.

Li, J., Zhang, S., Liu, Q., Yu, X., & Zhang, Z. (2023). Design and Evaluation of an Efficient High-Precision Ocean Surface Wave Model with a Multiscale Grid System (MSG_Wav1.0). *Geosci. Model Dev. Discuss.* 1, 1-34.

- (2) The main concern with the paper is that it relies on ERA5 to defined TC-induced wave fields, at 6-h resolution, but it does not provide direct demonstration that on a 0.5 grid, ERA5 is able to reproduce the wave field generated by a TC. The validation provided in the revised version is good, but it is performed on H_s , not the wave field (i.e. TC footprint as used in the paper)

Response: Thanks for your suggestion. In the new version of our manuscript, we have added a new subsection in the validation part (Lines 689-736), which is related to the validation of the TC wave footprint. As the reviewer suggested, we used a parametric wave model proposed by Young (1988) to calculate the TC H_s , and extracted the TC wave footprint based on $H_s \geq 2.5$ m. The comparison between ERA5 and the parametric model is shown in Supplementary Fig. 14a and b. We hope this additional analysis will address the reviewer's comment.

- (3) Check out Young (2007) for a review of empirical models that provide an estimate of wave footprint and fields during TC conditions. This should inform the article.

Response: Thanks for your suggestion. We have used the parametric model proposed by Young (1988) to compute the wave field and compared with wave field from ERA5. Please see Lines 689-736.

- (4) I would suggest direct validation against TC specific wave fields, to confirm that the statistical description of the footprint and sea components matches TC conditions.

Response: Thanks for your suggestion. In the new version of our manuscript, as the reviewer suggested, we have added a new subsection in the validation part (Lines 689-736), which is related to the validation of the TC wave footprint. We compared the instantaneous wave field, TC wave area and TC wave height from the parametric model and ERA5.

Additional points re ERA5 data:

- The end year of this article is 2018. Should the study be updated up to the most recent TC season? (at least to 2022)

Response: Thanks for your advices. We have updated to 2022 in the revised manuscript.

- Also, note that ERA5 is now available since the year 1940, although the study is limited by the

truncation with IBTrACS, and it could be argued that this is a reason to omit the previous period, this should merit some description in the article.

Response: Thanks for your advices. We have added description in the article. Please see Lines 425-427.

SPECIFIC COMMENTS:

The intro describes the focus on the TC wave footprint, but then Line 144 onwards describes results for the maximum height of TC waves, while other studies have found that extreme Hs are underestimated.

Response: Thank you for pointing this out. This is the *relative* trend in wave height. In this paper, we looked at various metrics for TC waves (e.g., height, area, duration, and energy) both globally and at basin scale. We highlight the value of this study in the relative trends of TC waves, which relies less on how well the ERA5 simulates the extreme wave height. We clarified this comment in Lines 782-786.

Line 412 . 5 degrees of spatial difference in the track seem relatively large, considering the size of a typical windfield in a TC. See Young (2007), for example.

Response: The spatial difference is up to a 5° between ERA5 and IBTrACS. We have calculated the mean difference from 1979 to 2022, the mean difference value is 0.4°.

Line 451. How does the 6-h resolution influence the wave maxima? The hourly time series should have more capacity (temporal resolution) to capture the maxima footprint.

Response: Thank you for pointing this out. The reason we used a 6-h resolution wave is that the calculation of TCW footprint is matched with IBTrACS tracks. The hourly track locations and TC intensity is interpolated from the 6-h TC tracks and TC intensity. So the maximum TC intensity can be captured by the 6-h tracks. That means our results include TCW area with maximum TC intensity. Furthermore, the TCW area used in this study is an annual mean value, which is not sensitive to the temporal resolution.

Lines 450-483 seem to describe the ERA5 model set up, but it is very detailed and has no clear connection with the analysis. Again, I lack rigorous validation of the statistical description that would allow the detection of TCW heights.

Response: Thanks for your advices. We have rephrase the sentences and deleted the detailed setup description of ERA5 wave model. We briefly describe wave spectra in the ERA5 wave data because they are in the centre of defining the TC wave energy in the subsection 3. As for the defination of Hs, we have shown the equations of Hs by Eqs2-3. Please see Lines 458-467.

Line 496 – what is a classic relationship? Describe.

Response: We have used both the wind-wave relationship and the parametric wave model from Young, (1988) to estimate Hs induced by wind speed of 9m/s. Please see Lines 479-482.

Line 498-502. The detection of waves beyond the 5-degree radius should be explained by waves that have propagated beyond the storm track. However, note that the TCWs are generated by the winds from the storm, and will be still relatively close to the wind field, which will not exceed 5 degrees; the 15-degree limit seems overly large. This is another reason why the paper requires validation with individual wave fields from storms.

Response: We thank the reviewer for these comments. In this revised version of the manuscript, we have added validation of wave field in Sfig. 14-16 and Lines 689-736.

Although we used a 15-degree geodesic circle to get the TCW area, only the waves at the contiguous grid points where $H_s \geq 2.5\text{m}$ are defined as the TCW footprint. That means the TCW area depends on the contiguous $H_s \geq 2.5\text{m}$ area. If the TC is weak, the area can be much smaller than the area of the circle. If there is a very intense TC, the TCW area can be included in the circle. The distribution of the area of the 6-hourly TCW is shown in Supplementary Fig. 6b. The comparison of different radius shows that 5 degree is too small to include larger TCW area induced by intensive TC. The distribution of the footprint area becomes stable when the radius is larger than 15 degree.

Wave fields will be larger in a distance from the track $\sim 2\text{-}3$ times the radius, as the waves are fetch-limited outside the storm reach. See for example figure 5 in Young (2017). The detection of waves generated in previous steps of the storm will depend on whether the storm moves slower than the group velocity or not (waves can outrun the storm).

Response: Yes, we agree that the wave fields induced by TC are asymmetrical, which are affected by TC translation speed and also waves generated in previous steps. Thus, it is difficult to extract a wave field with only influence of TC wind. In this study, we mainly focus on the mixed waves, and also analyze the swell and windsea components.

At least the authors should use a simplified wave model (e.g., parametric wave model from Young, 1988, for example) to provide some insight into the detection algorithm for wave fields. The tests for the 5 and 15 degrees search radius seem poorly based on wave climatology. Furthermore, with such large areas, other effects from high latitude swells may be reflected in the footprint.

Response: We thank the reviewer for these comments. In the revised version of manuscript, we have used both the wind-wave relationship and the parametric wave model from Young, (1988) to estimate H_s induced by wind speed of 9m/s . Please see lines 479-482.

To minimise the effects from high latitude swells driven by extratropical cyclones and strong westlies, we restrict the 6-hourly track points within 40°N and 40°S .

Page 22. Clearly indicate that the study only uses the ‘sea’ components of the spectra for the detection, correct? (the swell components in the 15 degree area are not factored in, please confirm and describe clearly in the manuscript)

Response: We are sorry for the confusion. In this study, we calculated mixed wave energy, and also both swell and windsea components. We have rephrased the subsection of TCW energy. Please see Lines 511-515.

Line 533. Does this equation include the wind component only or both?

Response: This equation includes energy of mixed waves (both swell and windsea components). We have added illustration in Lines 517-519.

Page 26. Refer to Fanti et al, in addition to your validation. Discuss conclusions from Fanti et al and implications for this work, especially related to extreme H_s .

Response: We have carefully read Fanti et al., (2023) and cited it in our validation part. Please see Lines 795-808.

Line 608. Does the validation include both swell and sea components? If this is the case, note that the validation is just confirming that ERA5 is able to reproduce the wave climatology, but not the TC induced wave field only.
The problem is that the TC wave field is not validated.

Response: Yes, the validation includes both swell and sea components. It is difficult to distinguish swell and windsea components from both buoy and satellite altimeter data. So we compared ERA5 and observation data with mixed waves. For the validation of TC wave field, there is no such kind of data from current observations. We compared ERA5 with a parametric wave model proposed by Young (1988). We have added validation of wave field in Sfig. 14-16 and Lines 689-736.

Line 631. Underste

Response: We are sorry for the error. This has been corrected in the revised manuscript.

Line 681. The validation with remote sensing is affected by the same limitations that in the case of buoys. It does not provide validation of the TC induced wave field.

Response: We thank the reviewer for these comments. We compared ERA5 with a parametric wave model proposed by Young (1988). We have added validation of wave field in Sfig. 14-16 and Lines 689-736.

Reviewer #2:

In this revision, the authors have greatly improved the original submission and addressed most of my comments. Some comments seem to have been overlooked, probably because they were not clear enough, leading to misunderstandings. Whenever significant, I have repeated these comments and reformulated them as necessary. There are some minor issues left, but in my opinion, no more major issues, making the manuscript nearly ready for publication, pending minor revision. Detailed comments follow.

Response: Thank you for your positive comments and valuable suggestions to improve the quality of our manuscript. According to your suggestions, we have made extensive corrections to our previous draft, the detailed corrections are listed below.

Re : duration. The authors have evidently made efforts to clarify their definition of annual TCW duration. However, I insist that the keyword *accumulated* is missing in a few key instances throughout the manuscript. The authors should consider including this word for disambiguation purposes in the following instances : abstract (L100), first appearance in the Results section (L232), first appearance in the Methods section (L531), Suppl. Fig. 4 caption (Suppl. Mat. L99), Suppl. Table 2 caption (Suppl. Mat. L213), Suppl. Table 4 caption (Suppl. Mat. L233).

Response: Thanks for your suggestions. We have added the keyword *accumulated* in the above sentences.

Re : NH values smaller on Fig. 1 and larger on Fig. 2. From the response provided by the authors, I understand that the contribution of swell is considered larger in the SH and smaller in the NH compared to the contribution of local wind sea, based on arguments related to basin geometry. Because of swell propagation away from the source, the swell contribution is spatially smoother than the wind sea contribution, allowing perhaps for less spatio-temporal variability than the more variable wind-induced large-wave patch in the vicinity of the eyewall. Thus, rotated Lagrangian composite averages may reach larger wave height values in the SH. In contrast, more intense TCs in the NH supposedly drive larger maximum surface waves on average. If such understanding is correct, then why is the assumed larger NH TC maximum wind speed not supported by values from individual basins on Suppl. Fig. 1 (red bars)? In fact, Suppl. Discussion points out weaker winds and smaller waves in the NI and EP but not in the SH.

Response: Thanks for this useful comment. We have carefully checked the values in Fig. 1,2 and Suppl. Fig. 1. In Fig. 2, the maximum Hs of NH looks larger than that of SH, that is the reason why we assumed more intense TCs in the NH. By calculating the 44-year mean value of the maximum Hs, the mean Hs values in NH and SH are 4.86m and 4.59m. The 44-year mean values of TC maximum wind speed in NH and SH are 17.1m and 17.2m. That means the average intensity of TCs and TC induced maximum waves in the two hemispheres are similar.

L66-67 : in response to a previous comment, the authors added the keyword *future*, which I think reads better. However, they still refer to the same published works, despite my cautionary note. Perhaps I did not express myself clearly enough, so I will try to rephrase (sorry for insisting) :
- Ref. 32 is not about future projected global warming. If the authors think this reference is really necessary here, I suggest them to rephrase the sentence as *There have been theoretical and modelling studies examining [...]*. Otherwise consider removing this reference.
- Ref. 34 should be removed as it does not deal with TC waves.

Response: Thanks for your suggestions. We have deleted these two refs.

L77 : *see the Methods section for details* is too vague, which section do the authors refer to ? Section 6, L742-752 seem to partly address this issue, in that it details the uncertainty in extreme waves from reanalysis data, but refers to section 3 for details regarding why relative trends in TC wave metrics are less sensitive to extreme wave height values than absolute trends. However,

while section 3 presents the methods used to compute TC wave metrics, there is nothing regarding the advantages of relative trends, which I think is lacking from the paper.

Response: Thanks. We have included description of relative trend advantages in Lines 782-786. Because there are too many subsections in Methods, we have specified the heading of the subsection. Please see Lines 83,141...

L90, 92, 95 : as already commented on the original manuscript, the uncertainty is large (~80 % of the mean) and the authors should comment on that. I feel it is not enough to show the error estimates and state how they are computed without commenting the wide error bars.

Response: Thanks. We have mentioned this uncertainty in Lines 313-316.

L122-123 : is ref. 8 (Timmermans et al. 2017) the correct reference for this statement ?

Response: Thank you for pointing this out. In Timmermans et al. 2017, they only stated why the effective fetch in the SH is larger, thus, we remove this ref to Line 114.

L195 : *for different thresholds of TC wave heights.*

Response: Thanks. The sentence has been rephrased. Please see Lines 186-187.

L210 : please rephrase as *smaller values of climatology of the area*. The WNP value (1.4 106 km²) is smaller than the SI (and SP) values but significantly larger compared to other basins.

Response: Thanks. The sentence has been rephrased. Please see Line 201.

L210-213 : I do not quite follow your interpretation. Could you please expand/clarify ?

Response: Thanks. We have rephrased the interpretation. Please see Lines 200-204.

Fig. 3 : the value of the 40-year mean should be indicated on each subplot. I already made that comment on the first round but it was apparently misunderstood.

Response: Sorry for the misunderstanding. We have added the values in the new revised version.

L249 and 252 : following up on a previous comment, the authors have corrected the relative contributions to the energy trend, but the reader is left to blindly believe the authors as I could not find any table indicating the associated absolute trends (and error bars, significance levels), which I think would make a useful addition to supplementary material.

Response: Thanks for your suggestions. The trend values for the TCW area and duration can be found in Supplementary Table 2. We have included the trend value of mean wave height in Supplementary Table 1.

Fig. 4 : converting duration units to days. In a previous comment, I suggest the authors divide the duration values (expressed as a number of 6-hourly time steps) by four. However, from visual comparison of the duration scales between the original and revised manuscript, I have the feeling that a factor larger than four was used. For example, on panel a, the 40-year mean of TC wave duration was roughly 1800 time steps in the original manuscript, but clearly lower than 450 days in this revision. Although I may be mistaken by optical distortion made easier by the coarse sampling of the y-axis on the figures, I request that you please carefully check your computations.

Response: Sorry for the error. During the last revision, we converting duration units to hours firstly. But finally decided to use units of days. In the converting process, we wrongly used a factor of 6. The figure has been corrected in the new revised version.

L337 : did you really mean the *maxima of spatial average*, or rather the (spatial) maxima of the composite averaged field ? In the authors' formulation, taking the maxima of a spatially-averaged field means computing the maximum of a time series resulting from spatial averaging, which I doubt is the case here.

Response: Sorry for the confusion. We means the maxima of the composite averaged field here.

L347-349 : again (see related comment on the original manuscript), the authors should also mention that they calculated the trend in the lifetime max TC **intensity** since they mention 'increases of already very high **winds**' in the previous sentence.

Response: Thanks for your suggestions. We have deleted the previous sentence.

L354-357 : I am afraid the authors have not properly addressed my previous related comment. From my understanding, the interpretation put forward by the authors (weak trends in lifetime maximum TC wave heights are due to saturating drag coefficient for strong winds) implicitly assumes that lifetime maximum winds are increasing. Such assumption is wrong in several basins, and in the northern hemisphere as a whole. I pointed out that trends in lifetime max wave heights tend to correlate with trends in lifetime max winds. Therefore, I suspect that lifetime max wave heights do not increase simply because lifetime max winds do not increase either (unlike the respective 6- hourly values). Not because of the explanation put forward (at least, not at first order).

Response: Sorry for the misunderstanding. We have added illustration on the relationship between the lifetime maximum winds and lifetime maximum wave height. Please see Lines 341-347.

L384-386 : sorry to insist. Ref. 48 should be moved to L385, between the parentheses right after ERA5. The extra references added by the authors (49-50) are fine.

Response: Thanks for your suggestions. We have move Ref. 48 to Line 373.

L401 and 403 : again, please remove any mention to *minimum mean sea level pressure*, because SLP data does not appear to be used in the paper.

Response: Thanks for your suggestions. We have rephrased the sentence.

L520 : as I already stated previously, ref. 48 should be removed here, it is not relevant for describing the method used to compute ERA5 wave spectra. Please see related comment on the first round of reviews.

Response: Thanks for your suggestions. We have deleted the ref here.

L526 : the authors must have missed this occurrence of *wind-wave* that rather means *wind-sea* (other instances look fine to me). Therefore *windwave* should also be *windsea* in Eq(8).

Response: Sorry for the error. We have replaced windwave by windsea in this line and also in Eq(8).

L537-538 : should be $TCN(y)$ in the formula (not just TCN). Also, rephrase as *TCN is the annual TC count* (or *number*). TCN is non-dimensional, whereas frequency is in the units of year⁻¹ .

Response: Sorry for the error. We have used $TCN(y)$ in the formula.

L624-625 : please cite Suppl. Table 3 when referring to sample size.

Response: Thanks for your suggestions. We have included the citation.

L644 : please remove *commonly*.

Response: Thanks for your suggestions. We have deleted the word.

L674-675 : slightly rephrase as *We only used [...] maximum intensity because waves [...]*. It has already been stated that data are being used at the time of lifetime maximum intensity (L668-671).

Response: Thanks for your suggestions. We have rephrased the sentence.

L797-801 : since you also included TC wave duration in Suppl. Table 4, the corresponding relative trend values with (4.9 ± 5.4 %/decade**) and without ENSO (5.3 ± 5.2 %/decade*) should also be included here (note the more robust trend after removing ENSO).

Response: Thanks for your suggestions. We have included results of TC wave duration.

L806-808 : the definition of the yearly ENSO index is clarified, but this should be moved back to the original location, right after the index was first mentioned (L790).

Response: Thanks for your suggestions. We have moved the sentences back to the original location

Suppl. Mat. L39-40 : should read ≥ 16.2 m/s and ≤ 15.4 m/s.

Response: Thanks for your suggestions. We have corrected the errors.

Suppl. Mat. L40-42 : should read 2.9-3.5 m. The end of this sentence is unclear to me.

Response: Thanks for your suggestions. We have corrected the errors, and deleted the last sentence.

Suppl. Mat. L48-50 : both high TC frequency and large TCW area probably contribute to the high TCW energy in both basins. SI (resp. WNP) has the second highest TC frequency (resp. third largest TCW area).

Response: Thanks for your useful comments. We have rephrased the sentence. Please see Suppl. Mat. L47-48.

Suppl. Mat. L115-116 : please remove this last sentence, it is unnecessary (I find that L109-111 are self-explanatory as they appear now), incomplete (no mention of winds) and has a typo (*maximum maximum*).

Response: Sorry for the error. We have deleted the sentence.

Suppl. Fig. 7b : an orange (resp. a blue) bar seems to be missing in the beginning (resp. end) of the plotted time range.

Response: Sorry for the error. We have replotted the figure.

Suppl. Mat. L137 : please remove (*shaded area*), which has no longer any meaning in the histogram plot.

Response: Sorry for the error. We have removed the unnecessary words.

Suppl. Mat. L233 : replace *TC waves* by *TC wave height*.

Response: Thanks for your suggestion. We have replaced *TC waves* by *TC wave height*.

REVIEWERS' COMMENTS

Reviewer #1 (Remarks to the Author):

Line 23. Indicate that this is from ERA5.

Line 91-92, line 137. Include here a connection with similar results to other increases detected in WE in ocean basins from extratropical activity too (e.g. reference 23)

Line 450 onwards, and line 513. It is not clear to me if you use the Hs from ERA5 or reconstruct the spectra from the different partitions. Make clear that you use the spectra partitions to reconstruct the spectra or that you use the ERA5 Hm0. Which one is correct?

Also, you calculate swell first, and the wind component as the difference with the overall Hm0? This requires a clearer description.

Line 456. Correct 'don't'

Discussion and section 6. The validation should be done against specific TC induced waves. This may require a direct comparison with a TC wave numerical model. Since this does not exist yet, I suggest you include this point in the discussion, in support of the satellite derived measurements and a parametric TC wave model, which both include limitations to represent the tail of the distribution for TC induce wave fields.

The main caveat in the article is the representation of TC fields in the ERA5 dataset. I'd recommend emphasizing this in the abstract, intro and discussion. As the authors state:

...we are not aiming to best simulate the most extreme waves. Instead, we investigate the long-term changes in TC waves defined by moderate wave height ($H_s \geq 2.5\text{m}$).

Reviewer #3 (Remarks to the Author):

I am satisfied with the corrections performed by the authors, with one small exception. Lines 386-391 (Methods section, 1. TC tracks), "minimum mean sea level pressure" is quoted twice (L388 & L390), alongside "maximum surface wind speed". Whereas MSW are used extensively throughout the paper, min SLP is never mentioned. What is the point of describing how SLP metrics are being computed if this is not used anywhere? I suspect this is a residual from a former version of the manuscript but I do not see how this could be relevant in the present version. Please remove both mentions to min MSLP. I had already made this comment but it has been overlooked and/or misunderstood.

Provided this minor edit is made, I recommend that the Editor accepts this manuscript for publication.

Responds to the reviewers' comments:

Reviewer #1:

Line 23. Indicate that this is from ERA5.

Response: We thank the reviewer for these comments which are very useful to improve the quality of our paper. We have explicitly mentioned that the trend is obtained from ERA5 in Line 22.

Line 91-92, line 137. Include here a connection with similar results to other increases detected in WE in ocean basins from extratropical activity too (e.g. reference 23)

Response: Thanks for your suggestion. In the introduction part, we have mentioned that global wave energy has increased by about 4%/decade (Line 47). Also, we have shown the differences among TC waves, overall ocean waves and high-latitude extreme waves (Lines 48-51).

Line 450 onwards, and line 513. It is not clear to me if you use the H_s from ERA5 or reconstruct the spectra from the different partitions. Make clear that you use the spectra partitions to reconstruct the spectra or that you use the ERA5 H_{m0} . Which one is correct?

Response: We apologize for any confusion caused by the unclear expression. To clarify, we have specified that the significant wave height of mixed (overall) waves, swell, or wind sea waves was obtained from ERA5 (See lines 402-403).

Also, you calculate swell first, and the wind component as the difference with the overall H_{m0} ? This requires a clearer description.

Response: Thanks for your suggestion. We have added the description of calculating wind sea component (Lines 479-481).

Line 456. Correct 'don't'

Response: We apologize for the error, and we want to clarify that it has been corrected in the new version of the manuscript.

Discussion and section 6. The validation should be done against specific TC induced waves. This may require a direct comparison with a TC wave numerical model. Since this does not exist yet, I suggest you include this point in the discussion, in support of the satellite derived measurements and a parametric TC wave model, which both include limitations to represent the tail of the distribution for TC induced wave fields.

Response: We are grateful for your valuable suggestion. We have conducted a comparison of ERA5 wave height with a parametric wave model and satellite wave data to demonstrate the accuracy of ERA5 in capturing the distribution of tropical cyclone waves. It's worth noting that the wave data in ERA5 is also calculated using a wave model. Therefore, the comparison between ERA5 wave data and other results from tropical cyclone wave numerical models primarily showcases the impact of grid resolution. These comparisons and the discussion of resolution effects can be found in Lines 755-759.

The main caveat in the article is the representation of TC fields in the ERA5 dataset. I'd recommend emphasizing this in the abstract, intro and discussion. As the authors state: ...we are not aiming to best simulate the most extreme waves. Instead, we investigate the long-term changes in TC waves defined by moderate wave height ($H_s \geq 2.5\text{m}$).

Response: We appreciate your helpful suggestion. To address this concern, we have illustrated the reason of using relative change in the introduction part (Lines 69-71). We also highlight the value

of this study in the relative trends of TC waves, which relies less on how well the ERA5 simulates the extreme wave height. This clarification can be found in Lines 720-737.

Reviewer #3:

I am satisfied with the corrections performed by the authors, with one small exception. Lines 386-391 (Methods section, 1. TC tracks), "minimum mean sea level pressure" is quoted twice (L388 & L390), alongside "maximum surface wind speed". Whereas MSW are used extensively throughout the paper, min SLP is never mentioned. What is the point of describing how SLP metrics are being computed if this is not used anywhere? I suspect this is a residual from a former version of the manuscript but I do not see how this could be relevant in the present version. Please remove both mentions to min MSLP. I had already made this comment but it has been overlooked and/or misunderstood.

Provided this minor edit is made, I recommend that the Editor accepts this manuscript for publication.

Response: We sincerely appreciate your positive comments and valuable suggestions for enhancing the quality of our manuscript. In response to your feedback, we have removed the sentences pertaining to MSLP in the revised manuscript.